# Faster Randomized Infeasible Interior Point Methods for Tall/Wide Linear Programs

**Agniva Chowdhury**
Department of Statistics
Purdue University
West Lafayette, IN, USA
chowdhu5@purdue.edu

**Palma London**
ORIE Department
Cornell University
Ithaca, NY, USA
plondon@cornell.edu

**Haim Avron**
School of Mathematical Sciences
Tel Aviv University
Tel Aviv, Israel
haimav@tauex.tau.ac.il

**Petros Drineas**
Department of Computer Science
Purdue University
West Lafayette, IN, USA
pdrineas@purdue.edu

## Abstract

Linear programming (LP) is used in many machine learning applications, such as $\ell_1$-regularized SVMs, basis pursuit, nonnegative matrix factorization, etc. Interior Point Methods (IPMs) are one of the most popular methods to solve LPs both in theory and in practice. Their underlying complexity is dominated by the cost of solving a system of linear equations at each iteration. In this paper, we consider *infeasible* IPMs for the special case where the number of variables is much larger than the number of constraints (i.e., wide), or vice-versa (i.e., tall) by taking the dual. Using tools from Randomized Linear Algebra, we present a preconditioning technique that, when combined with the Conjugate Gradient iterative solver, provably guarantees that infeasible IPM algorithms (suitably modified to account for the error incurred by the approximate solver), converge to a feasible, approximately optimal solution, without increasing their iteration complexity. Our empirical evaluations verify our theoretical results on both real and synthetic data.

## 1 Introduction

Linear programming (LP) is one of the most useful tools available to theoreticians and practitioners throughout science and engineering. In Machine Learning, LP appears in numerous settings, including $\ell_1$-regularized SVMs [57], basis pursuit (BP) [54], sparse inverse covariance matrix estimation (SICE) [55], the nonnegative matrix factorization (NMF) [45], MAP inference [37], etc. Not surprisingly, designing and analyzing LP algorithms is a topic of paramount importance in computer science and applied mathematics.

One of the most successful paradigms for solving LPs is the family of Interior Point Methods (IPMs), pioneered by Karmarkar in the mid 1980s [25]. Path-following IPMs and, in particular, long-step path following IPMs, are among the most practical approaches for solving linear programs. Consider the standard form of the primal LP problem:

$$\min \mathbf{c}^\mathsf{T}\mathbf{x}\,, \text{ subject to } \mathbf{A}\mathbf{x} = \mathbf{b}\,, \mathbf{x} \geq \mathbf{0}\,, \tag{1}$$

where $\mathbf{A} \in \mathbb{R}^{m \times n}$, $\mathbf{b} \in \mathbb{R}^m$, and $\mathbf{c} \in \mathbb{R}^n$ are the inputs, and $\mathbf{x} \in \mathbb{R}^n$ is the vector of the primal variables. The associated dual problem is

$$\max \mathbf{b}^\mathsf{T}\mathbf{y}\,, \text{ subject to } \mathbf{A}^\mathsf{T}\mathbf{y} + \mathbf{s} = \mathbf{c}\,, \mathbf{s} \geq \mathbf{0}\,, \tag{2}$$

where $\mathbf{y} \in \mathbb{R}^m$ and $\mathbf{s} \in \mathbb{R}^n$ are the vectors of the dual and slack variables respectively. Triplets $(\mathbf{x}, \mathbf{y}, \mathbf{s})$ that uphold both (1) and (2) are called *primal-dual solutions*. Path-following IPMs typically converge towards a primal-dual solution by operating as follows: given the current iterate $(\mathbf{x}^k, \mathbf{y}^k, \mathbf{s}^k)$, they compute the Newton search direction $(\Delta\mathbf{x}, \Delta\mathbf{y}, \Delta\mathbf{s})$ and update the current iterate by following a step towards the search direction. To compute the search direction, one standard approach [41] involves solving the *normal equations*[1]:

$$\mathbf{A}\mathbf{D}^2\mathbf{A}^\mathsf{T}\Delta\mathbf{y} = \mathbf{p}. \tag{3}$$

Here, $\mathbf{D} = \mathbf{X}^{1/2}\mathbf{S}^{-1/2}$ is a diagonal matrix, $\mathbf{X}, \mathbf{S} \in \mathbb{R}^{n \times n}$ are diagonal matrices whose $i$-th diagonal entries are equal to $\mathbf{x}_i$ and $\mathbf{s}_i$, respectively, and $\mathbf{p} \in \mathbb{R}^m$ is a vector whose exact definition is given in eqn. (16)[2]. Given $\Delta\mathbf{y}$, computing $\Delta\mathbf{s}$ and $\Delta\mathbf{x}$ only involves matrix-vector products.

The core computational bottleneck in IPMs is the need to solve the linear system of eqn. (3) at each iteration. This leads to two key challenges: first, for high-dimensional matrices $\mathbf{A}$, solving the linear system is computationally prohibitive. Most implementations of IPMs use a *direct solver*; see Chapter 6 of [41]. However, if $\mathbf{A}\mathbf{D}^2\mathbf{A}^\mathsf{T}$ is large and dense, direct solvers are computationally impractical. If $\mathbf{A}\mathbf{D}^2\mathbf{A}^\mathsf{T}$ is sparse, specialized direct solvers have been developed, but these do not apply to many LP problems arising in machine learning applications due to irregular sparsity patterns. Second, an alternative to direct solvers is the use of iterative solvers, but the situation is further complicated since $\mathbf{A}\mathbf{D}^2\mathbf{A}^\mathsf{T}$ is typically ill-conditioned. Indeed, as IPM algorithms approach the optimal primal-dual solution, the diagonal matrix $\mathbf{D}$ is ill-conditioned, which also results in the matrix $\mathbf{A}\mathbf{D}^2\mathbf{A}^\mathsf{T}$ being ill-conditioned. Additionally, using approximate solutions for the linear system of eqn. (3) causes certain invariants, which are crucial for guaranteeing the convergence of IPMs, to be violated; see Section 1.1 for details.

In this paper, we address the aforementioned challenges, for the special case where $m \ll n$, i.e., the number of constraints is much smaller than the number of variables; see Appendix A for a generalization. This is a common setting in ML applications of LP solvers, since $\ell_1$-SVMs and basis pursuit problems often exhibit such structure when the number of available features ($n$) is larger than the number of objects ($m$). This setting has been of interest in recent work on LPs [17, 4, 31]. For simplicity of exposition, we also assume that the constraint matrix $\mathbf{A}$ has full rank, equal to $m$. First, we propose and analyze a preconditioned Conjugate Gradient (CG) iterative solver for the normal equations of eqn. (3), using matrix sketching constructions from the Randomized Linear Algebra (RLA) literature. We develop a preconditioner for $\mathbf{A}\mathbf{D}^2\mathbf{A}^\mathsf{T}$ using matrix sketching which allows us to prove strong convergence guarantees for the *residual* of CG solvers. Second, building upon the work of [39], we propose and analyze a provably accurate long-step *infeasible* IPM algorithm. The proposed IPM solves the normal equations using iterative solvers. In this paper, for brevity and clarity, we primarily focus our description and analysis on the CG iterative solver. We note that a non-trivial concern is that the use of iterative solvers and matrix sketching tools implies that the normal equations at each iteration will be solved only approximately. In our proposed IPM, we develop a novel way to *correct* for the error induced by the approximate solution in order to guarantee convergence. Importantly, this correction step is relatively computationally light, unlike a similar step proposed in [39]. Third, we empirically show that our algorithm performs well in practice. We consider solving LPs that arise from $\ell_1$-regularized SVMs and test them on a variety of synthetic and real datasets. Several extensions of our work are discussed in Appendix A.

## 1.1 Our contributions

Our point of departure in this work is the introduction of preconditioned, iterative solvers for solving eqn. (3). Preconditioning is used to address the ill-conditioning of the matrix $\mathbf{A}\mathbf{D}^2\mathbf{A}^\mathsf{T}$. Iterative solvers allow the computation of approximate solutions using only matrix-vector products while avoiding matrix inversion, Cholesky or LU factorizations, etc. A preconditioned formulation of eqn. (3) is:

$$\mathbf{Q}^{-1}\mathbf{A}\mathbf{D}^2\mathbf{A}^\mathsf{T}\Delta\mathbf{y} = \mathbf{Q}^{-1}\mathbf{p}, \tag{4}$$

where $\mathbf{Q} \in \mathbb{R}^{m \times m}$ is the preconditioning matrix; $\mathbf{Q}$ should be easily invertible (see [3, 22] for background). An alternative yet equivalent formulation of eqn. (4), which is more amenable to

theoretical analysis, is

$$\mathbf{Q}^{-1/2}\mathbf{A}\mathbf{D}^2\mathbf{A}^{\mathsf{T}}\mathbf{Q}^{-1/2}\mathbf{z} = \mathbf{Q}^{-1/2}\mathbf{p}, \tag{5}$$

where $\mathbf{z} \in \mathbb{R}^m$ is a vector such that $\Delta\mathbf{y} = \mathbf{Q}^{-1/2}\mathbf{z}$. Note that the matrix in the left-hand side of the above equation is always symmetric, which is not necessarily the case for eqn. (4). We do emphasize that one can use eqn. (4) in the actual implementation of the preconditioned solver; eqn. (5) is much more useful in theoretical analyses.

Recall that we focus on the special case where $\mathbf{A} \in \mathbb{R}^{m \times n}$ has $m \ll n$, i.e., it is a short-and-fat matrix. Our first contribution starts with the design and analysis of a preconditioner for the Conjugate Gradient solver that satisfies, with high probability,

$$\frac{2}{2+\zeta} \leq \sigma^2_{\min}(\mathbf{Q}^{-\frac{1}{2}}\mathbf{A}\mathbf{D}) \leq \sigma^2_{\max}(\mathbf{Q}^{-\frac{1}{2}}\mathbf{A}\mathbf{D}) \leq \frac{2}{2-\zeta}, \tag{6}$$

for some error parameter $\zeta \in [0, 1]$. In the above, $\sigma_{\min}(\cdot)$ and $\sigma_{\max}(\cdot)$ correspond to the smallest and largest singular value of the matrix in parentheses. The above condition says that the preconditioner effectively reduces the condition number of $\mathbf{A}\mathbf{D}$ to a constant. We note that the particular form of the lower and upper bounds in eqn. (6) was chosen to simplify our derivations. RLA matrix-sketching techniques allow us to construct preconditioners for all short-and-fat matrices that satisfy the above inequality *and* can be inverted efficiently. Such constructions go back to the work of [2]; see Section 2 for details on the construction of $\mathbf{Q}$ and its inverse. Importantly, given such a preconditioner, we then prove that the resulting CG iterative solver satisfies

$$\|\mathbf{Q}^{-1/2}\mathbf{A}\mathbf{D}^2\mathbf{A}^{\mathsf{T}}\mathbf{Q}^{-1/2}\tilde{\mathbf{z}}^t - \mathbf{Q}^{-1/2}\mathbf{p}\|_2 \leq \zeta^t\|\mathbf{Q}^{-1/2}\mathbf{p}\|_2. \tag{7}$$

Here $\tilde{\mathbf{z}}^t$ is the approximate solution returned by the CG iterative solver after $t$ iterations. In words, the above inequality states that the *residual* achieved after $t$ iterations of the CG iterative solver drops exponentially fast. To the best of our knowledge, this result is not known in the CG literature: indeed, it is actually well-known that the residual of CG may oscillate [21], even in cases where the energy norm of the solution error decreases monotonically. However, we prove that if the preconditioner is sufficiently good, i.e., it satisfies the constraint of eqn. (6), then the residual decreases as well.

Our second contribution is the analysis of a novel variant of a long-step *infeasible* IPM algorithm proposed by [39]. Recall that such algorithms can, in general, start with an initial point that is not necessarily feasible, but does need to satisfy some, more relaxed, constraints. Following the lines of [56, 39], let $\mathcal{S}$ be the set of feasible and optimal solutions of the form $(\mathbf{x}^*, \mathbf{y}^*, \mathbf{s}^*)$ for the primal and dual problems of eqns. (1) and (2) and assume that $\mathcal{S}$ is not empty. Then, long-step infeasible IPMs can start with any initial point $(\mathbf{x}^0, \mathbf{y}^0, \mathbf{s}^0)$ that satisfies $(\mathbf{x}^0, \mathbf{s}^0) > 0$ *and* $(\mathbf{x}^0, \mathbf{s}^0) \geq (\mathbf{x}^*, \mathbf{s}^*)$, for some feasible and optimal solution $(\mathbf{x}^*, \mathbf{s}^*) \in \mathcal{S}$. In words, the starting primal and slack variables must be strictly positive *and* larger (element-wise) when compared to some feasible, optimal primal-dual solution. See Chapter 6 of [52] for a discussion regarding why such choices of starting points are relevant to computational practice and can be identified more efficiently than feasible points.

The flexibility of infeasible IPMs comes at a cost: long-step *feasible* IPMs converge in $\mathcal{O}(n \log 1/\epsilon)$ iterations, while long-step *infeasible* IPMs need $\mathcal{O}(n^2 \log 1/\epsilon)$ iterations to converge [56, 39] (Here $\epsilon$ is the accuracy of the approximate LP solution returned by the IPM; see Algorithm 2 for the exact definition.). Let

$$\mathbf{A}\mathbf{x}^0 - \mathbf{b} = \mathbf{r}_p^0, \tag{8}$$

$$\mathbf{A}^{\mathsf{T}}\mathbf{y}^0 + \mathbf{s}^0 - \mathbf{c} = \mathbf{r}_d^0, \tag{9}$$

where $\mathbf{r}_p^0 \in \mathbb{R}^n$ and $\mathbf{r}_d^0 \in \mathbb{R}^m$ are the *primal* and *dual* residuals, respectively, and characterize how far the initial point is from being feasible. As long-step infeasible IPM algorithms iterate and update the primal and dual solutions, the residuals are updated as well. Let $\mathbf{r}^k = (\mathbf{r}_p^k, \mathbf{r}_d^k) \in \mathbb{R}^{n+m}$ be the primal and dual residual at the $k$-th iteration: it is well-known that the convergence analysis of infeasible long-step IPMs critically depends on $\mathbf{r}^k$ lying on the line segment between 0 and $\mathbf{r}^0$. Unfortunately, using approximate solvers (such as the CG solver proposed above) for the normal equations violates this invariant. [39] proposed a simple solution to fix this problem by adding a perturbation vector $\mathbf{v}$ to the current primal-dual solution that guarantees that the invariant is satisfied. Again, we use RLA matrix sketching principles to propose an efficient construction for $\mathbf{v}$ that provably satisfies the invariant. Next, we combine the above two primitives to prove that Algorithm 2 in Section 3 satisfies the following theorem.

**Theorem 1** *Let $0 \le \epsilon \le 1$ be an accuracy parameter. Consider the long-step infeasible IPM Algorithm 2 (Section 3) that solves eqn. (5) using the CG solver of Algorithm 1 (Section 2). Assume that the CG iterative solver runs with accuracy parameter $\zeta = {}^1\!/{}_2$ and iteration count $t = \mathcal{O}(\log n)$. Then, with probability at least 0.9, the long-step infeasible IPM converges after $\mathcal{O}(n^2 \log {}^1\!/{}_\epsilon)$ iterations.*

We note that the 0.9 success probability above is for simplicity of exposition and can be easily amplified using standard techniques. Also, at each iteration of our infeasible long-step IPM algorithm, the running time is $\mathcal{O}((\mathsf{nnz}(\mathbf{A}) + m^3) \log n)$, ignoring constant terms. See Section 3 for a detailed discussion of the overall running time.

Our empirical evaluation demonstrates that our algorithm requires an order of magnitude much fewer inner CG iterations than a standard IPM using CG, while producing a comparably accurate solution (see Section 4).

## 1.2 Prior Work

There is a large body of literature on solving LPs using IPMs. We only review literature that is immediately relevant to our work. Recall that we solve the normal equations inexactly at each iteration, and develop a way to *correct* for the error incurred. We also focus on IPMs that can use a sufficiently positive, infeasible initial point (see Section 1.1). We discuss below two papers that present related ideas.

[39] proposed the use of an approximate iterative solver for eqn. (3), followed by a correction step to "fix" the approximate solution (see our discussion in Section 1.1). We propose efficient, RLA-based approaches to precondition and solve eqn. (3), as well as a novel approach to correct for the approximation error in order to guarantee the convergence of the IPM algorithm. Specifically, [39] propose to solve eqn. (3) using the so-called *maximum weight basis* preconditioner [46]. However, computing such a preconditioner needs access to a maximal linearly independent set of columns of $\mathbf{AD}$ in each iteration, which is costly, taking $\mathcal{O}(m^2 n)$ time in the worst-case. More importantly, while [38] was able to provide a bound on the condition number of the preconditioned matrix that depends only on properties of $\mathbf{A}$, and is independent of $\mathbf{D}$, this bound might, in general, be very large. In contrast, our bound is a constant and it does not depend on properties of $\mathbf{A}$ or its dimensions. In addition, [39] assumed a bound on the two-norm of the residual of the preconditioned system, but it is unclear how their preconditioner guarantees such a bound. Similar concerns exist for the construction of the correction vector $\mathbf{v}$ proposed by [39], which our work alleviates.

The line of research in the Theoretical Computer Science literature that is closest to our work is [15], who presented an IPM that uses an approximate solver in each iteration. However, their accuracy guarantee is in terms of the final objective value which is different from ours. More importantly, [15] focuses on *short-step*, feasible IPMs, whereas ours is *long-step* and does not require a feasible starting point. Finally, the approximate solver proposed by [15] works only for the special case of input matrices that correspond to graph Laplacians, following the lines of [47, 48].

We also note that in the Theoretical Computer Science literature, [26, 27, 28, 29, 30, 7, 12] proposed and analyzed theoretically ground-breaking algorithms for LPs based on novel tools such as the so-called *inverse maintenance* for accelerating the linear system solvers in IPMs. However, all these endeavors are primarily focused on the theoretically fast but practically inefficient short-step feasible IPMs and, to the best of our knowledge, no implementations of these approaches are available for comparisons to standard long-step IPMs. We highlight that our work is focused on infeasible *long-step* IPMs, known to work efficiently in practice.

Another relevant line of research is the work of [14], which proposed solving eqn. (3) using preconditioned Krylov subspace methods, including variants of *generalized minimum residual* (GMRES) or CG methods. Indeed, [14] conducted extensive numerical experiments on LP problems taken from standard benchmark libraries, but did not provide any theoretical guarantees.

From a matrix-sketching perspective, our work was also partially motivated by [8], which presented an iterative, sketching-based algorithm to solve under-constrained ridge regression problems, but did not address how to make use of such approaches in an IPM-based framework, as we do here. In another work, [1] proposed a similar sketching-based preconditioning technique. However, their efforts broadly revolved around speeding up and scaling *kernel ridge regression*. [43, 53] proposed the so-called *Newton sketch* to construct an approximate Hessian matrix for more general convex objective functions of which LP is a special case. Nevertheless, these randomized second-order

methods are significantly faster than the conventional approach only when the data matrix is over-constrained, *i.e.* $m \gg n$. It is unclear whether the approach of [43, 53] is faster than IPMs when the optimization problem to be solved is linear. [49] proposed a probabilistic algorithm to solve LP approximately in a random projection-based reduced feature-space. A possible drawback of this paper is that the approximate solution is infeasible with respect to the original region. Finally, we refer the interested reader to the surveys [51, 19, 33, 18, 24, 34] for more background on Randomized Linear Algebra.

## 1.3 Notation and Background

$\mathbf{A}, \mathbf{B}, \ldots$ denote matrices and $\mathbf{a}, \mathbf{b}, \ldots$ denote vectors. For vector $\mathbf{a}$, $\|\mathbf{a}\|_2$ denotes its Euclidean norm; for a matrix $\mathbf{A}$, $\|\mathbf{A}\|_2$ denotes its spectral norm and $\|\mathbf{A}\|_F$ denotes its Frobenius norm. We use $\mathbf{0}$ to denote a null vector or null matrix, dependent upon context, and $\mathbf{1}$ to denote the all-ones vector. For any matrix $\mathbf{X} \in \mathbb{R}^{m \times n}$ with $m \le n$ of rank $m$ its thin Singular Value Decomposition (SVD) is the product $\mathbf{U\Sigma V^\mathsf{T}}$, with $\mathbf{U} \in \mathbb{R}^{m \times m}$ (the matrix of the left singular vectors), $\mathbf{V} \in \mathbb{R}^{n \times m}$ ( the matrix of the top-$m$ right singular vectors), and $\mathbf{\Sigma} \in \mathbb{R}^{m \times m}$ a diagonal matrix whose entries are equal to the singular values of $\mathbf{X}$. We use $\sigma_i(\cdot)$ to denote the $i$-th singular value of the matrix in parentheses.

We now briefly discuss a result on matrix sketching [13, 11] that is particularly useful in our theoretical analyses. In our parlance, [13] proved that, for any matrix $\mathbf{Z} \in \mathbb{R}^{m \times n}$, there exists a sketching matrix $\mathbf{W} \in \mathbb{R}^{n \times w}$ such that

$$\left\| \mathbf{ZWW^\mathsf{T}Z^\mathsf{T}} - \mathbf{ZZ^\mathsf{T}} \right\|_2 \le \frac{\zeta}{4} \left( \|\mathbf{Z}\|_2^2 + \frac{\|\mathbf{Z}\|_F^2}{r} \right) \tag{10}$$

holds with probability at least $1 - \delta$ for any $r \ge 1$. Here $\zeta \in [0, 1]$ is a (constant) accuracy parameter. Ignoring constant terms, $w = \mathcal{O}(r \log(r/\delta))$; $\mathbf{W}$ has $s = \mathcal{O}(\log(r/\delta))$ non-zero entries per row with $s$ uniformly random entries are chosen without replacement and set to $\pm\frac{1}{s}$ independently; the product $\mathbf{ZW}$ can be computed in time $\mathcal{O}(\log(r/\delta) \cdot \mathsf{nnz}(\mathbf{Z}))$.

## 2 Conjugate Gradient Solver

In this section, we discuss the computation of the preconditioner $\mathbf{Q}$ (and its inverse), followed by a discussion on how such a preconditioner can be used to satisfy eqns. (6) and (7).

---

**Algorithm 1** Solving eqn. (5) via CG

---

    **Input:** $\mathbf{AD} \in \mathbb{R}^{m \times n}$, $\mathbf{p} \in \mathbb{R}^m$, sketching matrix $\mathbf{W} \in \mathbb{R}^{n \times w}$, iteration count $t$;

1: Compute $\mathbf{ADW}$ and its SVD: let $\mathbf{U_Q}$ be the matrix of its left singular vectors and let $\mathbf{\Sigma_Q^{1/2}}$ be the matrix of its singular values;
2: Compute $\mathbf{Q}^{-1/2} = \mathbf{U_Q \Sigma_Q^{-1/2} U_Q^\mathsf{T}}$;
3: Initialize $\tilde{\mathbf{z}}^0 \leftarrow \mathbf{0}_m$ and run standard CG on the preconditioned system of eqn. (5) for $t$ iterations;
    **Output:** $\tilde{\mathbf{z}}^t$;

---

Algorithm 1 takes as input the sketching matrix $\mathbf{W} \in \mathbb{R}^{n \times w}$, which we construct as discussed in Section 1.3. Our preconditioner $\mathbf{Q}$ is equal to

$$\mathbf{Q} = \mathbf{ADWW^\mathsf{T}DA^\mathsf{T}}. \tag{11}$$

Notice that we only need to compute $\mathbf{Q}^{-1/2}$ in order to use it to solve eqn. (5). Towards that end, we first compute the sketched matrix $\mathbf{ADW} \in \mathbb{R}^{m \times w}$. Then, we compute the SVD of the matrix $\mathbf{ADW}$: let $\mathbf{U_Q}$ be the matrix of its left singular vectors and let $\mathbf{\Sigma_Q^{1/2}}$ be the matrix of its singular values. Notice that the left singular vectors of $\mathbf{Q}^{-1/2}$ are equal to $\mathbf{U_Q}$ and its singular values are equal to $\mathbf{\Sigma_Q^{-1/2}}$. Therefore, $\mathbf{Q}^{-1/2} = \mathbf{U_Q \Sigma_Q^{-1/2} U_Q^\mathsf{T}}$.

Let $\mathbf{AD} = \mathbf{U\Sigma V^\mathsf{T}}$ be the thin SVD representation of $\mathbf{AD}$. We apply the results of [13] (see Section 1.3) to the matrix $\mathbf{Z} = \mathbf{V}^\mathsf{T} \in \mathbb{R}^{m \times n}$ with $r = m$ to get that, with probability at least $1 - \delta$,

$$\left\| \mathbf{V^\mathsf{T}WW^\mathsf{T}V} - \mathbf{I}_m \right\|_2 \le \zeta/2 \tag{12}$$

The running time needed to compute the sketch $\mathbf{ADW}$ is equal to (ignoring constant factors) $\mathcal{O}(\mathsf{nnz}(\mathbf{A}) \cdot \log(m/\delta))$. Note that $\mathsf{nnz}(\mathbf{AD}) = \mathsf{nnz}(\mathbf{A})$. The cost of computing the SVD of $\mathbf{ADW}$ (and therefore $\mathbf{Q}^{-1/2}$) is $\mathcal{O}(m^3 \log(m/\delta))$. Overall, computing $\mathbf{Q}^{-1/2}$ can be done in time

$$\mathcal{O}(\mathsf{nnz}(\mathbf{A}) \cdot \log(m/\delta) + m^3 \log(m/\delta)). \tag{13}$$

Given these results, we now discuss how to satisfy eqns. (6) and (7) using the sketching matrix $\mathbf{W}$. We start with the following bound, which is relatively straight-forward given prior RLA work (see Appendix C.1 for a proof).

**Lemma 2** *If the sketching matrix $\mathbf{W}$ satisfies eqn. (12), then, for all $i = 1 \ldots m$,*

$$(1 + \zeta/2)^{-1} \leq \sigma_i^2(\mathbf{Q}^{-1/2}\mathbf{AD}) \leq (1 - \zeta/2)^{-1}.$$

This lemma directly implies eqn. (6). We now proceed to show that the above construction for $\mathbf{Q}^{-1/2}$, when combined with the conjugate gradient solver to solve eqn. (5), indeed satisfies eqn. (7)[3]. We do note that in prior work most of the convergence guarantees for CG focus on the error of the approximate solution. However, in our work, we are interested in the convergence of the *residuals* and it is known that even if the energy norm of the error of the approximate solution decreases monotonically, the norms of the CG residuals may oscillate. Interestingly, we can combine a result on the residuals of CG from [6] with Lemma 2 to prove that in our setting the norms of the CG residuals also decrease monotonically (see Appendix C.2 for details).

We remark that one can consider using MINRES [42] instead of CG. Our results hinges on bounding the two-norm of the residual. MINRES finds, at each iteration, the optimal vector with respect the two-norm of the residual inside the same Krylov subspace of CG for the corresponding iteration. Thus, the bound we prove for CG applies to MINRES as well.

## 3 The Infeasible IPM algorithm

In order to avoid spurious solutions, primal-dual path-following IPMs bias the search direction towards the *central path* and restrict the iterates to a neighborhood of the central path. This search is controlled by the *centering parameter* $\sigma \in [0, 1]$. At each iteration, given the current solution $(\mathbf{x}^k, \mathbf{y}^k, \mathbf{s}^k)$, a standard infeasible IPM obtains the search direction $(\Delta\mathbf{x}^k, \Delta\mathbf{y}^k, \Delta\mathbf{s}^k)$ by solving the following system of linear equations:

$$\mathbf{AD}^2\mathbf{A}^\mathsf{T}\Delta\mathbf{y}^k = \mathbf{p}^k, \tag{14a}$$

$$\Delta\mathbf{s}^k = -\mathbf{r}_d^k - \mathbf{A}^\mathsf{T}\Delta\mathbf{y}^k, \tag{14b}$$

$$\Delta\mathbf{x}^k = -\mathbf{x}^k + \sigma\mu_k\mathbf{S}^{-1}\mathbf{1}_n - \mathbf{D}^2\Delta\mathbf{s}^k. \tag{14c}$$

Here $\mathbf{D}$ and $\mathbf{S}$ are computed given the current iterate ($\mathbf{x}^k$ and $\mathbf{s}^k$). After solving the above system, the infeasible IPM Algorithm 2 proceeds by computing a step-size $\bar{\alpha}$ to return:

$$(\mathbf{x}^{k+1}, \mathbf{y}^{k+1}, \mathbf{s}^{k+1}) = (\mathbf{x}^k, \mathbf{y}^k, \mathbf{s}^k) + \bar{\alpha}(\Delta\mathbf{x}^k, \Delta\mathbf{y}^k, \Delta\mathbf{s}^k). \tag{15}$$

Recall that $\mathbf{r}^k = (\mathbf{r}_p^k, \mathbf{r}_d^k)$ is a vector with $\mathbf{r}_p^k = \mathbf{A}\mathbf{x}^k - \mathbf{b}$ and $\mathbf{r}_d^k = \mathbf{A}^\mathsf{T}\mathbf{y}^k + \mathbf{s}^k - \mathbf{c}$ (the primal and dual residuals). We also use the *duality measure* $\mu_k = \mathbf{x}^{k\mathsf{T}}\mathbf{s}^k/n$ and the vector

$$\mathbf{p}^k = -\mathbf{r}_p^k - \sigma\mu_k\mathbf{A}\mathbf{S}^{-1}\mathbf{1}_n + \mathbf{A}\mathbf{x}^k - \mathbf{A}\mathbf{D}^2\mathbf{r}_d^k. \tag{16}$$

Given $\Delta\mathbf{y}^k$ from eqn. (14a), $\Delta\mathbf{s}^k$ and $\Delta\mathbf{x}^k$ are easy to compute from eqns. (14b) and (14c), as they only involve matrix-vector products. However, since we will use Algorithm 1 to solve eqn. (14a) approximately using the sketching-based preconditioned CG solver, the primal and dual residuals *do not* lie on the line segment between $\mathbf{0}$ and $\mathbf{r}^0$. This invalidates known proofs of convergence for infeasible IPMs.

For notational simplicity, we now drop the dependency of vectors and scalars on the iteration counter $k$. Let $\hat{\Delta\mathbf{y}} = \mathbf{Q}^{-1/2}\tilde{\mathbf{z}}^t$ be the approximate solution to eqn. (14a). In order to account for the loss of accuracy due to the approximate solver, we compute $\hat{\Delta\mathbf{x}}$ as follows:

$$\hat{\Delta\mathbf{x}} = -\mathbf{x} + \sigma\mu\mathbf{S}^{-1}\mathbf{1}_n - \mathbf{D}^2\hat{\Delta\mathbf{s}} - \mathbf{S}^{-1}\mathbf{v}. \tag{17}$$

Here $\mathbf{v} \in \mathbb{R}^n$ is a perturbation vector that needs to exactly satisfy the following invariant at each iteration of the infeasible IPM:

$$\mathbf{A}\mathbf{S}^{-1}\mathbf{v} = \mathbf{A}\mathbf{D}^2\mathbf{A}^\mathsf{T}\hat{\Delta}\mathbf{y} - \mathbf{p}. \tag{18}$$

We note that the computation of $\hat{\Delta}\mathbf{s}$ is still done using eqn. (14b), which does not change. [39] argued that if $\mathbf{v}$ satisfies eqn. (18), the primal and dual residuals lie in the correct line segment.

**Construction of v.** There are many choices for $\mathbf{v}$ satisfying eqn. (18). A general choice is $\mathbf{v} = (\mathbf{A}\mathbf{S}^{-1})^\dagger(\mathbf{A}\mathbf{D}^2\mathbf{A}^\mathsf{T}\hat{\Delta}\mathbf{y} - \mathbf{p})$, which involves the computation of the pseudoinverse $(\mathbf{A}\mathbf{S}^{-1})^\dagger$, which is expensive, taking time $\mathcal{O}(m^2 n)$. Instead, we propose to construct $\mathbf{v}$ using the sketching matrix $\mathbf{W}$ of Section 1.3. More precisely, we construct the perturbation vector

$$\mathbf{v} = (\mathbf{X}\mathbf{S})^{1/2}\mathbf{W}(\mathbf{A}\mathbf{D}\mathbf{W})^\dagger(\mathbf{A}\mathbf{D}^2\mathbf{A}^\mathsf{T}\hat{\Delta}\mathbf{y} - \mathbf{p}). \tag{19}$$

The following lemma proves that the proposed $\mathbf{v}$ satisfies eqn. (18); see Appendix C.3 for the proof.

**Lemma 3** *Let $\mathbf{W} \in \mathbb{R}^{n \times w}$ be the sketching matrix of Section 1.3 and $\mathbf{v}$ be the perturbation vector of eqn. (19). Then, with probability at least $1 - \delta$, $\mathrm{rank}(\mathbf{A}\mathbf{D}\mathbf{W}) = m$ and $\mathbf{v}$ satisfies eqn. (18).*

We emphasize here that we will use the same exact sketching matrix $\mathbf{W} \in \mathbb{R}^{n \times w}$ to form the preconditioner used in the CG algorithm of Section 2 *as well as* the vector $\mathbf{v}$ in eqn.(19). This allows us to form the sketching matrix only once, thus saving time in practice. Next, we present a bound for the two-norm of the perturbation vector $\mathbf{v}$ of eqn. (19); see Appendix C.4 for the proof.

**Lemma 4** *With probability at least $1 - \delta$, our perturbation vector $\mathbf{v}$ in Lemma 3 satisfies*

$$\|\mathbf{v}\|_2 \leq \sqrt{3n\mu}\,\|\tilde{\mathbf{f}}^{(t)}\|_2, \tag{20}$$

*with* $\tilde{\mathbf{f}}^{(t)} = \mathbf{Q}^{-1/2}\mathbf{A}\mathbf{D}^2\mathbf{A}^\mathsf{T}\mathbf{Q}^{-1/2}\tilde{\mathbf{z}}^t - \mathbf{Q}^{-1/2}\mathbf{p}$.

Intuitively, the bound in eqn. (20) implies that $\|\mathbf{v}\|_2$ depends on how close the approximate solution $\hat{\Delta}\mathbf{y}$ is to the exact solution. Lemma 4 is particularly useful in proving the convergence of Algorithm 2, which needs $\|\mathbf{v}\|_2$ to be a small quantity. More precisely, combining a result from [39] with our preconditioner $\mathbf{Q}^{-1/2}$, we can prove that $\|\mathbf{Q}^{-1/2}\mathbf{p}\|_2 \leq \mathcal{O}(n)\sqrt{\mu}$. This bound allows us to prove that if we run Algorithm 1 for $\mathcal{O}(\log n)$ iterations, then $\|\tilde{\mathbf{f}}^{(t)}\|_2 \leq \frac{\gamma\sigma}{4\sqrt{n}}\sqrt{\mu}$ and $\|\mathbf{v}\|_2 \leq \frac{\gamma\sigma}{4}\mu$. The last two inequalities are critical in the convergence analysis of Algorithm 2; see Appendix F.1 and Appendix F.2 for details.

We are now ready to present the infeasible IPM algorithm. We will need the following definition for the neighborhood $\mathcal{N}(\gamma) = \{(\mathbf{x}^k, \mathbf{y}^k, \mathbf{s}^k) : (\mathbf{x}^k, \mathbf{s}^k) > \mathbf{0}, x_i^k s_i^k \geq (1-\gamma)\mu \text{ and } \|\mathbf{r}^k\|_2/\|\mathbf{r}^0\|_2 \leq \mu_k/\mu_0\}$. Here $\gamma \in (0, 1)$ and we note that the duality measure $\mu_k$ steadily reduces at each iteration.

---

**Algorithm 2** Infeasible IPM

---

    **Input:** $\mathbf{A} \in \mathbb{R}^{m \times n}$, $\mathbf{b} \in \mathbb{R}^m$, $\mathbf{c} \in \mathbb{R}^n$, $\gamma \in (0, 1)$, tolerance $\epsilon > 0$, $\sigma \in (0, 4/5)$;
    **Initialize:** $k \leftarrow 0$; initial point $(\mathbf{x}^0, \mathbf{y}^0, \mathbf{s}^0)$;
1: **while** $\mu_k > \epsilon$ **do**
2:     Compute sketching matrix $\mathbf{W} \in \mathbb{R}^{n \times w}$ (Section 1.3) with $\zeta = 1/2$ and $\delta = O(n^{-2})$;
3:     Compute $\mathbf{r}_p^k = \mathbf{A}\mathbf{x}^k - \mathbf{b}$; $\mathbf{r}_d^k = \mathbf{A}^\mathsf{T}\mathbf{y}^k + \mathbf{s}^k - \mathbf{c}$; and $\mathbf{p}^k$ from eqn. (16);
4:     Solve the linear system of eqn. (5) for $\mathbf{z}$ using Algorithm 1 with $\mathbf{W}$ from step (2) and $t = \mathcal{O}(\log n)$. Compute $\hat{\Delta}\mathbf{y} = \mathbf{Q}^{-1/2}\mathbf{z}$;
5:     Compute $\mathbf{v}$ using eqn. (19) with $\mathbf{W}$ from step (2); $\hat{\Delta}\mathbf{s}$ using eqn. (14b); $\hat{\Delta}\mathbf{x}$ using eqn. (17);
6:     Compute $\tilde{\alpha} = \mathrm{argmax}\{\alpha \in [0, 1] : (\mathbf{x}^k, \mathbf{y}^k, \mathbf{s}^k) + \alpha(\hat{\Delta}\mathbf{x}^k, \hat{\Delta}\mathbf{y}^k, \hat{\Delta}\mathbf{s}^k) \in \mathcal{N}(\gamma)\}$.
7:     Compute $\bar{\alpha} = \mathrm{argmin}\{\alpha \in [0, \tilde{\alpha}] : (\mathbf{x}^k + \alpha\hat{\Delta}\mathbf{x}^k)^\mathsf{T}(\mathbf{s}^k + \alpha\hat{\Delta}\mathbf{s}^k)\}$.
8:     Compute $(\mathbf{x}^{k+1}, \mathbf{y}^{k+1}, \mathbf{s}^{k+1}) = (\mathbf{x}^k, \mathbf{y}^k, \mathbf{s}^k) + \bar{\alpha}(\hat{\Delta}\mathbf{x}^k, \hat{\Delta}\mathbf{y}^k, \hat{\Delta}\mathbf{s}^k)$; set $k \leftarrow k + 1$;
9: **end while**

---

**Running time of Algorithm 2.** We start by discussing the running time to compute $\mathbf{v}$. As discussed in Section 2, $(\mathbf{A}\mathbf{D}\mathbf{W})^\dagger$ can be computed in $\mathcal{O}(\mathrm{nnz}(\mathbf{A}) \cdot \log(m/\delta) + m^3 \log(m/\delta))$ time. Now, as

**W** has $\mathcal{O}(\log(m/\delta))$ non-zero entries per row, pre-multiplying by **W** takes $\mathcal{O}(\mathsf{nnz}(\mathbf{A})\log(m/\delta))$ time (assuming $\mathsf{nnz}(\mathbf{A}) \geq n$). Since **X** and **S** are diagonal matrices, computing **v** takes $\mathcal{O}(\mathsf{nnz}(\mathbf{A}) \cdot \log(m/\delta) + m^3 \log(m/\delta))$ time, which is asymptotically the same as computing $\mathbf{Q}^{-1/2}$ (see eqn. (13)).

We now discuss the overall running time of Algorithm 2. At each iteration, with failure probability $\delta$, the preconditioner $\mathbf{Q}^{-1/2}$ and the vector **v** can be computed in $\mathcal{O}(\mathsf{nnz}(\mathbf{A}) \cdot \log(m/\delta) + m^3 \log(m/\delta))$ time. In addition, for $t = \mathcal{O}(\log n)$ iterations of Algorithm 1, all the matrix-vector products in the CG solver can be computed in $\mathcal{O}(\mathsf{nnz}(\mathbf{A}) \cdot \log n)$ time. Therefore, the computational time for steps (2)-(5) is given by $\mathcal{O}(\mathsf{nnz}(\mathbf{A}) \cdot (\log n + \log(m/\delta)) + m^3 \log(m/\delta))$. Finally, taking a union bound over all iterations with $\delta = \mathcal{O}(n^{-2})$ (ignoring constant factors), Algorithm 2 converges with probability at least 0.9. The running time at each iteration is given by $\mathcal{O}((\mathsf{nnz}(\mathbf{A}) + m^3) \log n)$.

## 4  Experiments

We demonstrate the empirical performance of our algorithm on a variety of synthetic and real-world datasets from the UCI ML Repository [20], such as ARCENE, DEXTER [23], DrivFace [16], and a gene expression cancer RNA-Sequencing dataset that is part of the PANCAN dataset [50]. See Appendix G, Table 1 for a description of the datasets. We observed that the results for both synthetic (Appendix G.2) and real-world data were qualitatively similar; we highlight results on representative real datasets. The experiments were implemented in Python and run on a server with Intel E5-2623V3@3.0GHz 8 cores and 64GB RAM. As an application, we consider $\ell_1$-regularized SVMs: all of the datasets are concerned with binary classification with $m \ll n$, where $n$ is the number of features. In Appendix G.1, we describe the $\ell_1$-SVM problem and how it can be formulated as an LP. Here, $m$ is the number of training points, $n$ is the feature dimension, and the size of the constraint matrix in the LP becomes $m \times (2n + 1)$.

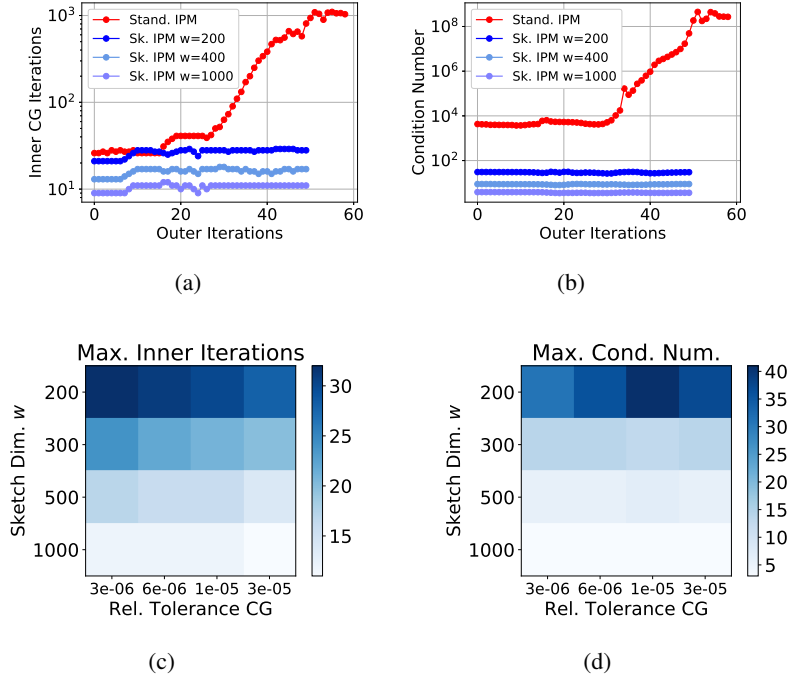

(a)    (b)

(c)    (d)

Figure 1: *ARCENE data set:* Our Algorithm 2 (Sk. IPM) requires an order of magnitude fewer (a) inner iterations than the Standard IPM with CG, at each outer iteration, due to the improved (b) conditioning of $\mathbf{Q}^{-1/2}\mathbf{A}\mathbf{D}^2\mathbf{A}^\mathsf{T}\mathbf{Q}^{-1/2}$ compared to $\mathbf{A}\mathbf{D}^2\mathbf{A}^T$. For various $(w, tolCG)$ settings, (c) the maximum number of inner iterations used by our algorithm and (d) the maximum condition number of $\mathbf{Q}^{-1/2}\mathbf{A}\mathbf{D}^2\mathbf{A}^\mathsf{T}\mathbf{Q}^{-1/2}$, across outer iterations. The standard IPM, across all settings, needed on the order of 1,000 iterations and $\kappa(\mathbf{A}\mathbf{D}^2\mathbf{A}^\mathsf{T})$ was on the order of $10^8$.

**Experimental Results**. We compare our Algorithm 2 with a standard IPM (see Chapter 10, [44]) using CG and a standard IPM using a direct solver. We also use CVXPY as a benchmark to compare the accuracy of the solutions; we define the *relative error* $\|\hat{\mathbf{x}} - \mathbf{x}^\star\|_2 / \|\mathbf{x}^\star\|_2$, where $\hat{\mathbf{x}}$ is our solution and $\mathbf{x}^\star$ is the solution generated by CVXPY. We also consider the number of *outer iterations*, namely the number of iterations of the IPM algorithm, as well as the number of *inner iterations*, namely the number of iterations of the CG solver. We denote the relative stopping tolerance for CG by *tolCG* and we denote the outer iteration residual by $\tau$. If not specified: $\tau = 10^{-9}$, *tolCG* $= 10^{-5}$, and $\sigma = 0.5$. We evaluated a Gaussian sketching matrix and the initial triplet $(\mathbf{x}, \mathbf{y}, \mathbf{s})$ for all IPM algorithms was set to be all ones.

Figure 1(a) shows that our Algorithm 2 uses an order of magnitude fewer *inner* iterations than the un-preconditioned standard solver. This is due to the improved conditioning of the respective matrices in the normal equations, as demonstrated in Figure 1(b). Across various real and synthetic data sets, the results were qualitatively similar to those shown in Figure 1. Results for several real data sets are summarized in Appendix G, Table 1. The number of *outer* iterations is unaffected by our internal approximation methods and is generally the same for our Algorithm 2, the standard IPM with CG, and the standard IPM with a direct linear solver (denoted IPM w/Dir), as seen in Appendix G, Table 1. Figure 1 also demonstrates the relative insensitivity to the choice of $w$ (the sketching dimension, i.e., the number of columns of the sketching matrix $\mathbf{W}$ of Section 1.3). For smaller values of $w$, our algorithm requires more inner iterations. However, across various choices of $w$, the number of inner iterations is always an order of magnitude smaller than the number required by the standard solver.

Figures 1(c)-1(d) show the performance of our algorithm for a range of $(w, \textit{tolCG})$ pairs. Figure 1(c) demonstrates that the number of the inner iterations is robust to the choice of *tolCG* and $w$. The number of inner iterations varies between $15$ and $35$ for the ARCENE data set, while the standard IPM took on the order of $1,000$ iterations across all parameter settings. Across all settings, the relative error was fixed at $0.04\%$. In general, our sketched IPM is able to produce an extremely high accuracy solution across parameter settings. Thus we do not report additional numerical results for the relative error, which was consistently $10^{-3}$ or less. Figure 1(d) demonstrates a tradeoff of our approach: as both *tolCG* and $w$ are increased, the condition number $\kappa(\mathbf{Q}^{-1/2}\mathbf{A}\mathbf{D}^2\mathbf{A}^\top\mathbf{Q}^{-1/2})$ decreases, corresponding to better conditioned systems. As a result, fewer inner iterations are required. Additional experiments can be found in Appendix G.4.

## 5 Conclusions

We proposed and analyzed an infeasible IPM algorithm using a preconditioned conjugate gradient solver for the normal equations and a novel perturbation vector to correct for the error due to the approximate solver. Thus, we speed up each iteration of the IPM algorithm, without increasing the overall number of iterations. We demonstrate empirically that our IPM requires an order of magnitude fewer inner iterations within each linear solve than standard IPMs. Several extensions of our work are discussed in Appendix A.

## Broader Impact

Our work is focused on speeding up algorithms for tall/wide LPs. As such, it could have significant broader impacts by allowing users to solve increasingly larger LPs in the numerous settings discussed in our introduction. While applications of our work to real data could result into ethical considerations, this is an indirect (and unpredictable) side-effect of our work. Our experimental work uses publicly available datasets to evaluate the performance of our algorithms; no ethical considerations are raised.

**Acknowledgements,** We thank the anonymous reviewers for their helpful comments. AC and PD were partially supported by NSF FRG 1760353 and NSF CCF-BSF 1814041. HA was partially supported by BSF grant 2017698. PL was supported by an Amazon Graduate Fellowship in Artificial Intelligence.

## Footnotes

[1]Another widely used approach is to solve the augmented system [41] which is less relevant for this paper.

[2]The superscript $k$ in eqn. (16) simply indicates iteration count and is omitted here for notational simplicity.

[3]See Chapter 9 of [32] for a detailed overview of CG.

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
