[Supplementary Material 1 · IPM_supp.pdf]

# Appendix to
# Faster Randomized Infeasible Interior Point Methods for Tall/Wide Linear Programs

## Appendix A    Extensions

We briefly discuss extensions of our work. First, there is nothing special about using a CG solver for solving eqn. (5). We analyze two more solvers that could replace the proposed CG solver without any loss in accuracy or any increase in the number of iterations for the long-step infeasible IPM Algorithm 2 of Section 3. In Appendix D, we analyze the performance of the preconditioned Richardson Iteration and in Appendix E, we analyze the performance of the preconditioned Steepest Descent. In both cases, if the respective preconditioned solver (with the preconditioner of Section 2) runs for $t = \mathcal{O}(\log n)$ steps, Theorem 1 still holds, with small differences in the constant terms. While preconditioned Richardson iteration and preconditioned Steepest Descent are interesting from a theoretical perspective, they are not particularly practical.

Second, recall that our approach focused on full rank input matrices $\mathbf{A} \in \mathbb{R}^{m \times n}$ with $m \ll n$. Our overall approach still works if $\mathbf{A}$ in any $m \times n$ matrix that is low-rank, e.g., $\mathrm{rank}(\mathbf{A}) = k \ll \min\{m, n\}$. In that case, using the thin SVD of $\mathbf{A}$, we can rewrite the linear constraints as follows $\mathbf{U_A \Sigma_A V_A^\mathsf{T} x = b}$, where $\mathbf{U_A} \in \mathbb{R}^{m \times k}$ and $\mathbf{V_A} \in \mathbb{R}^{n \times k}$ are the matrices of left and right singular vecors of $\mathbf{A}$ respectively; $\mathbf{\Sigma_A} \in \mathbb{R}^{k \times k}$ is the diagonal matrix with the $k$ non-zero singular values of $\mathbf{A}$ as its diagonal elements. The LP of eqn. (1) can be restated as

$$\min \mathbf{c}^\mathsf{T}\mathbf{x}, \text{ subject to } \mathbf{V_A^\mathsf{T} x} = \widetilde{\mathbf{b}}, \mathbf{x} \geq \mathbf{0}, \tag{21}$$

where $\widetilde{\mathbf{b}} = \mathbf{\Sigma_A^{-1} U_A^\mathsf{T} b}$. Note that, $\mathrm{rank}(\mathbf{V_A}) = k \ll n$ and therefore eqn. (21) can be solved using our framework. The matrices $\mathbf{U_A}$, $\mathbf{V_A}$, and $\mathbf{\Sigma_A}$ can be approximately recovered using the fast SVD algorithms of [23, 5, 10]. However, the accuracy of the final solution will depend on the accuracy of the approximate SVD and we defer this analysis to future work.

Third, even though we chose to use the Count-Min sketch and its analysis from [13] (Section 1.3), there are many other alternative sketching matrix constructions that would lead to similar results. A particularly simple one is the Gaussian sketching matrix $\mathbf{W}_G \in \mathbb{R}^{n \times w}$, where every entry is a $\mathcal{N}(0, 1)$ random variable. Setting $w = \mathcal{O}\left(\frac{m + \log(1/\delta)}{\zeta^2}\right)$ would result in the same accuracy guarantees as the sketching matrix of Section 1.3. However, the (theoretical) running time needed to compute $\mathbf{ADW}$ increases to $\mathcal{O}(m \cdot \mathsf{nnz}(\mathbf{A}))$. In practice, at least for relatively small matrices, using Gaussian sketching matrices is a reasonable alternative; see the discussion in [35] which argued that the Gaussian matrix sketching-based solvers are considerably better than direct solvers. We also opted to use Gaussian matrices in our empirical evaluation, since we primarily interested in measuring the accuracy of the final solution as a function of the number of iterations of the solver and the IPM algorithm. Other known constructions of sketching matrices that are also applicable in our setting include (any) sub-gaussian sketching matrix; the Subsampled Randomized Hadamard transform (SRHT); and any of the Sparse Subspace Embeddings of [9, 39, 34, 11].

We conclude by noting that our work can also be extended to analyze feasible IPMs, namely Algorithm 2 can start with a strictly feasible point. In this case, the analysis is somewhat simpler and the iteration complexity of the IPM algorithm reduces to $\mathcal{O}(n \log(1/\epsilon))$, which is the best known for feasible long-step path following IPM algorithms. We chose to present the more technically challenging infeasible IPM in this paper and delegate the feasible case to future work.

## Appendix B    Additional Notations

As before, we take $\mathbf{AD} = \mathbf{U\Sigma V^\mathsf{T}}$ to be the thin SVD representation of $\mathbf{AD}$. Additionally, for any two symmetric positive semidefinite (positive definite) matrices $\mathbf{A}_1$ and $\mathbf{A}_2$ with same order, $\mathbf{A}_1 \preccurlyeq \mathbf{A}_2$ ($\mathbf{A}_1 \prec \mathbf{A}_2$) denotes that $\mathbf{A}_2 - \mathbf{A}_1$ is positive semidefinite (positive definite). For any two vectors $\mathbf{a} = (a_1, \dots, a_\ell)^\mathsf{T}$ and $\mathbf{b} = (b_1, \dots, b_\ell)^\mathsf{T}$ let $\mathbf{a} \circ \mathbf{b} = (a_1 b_1, \dots, a_\ell b_\ell)^\mathsf{T}$. For any vector $\mathbf{a} \in \mathbb{R}^n$ its $\ell_\infty$ norm is defined as $\|\mathbf{a}\|_\infty = \max_i |a_i|$.

## Appendix C Proofs

### C.1 Proof of Lemma 2

**Proof** Consider the condition of eqn. (12):

$$\|\mathbf{V}^\mathsf{T}\mathbf{W}\mathbf{W}^\mathsf{T}\mathbf{V} - \mathbf{I}_m\|_2 \leq \frac{\zeta}{2} \iff -\frac{\zeta}{2}\mathbf{I}_m \preccurlyeq \mathbf{V}^\mathsf{T}\mathbf{W}\mathbf{W}^\mathsf{T}\mathbf{V} - \mathbf{I}_m \preccurlyeq \frac{\zeta}{2}\mathbf{I}_m \qquad (22)$$

$$\iff -\frac{\zeta}{2}\mathbf{A}\mathbf{D}^2\mathbf{A}^\mathsf{T} \preccurlyeq \mathbf{A}\mathbf{D}\mathbf{W}\mathbf{W}^\mathsf{T}\mathbf{D}\mathbf{A}^\mathsf{T} - \mathbf{A}\mathbf{D}^2\mathbf{A}^\mathsf{T} \preccurlyeq \frac{\zeta}{2}\mathbf{A}\mathbf{D}^2\mathbf{A}^\mathsf{T} \qquad (23)$$

$$\iff \left(1 - \frac{\zeta}{2}\right)\mathbf{A}\mathbf{D}^2\mathbf{A}^\mathsf{T} \preccurlyeq \underbrace{\mathbf{A}\mathbf{D}\mathbf{W}\mathbf{W}^\mathsf{T}\mathbf{D}\mathbf{A}^\mathsf{T}}_{\mathbf{Q}} \preccurlyeq \left(1 + \frac{\zeta}{2}\right)\mathbf{A}\mathbf{D}^2\mathbf{A}^\mathsf{T}, \qquad (24)$$

where we obtain eqn. (23) by pre- and post-multiplying the previous inequality by $\mathbf{U}\boldsymbol{\Sigma}$ and $\boldsymbol{\Sigma}\mathbf{U}^\mathsf{T}$ respectively and using the facts that $\mathbf{A}\mathbf{D} = \mathbf{U}\boldsymbol{\Sigma}\mathbf{V}^\mathsf{T}$ and $\mathbf{A}\mathbf{D}^2\mathbf{A}^\mathsf{T} = \mathbf{U}\boldsymbol{\Sigma}^2\mathbf{U}^\mathsf{T}$. Also, from eqn. (22), note that all the eigenvalues of $\mathbf{V}^\mathsf{T}\mathbf{W}\mathbf{W}^\mathsf{T}\mathbf{V}$ lie between $(1 - \frac{\zeta}{2})$ and $(1 + \frac{\zeta}{2})$ *i.e.*, $\mathrm{rank}(\mathbf{V}^\mathsf{T}\mathbf{W}) = m$. Therefore, $\mathrm{rank}(\mathbf{A}\mathbf{D}\mathbf{W}) = \mathrm{rank}(\mathbf{U}\boldsymbol{\Sigma}\mathbf{V}^\mathsf{T}\mathbf{W}) = m$, as $\mathbf{U}\boldsymbol{\Sigma}$ is non-singular and we know rank of a matrix remains unaltered by pre (or post)-multiplying by a non-singular matrix. So, we have $\mathrm{rank}(\mathbf{Q}) = m$; in words $\mathbf{Q}$ has full rank. Therefore, all the diagonal entries of $\boldsymbol{\Sigma}_\mathbf{Q}$ are positive and $\mathbf{Q}^{-1/2}\mathbf{Q}\mathbf{Q}^{-1/2} = (\mathbf{U}_\mathbf{Q}\boldsymbol{\Sigma}_\mathbf{Q}^{-1/2}\mathbf{U}_\mathbf{Q}^\mathsf{T})\,\mathbf{U}_\mathbf{Q}\boldsymbol{\Sigma}_\mathbf{Q}\mathbf{U}_\mathbf{Q}^\mathsf{T}\,(\mathbf{U}_\mathbf{Q}\boldsymbol{\Sigma}_\mathbf{Q}^{-1/2}\mathbf{U}_\mathbf{Q}^\mathsf{T}) = \mathbf{I}_m$.

Using above arguments, pre- and post- multiplying eqn. (24) by $\mathbf{Q}^{-1/2}$, we obtain

$$\left(1 - \frac{\zeta}{2}\right)\mathbf{Q}^{-1/2}\mathbf{A}\mathbf{D}^2\mathbf{A}^\mathsf{T}\mathbf{Q}^{-1/2} \preccurlyeq \mathbf{I}_m \preccurlyeq \left(1 + \frac{\zeta}{2}\right)\mathbf{Q}^{-1/2}\mathbf{A}\mathbf{D}^2\mathbf{A}^\mathsf{T}\mathbf{Q}^{-1/2}$$

$$\iff \left(1 + \frac{\zeta}{2}\right)^{-1}\mathbf{I}_m \preccurlyeq \mathbf{Q}^{-1/2}\mathbf{A}\mathbf{D}^2\mathbf{A}^\mathsf{T}\mathbf{Q}^{-1/2} \preccurlyeq \left(1 - \frac{\zeta}{2}\right)^{-1}\mathbf{I}_m. \qquad (25)$$

Eqn. (25) implies and is implied by the fact that all the eigenvalues of $\mathbf{Q}^{-1/2}\mathbf{A}\mathbf{D}^2\mathbf{A}^\mathsf{T}\mathbf{Q}^{-1/2}$ are bounded between $\left(1 + \frac{\zeta}{2}\right)^{-1}$ and $\left(1 - \frac{\zeta}{2}\right)^{-1}$. Therefore, we have

$$\left(1 + \frac{\zeta}{2}\right)^{-1} \leq \sigma_i^2(\mathbf{Q}^{-1/2}\mathbf{A}\mathbf{D}) \leq \left(1 - \frac{\zeta}{2}\right)^{-1}, \quad \textbf{for } i = 1, \ldots, m.$$

∎

### C.2 Satisfying eqn. (7) using CG Solver

Let $\tilde{\mathbf{f}}^{(j)}$ be the residual at the $j$-th iteration of the CG algorithm, i.e., $\tilde{\mathbf{f}}^{(j)} = \mathbf{Q}^{-1/2}\mathbf{A}\mathbf{D}^2\mathbf{A}^\mathsf{T}\mathbf{Q}^{-1/2}\tilde{\mathbf{z}}^j - \mathbf{Q}^{-1/2}\mathbf{p}$. Recall from Algorithm 1 that $\tilde{\mathbf{z}}^0 = \mathbf{0}$ and thus $\tilde{\mathbf{f}}^{(0)} = -\mathbf{Q}^{-1/2}\mathbf{p}$. In our parlance, Theorem 8 of [6] proved the following bound.

**Lemma 5 (Theorem 8 of [6])** *Let $\tilde{\mathbf{f}}^{(j-1)}$ and $\tilde{\mathbf{f}}^{(j)}$ be the residuals obtained by the CG solver at steps $j - 1$ and $j$. Then,*

$$\|\tilde{\mathbf{f}}^{(j)}\|_2 \leq \frac{\kappa^2(\mathbf{Q}^{-1/2}\mathbf{A}\mathbf{D}) - 1}{2}\|\tilde{\mathbf{f}}^{(j-1)}\|_2,$$

*where $\kappa(\mathbf{Q}^{-1/2}\mathbf{A}\mathbf{D})$ is the condition number of $\mathbf{Q}^{-1/2}\mathbf{A}\mathbf{D}$.*

From Lemma 2, we get

$$\kappa^2(\mathbf{Q}^{-1/2}\mathbf{A}\mathbf{D}) = \frac{\sigma_{\max}^2(\mathbf{Q}^{-1/2}\mathbf{A}\mathbf{D})}{\sigma_{\min}^2(\mathbf{Q}^{-1/2}\mathbf{A}\mathbf{D})} \leq \frac{1 + \zeta/2}{1 - \zeta/2}. \qquad (26)$$

Combining eqn. (26) with Lemma 5,

$$\|\tilde{\mathbf{f}}^{(j)}\|_2 \leq \frac{\frac{1+\zeta/2}{1-\zeta/2} - 1}{2}\|\tilde{\mathbf{f}}^{(j-1)}\|_2 = \frac{\zeta}{2 - \zeta}\|\tilde{\mathbf{f}}^{(j-1)}\|_2 \leq \zeta\|\tilde{\mathbf{f}}^{(j-1)}\|_2, \qquad (27)$$

where the last inequality follows from $\zeta \leq 1$. Applying eqn. (27) recursively, we get

$$\|\tilde{\mathbf{f}}^{(t)}\|_2 \leq \zeta \|\tilde{\mathbf{f}}^{(t-1)}\|_2 \leq \cdots \leq \zeta^t \|\tilde{\mathbf{f}}^{(0)}\|_2 = \zeta^t \|\mathbf{Q}^{-1/2}\mathbf{p}\|_2\,,$$

which proves the condition of eqn. (7).

## C.3  Proof of Lemma 3

**Proof**  Let $\mathbf{AD} = \mathbf{U\Sigma V}^\mathsf{T}$ be the thin SVD representation of $\mathbf{AD}$. We use the exact same $\mathbf{W}$ as discussed in Section 2. Therefore, eqn. (12) holds with probability $1 - \delta$ and it directly follows from the proof of Lemma 2 that $\mathrm{rank}(\mathbf{ADW}) = m$.

Now, as $\mathbf{ADW}$ has full *row-rank*, right-inverse exists and $\mathbf{ADW}\,(\mathbf{ADW})^\dagger = \mathbf{I}_m$. Therefore, taking $\mathbf{v} = (\mathbf{XS})^{1/2}\mathbf{W}(\mathbf{ADW})^\dagger(\mathbf{AD}^2\mathbf{A}^\mathsf{T}\hat{\Delta}\mathbf{y} - \mathbf{p})$, we finally have

$$\begin{aligned}
\mathbf{AS}^{-1}\,\mathbf{v} &= \mathbf{AS}^{-1}(\mathbf{XS})^{1/2}\mathbf{W}(\mathbf{ADW})^\dagger(\mathbf{AD}^2\mathbf{A}^\mathsf{T}\hat{\Delta}\mathbf{y} - \mathbf{p}) \\
&= \mathbf{ADW}(\mathbf{ADW})^\dagger(\mathbf{AD}^2\mathbf{A}^\mathsf{T}\hat{\Delta}\mathbf{y} - \mathbf{p}) \\
&= \mathbf{AD}^2\mathbf{A}^\mathsf{T}\hat{\Delta}\mathbf{y} - \mathbf{p}\,,
\end{aligned}$$

where the second equality follows from the fact that $\mathbf{D} = \mathbf{X}^{1/2}\mathbf{S}^{-1/2}$. This concludes the proof.  ∎

## C.4  Proof of Lemma 4

**Proof**  We already have, $\mathbf{Q} = \mathbf{ADW}(\mathbf{ADW})^\mathsf{T} = \mathbf{U_Q \Sigma_Q U_Q^\mathsf{T}}$. From this, we know that $\mathbf{U_Q}$ and $\mathbf{\Sigma_Q^{1/2}}$ are respectively the matrices of left singular vectors and singular values of $\mathbf{ADW}$. Now, let $\widehat{\mathbf{V}}$ be the right singular vector of $\mathbf{ADW}$. Therefore, $\mathbf{ADW} = \mathbf{U_Q \Sigma_Q^{1/2}}\widehat{\mathbf{V}}^\mathsf{T}$ is the thin SVD representation of $\mathbf{ADW}$. Also, from Lemma 2, we know $\mathbf{Q}$ has full rank. Therefore, $\mathbf{Q}^{1/2}\mathbf{Q}^{-1/2} = \mathbf{I}_m$.

Next, we bound $\|\mathbf{v}\|_2$ in the following way

$$\begin{aligned}
\|\mathbf{v}\|_2 &= \|(\mathbf{XS})^{1/2}\mathbf{W}(\mathbf{ADW})^\dagger(\mathbf{AD}^2\mathbf{A}^\mathsf{T}\hat{\Delta}\mathbf{y} - \mathbf{p})\|_2 \\
&= \|(\mathbf{XS})^{1/2}\mathbf{W}(\mathbf{ADW})^\dagger\mathbf{Q}^{1/2}\mathbf{Q}^{-1/2}(\mathbf{AD}^2\mathbf{A}^\mathsf{T}\hat{\Delta}\mathbf{y} - \mathbf{p})\|_2 \\
&\leq \|(\mathbf{XS})^{1/2}\mathbf{W}(\mathbf{ADW})^\dagger\mathbf{Q}^{1/2}\|_2\,\|\tilde{\mathbf{f}}^{(t)}\|_2\,,
\end{aligned} \tag{28}$$

where we have used the fact that $\mathbf{Q}^{-1/2}(\mathbf{AD}^2\mathbf{A}^\mathsf{T}\hat{\Delta}\mathbf{y} - \mathbf{p}) = \tilde{\mathbf{f}}^{(t)}$ and the last inequality follows from the sub-multiplicativity property of spectral-norm.

Again, using SVD of $\mathbf{ADW}$ and $\mathbf{Q}$, we have $(\mathbf{ADW})^\dagger\mathbf{Q}^{1/2} = \widehat{\mathbf{V}}\mathbf{\Sigma_Q^{-1/2}}\mathbf{U_Q^\mathsf{T}}\mathbf{U_Q \Sigma_Q^{1/2}}\mathbf{U_Q^\mathsf{T}} = \widehat{\mathbf{V}}\mathbf{U_Q^\mathsf{T}}$. Now, note that $\mathbf{U_Q} \in \mathbb{R}^{m \times m}$ is an orthogonal matrix and $\widehat{\mathbf{V}} \in \mathbb{R}^{w \times m}$ has orthogonal columns *i.e.* $\|\widehat{\mathbf{V}}\|_2 = 1$. Therefore, combining these with eqn. (28) yields,

$$\begin{aligned}
\|\mathbf{v}\|_2 &\leq \|(\mathbf{XS})^{1/2}\mathbf{W}\widehat{\mathbf{V}}\mathbf{U_Q^\mathsf{T}}\|_2\|\tilde{\mathbf{f}}^{(t)}\|_2 = \|(\mathbf{XS})^{1/2}\mathbf{W}\widehat{\mathbf{V}}\|_2\|\tilde{\mathbf{f}}^{(t)}\|_2 \\
&\leq \|(\mathbf{XS})^{1/2}\mathbf{W}\|_2\|\widehat{\mathbf{V}}\|_2\|\tilde{\mathbf{f}}^{(t)}\|_2 = \|(\mathbf{XS})^{1/2}\mathbf{W}\|_2\|\tilde{\mathbf{f}}^{(t)}\|_2\,,
\end{aligned} \tag{29}$$

where the first equality in eqn. (29) follows from the unitary invariance property of the spectral norm; the second inequality follows from the sub-multiplicativity of the spectral norm and the last equality is due to $\|\widehat{\mathbf{V}}\|_2 = 1$. Now, as we use the exact same $\mathbf{W}$ discussed in Section 2 to construct $\mathbf{v}$ and note that eqn. (10) holds for any matrix $\mathbf{Z}$ (irrespective of its dimensions). Therefore, taking $\mathbf{Z} = (\mathbf{XS})^{1/2}$ with that $\mathbf{W}$, eqn. (10) in Section 1.3 boils down to

$$\left\|(\mathbf{XS})^{1/2}\mathbf{WW}^\mathsf{T}(\mathbf{XS})^{1/2} - (\mathbf{XS})\right\|_2 \leq \frac{\zeta}{4}\left(\|(\mathbf{XS})^{1/2}\|_2^2 + \frac{\|(\mathbf{XS})^{1/2}\|_F^2}{m}\right) \tag{30}$$

holds with probability at least $1 - \delta$.

Now, applying Weyl's inequality on the left hand side of the eqn. (30), we further have

$$\left|\left\|(\mathbf{XS})^{1/2}\mathbf{W}\right\|_2^2 - \left\|(\mathbf{XS})^{1/2}\right\|_2^2\right| \leq \frac{\zeta}{4}\left(\|(\mathbf{XS})^{1/2}\|_2^2 + \frac{\|(\mathbf{XS})^{1/2}\|_F^2}{m}\right) \tag{31}$$

Now, using the facts that $\frac{\zeta}{4} \leq 1$, $\|(\mathbf{XS})^{1/2}\|_2 \leq \|(\mathbf{XS})^{1/2}\|_F$, and $\frac{\|(\mathbf{XS})^{1/2}\|_F^2}{m} \leq \|(\mathbf{XS})^{1/2}\|_F^2$, from eqn. (31),

$$\left\|(\mathbf{XS})^{1/2}\mathbf{W}\right\|_2^2 \leq 3\|(\mathbf{XS})^{1/2}\|_F^2 = 3n\mu\,, \tag{32}$$

where the last equality follows from $\|(\mathbf{XS})^{1/2}\|_F^2 = \mathbf{x}^\mathsf{T}\mathbf{s} = n\mu$.

Finally, combining eqns. (29) and (32), we conclude

$$\|\mathbf{v}\|_2 \leq \sqrt{3n\mu}\|\tilde{\mathbf{f}}^{(t)}\|_2.$$

$\blacksquare$

## Appendix D    Richardson Iteration

Here, we show that all our analyses still hold, even if we replace Step 3 of Algorithm 1 (CG solver) with Richardson iteration. Basically, all we need to show is the condition in eqn. (7) holds. Note that the condition in eqn. (6) already holds from Lemma 2, as we use the exact same sketching matrix $\mathbf{W} \in \mathbb{R}^{n \times w}$ discussed in Section 2.

---

**Algorithm 3** Richardson Iteration Solver

---

**Input:** $\mathbf{AD} \in \mathbb{R}^{m \times n}$, $\mathbf{p} \in \mathbb{R}^m$; number of iterations $t > 0$; sketching matrix $\mathbf{W} \in \mathbb{R}^{n \times w}$;
**Initialize:** $\tilde{\mathbf{z}}^0 \leftarrow \mathbf{0}_m$;
**for** $j = 1$ **to** $t$ **do**
$\quad \tilde{\mathbf{z}}^j \leftarrow \tilde{\mathbf{z}}^{j-1} + \mathbf{Q}^{-1/2}(\mathbf{p} - \mathbf{AD}^2\mathbf{A}^\mathsf{T}\mathbf{Q}^{-1/2}\tilde{\mathbf{z}}^{j-1})$;
**end for**
**Output:** return $\tilde{\mathbf{z}}^t$;

---

Our first result expresses the residual vector $\tilde{\mathbf{f}}^{(j)}$ in terms of $\tilde{\mathbf{f}}^{(j-1)}$ for $j = 1, 2, \ldots, t$.

**Lemma 6** *Let $\tilde{\mathbf{f}}^{(j)}$, $j = 1, 2, \ldots, t$ be the residual vectors at each iteration.Then,*

$$\tilde{\mathbf{f}}^{(j)} = \left(\mathbf{I}_m - \mathbf{Q}^{-1/2}\mathbf{AD}^2\mathbf{A}^\mathsf{T}\mathbf{Q}^{-1/2}\right)\tilde{\mathbf{f}}^{(j-1)}\,. \tag{33}$$

*Recall that $\mathbf{Q} = \mathbf{ADWW}^\mathsf{T}\mathbf{DA}^\mathsf{T}$ and $\tilde{\mathbf{f}}^{(j)} = \mathbf{Q}^{-1/2}(\mathbf{AD}^2\mathbf{A}^\mathsf{T}\mathbf{Q}^{-1/2}\tilde{\mathbf{z}}^j - \mathbf{p})$.*

**Proof** Using Algorithm 3, we express $\tilde{\mathbf{f}}^{(j)}$ as

$$
\begin{aligned}
\tilde{\mathbf{f}}^{(j)} &= \mathbf{Q}^{-1/2}\mathbf{AD}^2\mathbf{A}^\mathsf{T}\mathbf{Q}^{-1/2}\tilde{\mathbf{z}}^j - \mathbf{Q}^{-1/2}\mathbf{p} \\
&= \mathbf{Q}^{-1/2}\mathbf{AD}^2\mathbf{A}^\mathsf{T}\mathbf{Q}^{-1/2}\left(\tilde{\mathbf{z}}^{j-1} + \mathbf{Q}^{-1/2}(\mathbf{p} - \mathbf{AD}^2\mathbf{A}^\mathsf{T}\mathbf{Q}^{-1/2}\tilde{\mathbf{z}}^{j-1})\right) - \mathbf{Q}^{-1/2}\mathbf{p} \\
&= \left(\mathbf{Q}^{-1/2}\mathbf{AD}^2\mathbf{A}^\mathsf{T}\mathbf{Q}^{-1/2}\tilde{\mathbf{z}}^{j-1} - \mathbf{Q}^{-1/2}\mathbf{p}\right) \\
&\qquad - \mathbf{Q}^{-1/2}\mathbf{AD}^2\mathbf{A}^\mathsf{T}\mathbf{Q}^{-1/2}\left(\mathbf{Q}^{-1/2}\mathbf{AD}^2\mathbf{A}^\mathsf{T}\mathbf{Q}^{-1/2}\tilde{\mathbf{z}}^{j-1} - \mathbf{Q}^{-1/2}\mathbf{p}\right) \\
&= \left(\mathbf{I}_m - \mathbf{Q}^{-1/2}\mathbf{AD}^2\mathbf{A}^\mathsf{T}\mathbf{Q}^{-1/2}\right)\left(\mathbf{Q}^{-1/2}\mathbf{AD}^2\mathbf{A}^\mathsf{T}\mathbf{Q}^{-1/2}\tilde{\mathbf{z}}^{j-1} - \mathbf{Q}^{-1/2}\mathbf{p}\right) \\
&= \left(\mathbf{I}_m - \mathbf{Q}^{-1/2}\mathbf{AD}^2\mathbf{A}^\mathsf{T}\mathbf{Q}^{-1/2}\right)\tilde{\mathbf{f}}^{(j-1)}\,,
\end{aligned}
$$

which concludes the proof. $\blacksquare$

In the next result, we show that the spectral norm of $(\mathbf{I}_m - \mathbf{Q}^{-1/2}\mathbf{AD}^2\mathbf{A}^\mathsf{T}\mathbf{Q}^{-1/2})$ is upper bounded by $\zeta$.

**Lemma 7** *Let the condition in eqn. (6) holds for the sketching matrix $\mathbf{W} \in \mathbb{R}^{n \times w}$, then*

$$\|\mathbf{Q}^{-1/2}\mathbf{AD}^2\mathbf{A}^\mathsf{T}\mathbf{Q}^{-1/2} - \mathbf{I}_m\|_2 \leq \zeta\,.$$

**Proof** As the condition in eqn. (6) holds, we can go backwards in the proof of Lemma 2 and see that eqn. (25) holds. So, we subtract $\mathbf{I}_m$ from each side of eqn. (25) to get

$$\left(\frac{2}{2+\zeta}-1\right)\mathbf{I}_m \preccurlyeq \mathbf{Q}^{-1/2}\mathbf{AD}^2\mathbf{A}^\mathsf{T}\mathbf{Q}^{-1/2}-\mathbf{I}_m \preccurlyeq \left(\frac{2}{2-\zeta}-1\right)\mathbf{I}_m$$

$$\Leftrightarrow -\frac{\zeta}{2+\zeta}\mathbf{I}_m \preccurlyeq \mathbf{Q}^{-1/2}\mathbf{AD}^2\mathbf{A}^\mathsf{T}\mathbf{Q}^{-1/2}-\mathbf{I}_m \preccurlyeq \frac{\zeta}{2-\zeta}\mathbf{I}_m$$

$$\Rightarrow -\frac{\zeta}{2-\zeta}\mathbf{I}_m \preccurlyeq \mathbf{Q}^{-1/2}\mathbf{AD}^2\mathbf{A}^\mathsf{T}\mathbf{Q}^{-1/2}-\mathbf{I}_m \preccurlyeq \frac{\zeta}{2-\zeta}\mathbf{I}_m \tag{34}$$

$$\Leftrightarrow \|\mathbf{Q}^{-1/2}\mathbf{AD}^2\mathbf{A}^\mathsf{T}\mathbf{Q}^{-1/2}-\mathbf{I}_m\|_2 \le \frac{\zeta}{2-\zeta} \le \zeta, \tag{35}$$

where eqn. (34) holds as $\frac{\zeta}{2+\zeta} \le \frac{\zeta}{2-\zeta}$ and the last inequality of eqn. (35) follows from $\zeta < 1$. ∎

**Satisfying eqn. (6).** Note that the condition in eqn. (6) already holds from Lemma 2, as we use the exact same sketching matrix $\mathbf{W} \in \mathbb{R}^{n \times w}$ discussed in Section 2.

**Satisfying eqn. (7).** Using Lemma 7 and applying Lemma 6 recursively, we get

$$\|\tilde{\mathbf{f}}^{(t)}\|_2 \le \zeta\|\tilde{\mathbf{f}}^{(t-1)}\|_2 \le \cdots \le \zeta^t\|\tilde{\mathbf{f}}^{(0)}\|_2 = \zeta^t\|\mathbf{Q}^{-1/2}\mathbf{p}\|_2.$$

# Appendix E   Steepest Descent

We will now replace Step 3 of Algorithm 1 (our proposed CG solver) by preconditioned steepest descent. We will again prove that our analysis of the proposed infeasible long-step IPM remains essentially the same.

First, we construct the sketching matrix $\mathbf{W}$ as discussed in Section 1.3, with a slightly more stringent accuracy guarantee. More specifically, we necessitate that

$$\left\|\mathbf{V}^\mathsf{T}\mathbf{WW}^\mathsf{T}\mathbf{V} - \mathbf{I}_m\right\|_2 \le \frac{\zeta(1-\zeta)}{2} \tag{36}$$

holds with probability at least $1 - \delta$ for a constant $\zeta \in [0,1]$. Notice that the sketching dimension $w = \mathcal{O}(m\log(m/\delta))$ and the running time needed to compute $\mathbf{Q}^{-1/2}$ (which is $\mathcal{O}(\mathsf{nnz}(\mathbf{A})\cdot\log(m/\delta)+m^3\log(m/\delta)))$ remain, asymptotically, the same. In the case of steepest descent, it turns out that at each iteration the search direction is the negative of the gradient, which is equal to the residual $\tilde{\mathbf{f}}^{(j)}$. Moreover, the step size $\alpha_j$ is determined by an exact *line search* that minimizes the underlying quadratic function:

$$\alpha_j = \frac{\tilde{\mathbf{f}}^{(j)\mathsf{T}}\tilde{\mathbf{f}}^{(j)}}{\tilde{\mathbf{f}}^{(j)\mathsf{T}}\mathbf{Q}^{-1/2}\mathbf{AD}^2\mathbf{A}^\mathsf{T}\mathbf{Q}^{-1/2}\tilde{\mathbf{f}}^{(j)}}.$$

For this choice of $\alpha_j$, it is easy to verify that the current gradient is orthogonal to the previous one.

---

**Algorithm 4** Steepest Descent Solver

---

**Input:** $\mathbf{AD} \in \mathbb{R}^{m \times n}$, $\mathbf{p} \in \mathbb{R}^m$; number of iterations $t > 0$; sketching matrix $\mathbf{W} \in \mathbb{R}^{n \times w}$;
**Initialize:** $\tilde{\mathbf{z}}^0 \leftarrow \mathbf{0}_m$;
**for** $j = 0$ **to** $t - 1$ **do**

$\quad \alpha_j = \frac{\tilde{\mathbf{f}}^{(j)\mathsf{T}}\tilde{\mathbf{f}}^{(j)}}{\tilde{\mathbf{f}}^{(j)\mathsf{T}}\mathbf{Q}^{-1/2}\mathbf{AD}^2\mathbf{A}^\mathsf{T}\mathbf{Q}^{-1/2}\tilde{\mathbf{f}}^{(j)}}$;

$\quad \tilde{\mathbf{z}}^{j+1} \leftarrow \tilde{\mathbf{z}}^j - \alpha_j\tilde{\mathbf{f}}^{(j)}$;
**end for**

**Output:** return = $\tilde{\mathbf{z}}^t$;

---

Similar to Lemma 6, our next result reveals a recursive relation between the search directions which, later on, will be instrumental in bounding $\tilde{\mathbf{f}}^{(t)}$.

**Lemma 8** *Let $\tilde{\mathbf{f}}^{(j)}$, $j = 1, 2, \ldots, t$ be the residual vectors at each iteration and $\alpha_j$ is given by Algorithm 4. Then,*

$$\tilde{\mathbf{f}}^{(j+1)} = \left(\mathbf{I}_m - \alpha_j \mathbf{Q}^{-1/2} \mathbf{A} \mathbf{D}^2 \mathbf{A}^\mathsf{T} \mathbf{Q}^{-1/2}\right) \tilde{\mathbf{f}}^{(j)}, \tag{37}$$

*Recall that $\mathbf{Q} = \mathbf{A} \mathbf{D} \mathbf{W} \mathbf{W}^\mathsf{T} \mathbf{D} \mathbf{A}^\mathsf{T}$ and $\tilde{\mathbf{f}}^{(j)} = \mathbf{Q}^{-1/2}(\mathbf{A} \mathbf{D}^2 \mathbf{A}^\mathsf{T} \mathbf{Q}^{-1/2} \tilde{\mathbf{z}}^j - \mathbf{p})$.*

**Proof** In Algorithm 4, we pre-multiply $\tilde{\mathbf{z}}^{j+1}$ with $\mathbf{Q}^{-1/2} \mathbf{A} \mathbf{D}^2 \mathbf{A}^\mathsf{T} \mathbf{Q}^{-1/2}$ and then subtract $\mathbf{Q}^{-1/2} \mathbf{p}$ to get

$$\begin{aligned}
\tilde{\mathbf{f}}^{(j+1)} &= \mathbf{Q}^{-1/2} \mathbf{A} \mathbf{D}^2 \mathbf{A}^\mathsf{T} \mathbf{Q}^{-1/2} \tilde{\mathbf{z}}^{j+1} - \mathbf{Q}^{-1/2} \mathbf{p} \\
&= \mathbf{Q}^{-1/2} \mathbf{A} \mathbf{D}^2 \mathbf{A}^\mathsf{T} \mathbf{Q}^{-1/2} \tilde{\mathbf{z}}^j - \mathbf{Q}^{-1/2} \mathbf{p} - \alpha_j \mathbf{Q}^{-1/2} \mathbf{A} \mathbf{D}^2 \mathbf{A}^\mathsf{T} \mathbf{Q}^{-1/2} \tilde{\mathbf{f}}^{(j)} \\
&= \tilde{\mathbf{f}}^{(j)} - \alpha_j \mathbf{Q}^{-1/2} \mathbf{A} \mathbf{D}^2 \mathbf{A}^\mathsf{T} \mathbf{Q}^{-1/2} \tilde{\mathbf{f}}^{(j)} = \left(\mathbf{I}_m - \alpha_j \mathbf{Q}^{-1/2} \mathbf{A} \mathbf{D}^2 \mathbf{A}^\mathsf{T} \mathbf{Q}^{-1/2}\right) \tilde{\mathbf{f}}^{(j)},
\end{aligned}$$

which concludes the proof. ∎

Next, using this new condition in eqn. (36), we will bound $\left\|\mathbf{I}_m - \alpha_j \mathbf{Q}^{-1/2} \mathbf{A} \mathbf{D}^2 \mathbf{A}^\mathsf{T} \mathbf{Q}^{-1/2}\right\|_2$ through a couple of results.

**Lemma 9** *If eqn. (36) is satisfied, then $|\alpha_j - 1| \le \frac{\zeta(1-\zeta)}{2}$.*

**Proof** First, we rewrite eqn. (36) as follows,

$$-\frac{\zeta(1-\zeta)}{2} \mathbf{I}_m \preccurlyeq \mathbf{V}^\mathsf{T} \mathbf{W} \mathbf{W}^\mathsf{T} \mathbf{V} - \mathbf{I}_m \preccurlyeq \frac{\zeta(1-\zeta)}{2} \mathbf{I}_m$$

Next, we pre and post-multiply the the above expression by $\mathbf{U}\mathbf{\Sigma}$ and $\mathbf{\Sigma}\mathbf{U}^\mathsf{T}$ to get

$$-\frac{\zeta(1-\zeta)}{2} \mathbf{A} \mathbf{D}^2 \mathbf{A}^\mathsf{T} \preccurlyeq \underbrace{\mathbf{A} \mathbf{D} \mathbf{W} \mathbf{W}^\mathsf{T} \mathbf{D} \mathbf{A}^\mathsf{T}}_{\mathbf{Q}} - \mathbf{A} \mathbf{D}^2 \mathbf{A}^\mathsf{T} \preccurlyeq \frac{\zeta(1-\zeta)}{2} \mathbf{A} \mathbf{D}^2 \mathbf{A}^\mathsf{T} \tag{38}$$

Now, pre and post-multiplying eqn. (38) again by $\mathbf{Q}^{-1/2}$, we have

$$\left(1 - \frac{\zeta(1-\zeta)}{2}\right) \mathbf{Q}^{-1/2} \mathbf{A} \mathbf{D}^2 \mathbf{A}^\mathsf{T} \mathbf{Q}^{-1/2} \preccurlyeq \mathbf{I}_m \preccurlyeq \left(1 + \frac{\zeta(1-\zeta)}{2}\right) \mathbf{Q}^{-1/2} \mathbf{A} \mathbf{D}^2 \mathbf{A}^\mathsf{T} \mathbf{Q}^{-1/2}$$

$$\Rightarrow \left(1 - \frac{\zeta(1-\zeta)}{2}\right) \tilde{\mathbf{f}}^{(j)\mathsf{T}} \mathbf{Q}^{-1/2} \mathbf{A} \mathbf{D}^2 \mathbf{A}^\mathsf{T} \mathbf{Q}^{-1/2} \tilde{\mathbf{f}}^{(j)} \le \tilde{\mathbf{f}}^{(j)\mathsf{T}} \tilde{\mathbf{f}}^{(j)} \le \left(1 + \frac{\zeta(1-\zeta)}{2}\right) \tilde{\mathbf{f}}^{(j)\mathsf{T}} \mathbf{Q}^{-1/2} \mathbf{A} \mathbf{D}^2 \mathbf{A}^\mathsf{T} \mathbf{Q}^{-1/2} \tilde{\mathbf{f}}^{(j)}$$

$$\Rightarrow \left(1 - \frac{\zeta(1-\zeta)}{2}\right) \le \frac{\tilde{\mathbf{f}}^{(j)\mathsf{T}} \tilde{\mathbf{f}}^{(j)}}{\tilde{\mathbf{f}}^{(j)\mathsf{T}} \mathbf{Q}^{-1/2} \mathbf{A} \mathbf{D}^2 \mathbf{A}^\mathsf{T} \mathbf{Q}^{-1/2} \tilde{\mathbf{f}}^{(j)}} \le \left(1 + \frac{\zeta(1-\zeta)}{2}\right)$$

$$\Leftrightarrow |\alpha_j - 1| \le \frac{\zeta(1-\zeta)}{2}, \text{ for } j = 1, 2, \ldots, t. \tag{39}$$

∎

Our next result shows that under eqn. (36), $\left\|\mathbf{I}_m - \alpha_j \mathbf{Q}^{-1/2} \mathbf{A} \mathbf{D}^2 \mathbf{A}^\mathsf{T} \mathbf{Q}^{-1/2}\right\|_2$ is upper bounded by a small quantity for for $j = 1, 2, \ldots, t$.

**Lemma 10** *If eqn. (36) is satisfied, then $\left\|\mathbf{I}_m - \alpha_j \mathbf{Q}^{-1/2} \mathbf{A} \mathbf{D}^2 \mathbf{A}^\mathsf{T} \mathbf{Q}^{-1/2}\right\|_2 \le \zeta$, for $j = 1, 2, \ldots, t$.*

**Proof** We note that eqn. (36) directly implies

$$\left\|\mathbf{V}^\mathsf{T} \mathbf{W} \mathbf{W}^\mathsf{T} \mathbf{V} - \mathbf{I}_m\right\|_2 \le \frac{\zeta}{2} \tag{40}$$

Now, as eqn. (40) holds, from eqn. (25) in the proof of Lemma 2, we have

$$\left(1 + \frac{\zeta}{2}\right)^{-1} \mathbf{I}_m \preccurlyeq \mathbf{Q}^{-1/2} \mathbf{A} \mathbf{D}^2 \mathbf{A}^\mathsf{T} \mathbf{Q}^{-1/2} \preccurlyeq \left(1 - \frac{\zeta}{2}\right)^{-1} \mathbf{I}_m$$

$$\Leftrightarrow \left(\frac{2\alpha_j}{2+\zeta} - 1\right)\mathbf{I}_m \preccurlyeq \alpha_j \mathbf{Q}^{-1/2}\mathbf{A}\mathbf{D}^2\mathbf{A}^\mathsf{T}\mathbf{Q}^{-1/2} - \mathbf{I}_m \preccurlyeq \left(\frac{2\alpha_j}{2-\zeta} - 1\right)\mathbf{I}_m$$

$$\Leftrightarrow \frac{2(\alpha_j - 1) - \zeta}{2 + \zeta}\mathbf{I}_m \preccurlyeq \alpha_j \mathbf{Q}^{-1/2}\mathbf{A}\mathbf{D}^2\mathbf{A}^\mathsf{T}\mathbf{Q}^{-1/2} - \mathbf{I}_m \preccurlyeq \frac{2(\alpha_j - 1) + \zeta}{2 - \zeta}\mathbf{I}_m, \qquad (41)$$

where the above expression follows from multiplying eqn. (25) by $\alpha_j$ and then subtracting $\mathbf{I}_m$.

Now, from Lemma 9, we have, $-\zeta(1 - \zeta) \le 2(\alpha_j - 1) \le \zeta(1 - \zeta)$ for $j = 1, 2, \ldots, t$. Using this in eqn. (41), we further have

$$-\frac{\zeta(1-\zeta)+\zeta}{2+\zeta}\mathbf{I}_m \preccurlyeq \alpha_j\mathbf{Q}^{-1/2}\mathbf{A}\mathbf{D}^2\mathbf{A}^\mathsf{T}\mathbf{Q}^{-1/2} - \mathbf{I}_m \preccurlyeq \frac{\zeta(1-\zeta)+\zeta}{2-\zeta}\mathbf{I}_m$$

$$\Leftrightarrow -\frac{\zeta(2-\zeta)}{2+\zeta}\mathbf{I}_m \preccurlyeq \alpha_j\mathbf{Q}^{-1/2}\mathbf{A}\mathbf{D}^2\mathbf{A}^\mathsf{T}\mathbf{Q}^{-1/2} - \mathbf{I}_m \preccurlyeq \zeta\mathbf{I}_m$$

$$\Rightarrow -\zeta\,\mathbf{I}_m \preccurlyeq \alpha_j\mathbf{Q}^{-1/2}\mathbf{A}\mathbf{D}^2\mathbf{A}^\mathsf{T}\mathbf{Q}^{-1/2} - \mathbf{I}_m \preccurlyeq \zeta\,\mathbf{I}_m \qquad (42)$$

$$\Rightarrow \left\|\mathbf{I}_m - \alpha_j\mathbf{Q}^{-1/2}\mathbf{A}\mathbf{D}^2\mathbf{A}^\mathsf{T}\mathbf{Q}^{-1/2}\right\|_2 \le \zeta,$$

where eqn. (42) is due to the fact that $\frac{2-\zeta}{2+\zeta} \le 1$. ∎

**Satisfying eqn. (6).** As eqn. (40) holds, eqn. (6) directly follows from Lemma 2.

**Satisfying eqn. (7).** Using Lemma 10 and applying Lemma 8 recursively, we get

$$\|\tilde{\mathbf{f}}^{(t)}\|_2 \le \zeta\|\tilde{\mathbf{f}}^{(t-1)}\|_2 \le \cdots \le \zeta^t\|\tilde{\mathbf{f}}^{(0)}\|_2 = \zeta^t\|\mathbf{Q}^{-1/2}\mathbf{p}\|_2\,.$$

# Appendix F    Convergence Analysis of Algorithm 2

## F.1    Number of Iterations for the CG Solver

In this section, most of the proofs follow [38] except for the fact that we used our sketching based preconditioner $\mathbf{Q}^{-1/2}$. Recall that $\mathcal{S}$ is the set of optimal and feasible solutions for the proposed LP.

**Lemma 11** *Let $(\mathbf{x}^0, \mathbf{y}^0, \mathbf{s}^0)$ be the initial point with $(\mathbf{x}^0, \mathbf{s}^0) > \mathbf{0}$ and $(\mathbf{x}^*, \mathbf{y}^*, \mathbf{s}^*) \in \mathcal{S}$ such that $(\mathbf{x}^*, \mathbf{s}^*) \le (\mathbf{x}^0, \mathbf{s}^0)$ with $\mathbf{s}^0 \ge |\mathbf{A}^\mathsf{T}\mathbf{y}^0 - \mathbf{c}|$. Then, for any point $(\mathbf{x}, \mathbf{y}, \mathbf{s}) \in \mathcal{N}(\gamma)$ such that $\mathbf{r} = \eta\,\mathbf{r}^0$ and $0 \le \eta \le \min\left\{1, \frac{\mathbf{s}^\mathsf{T}\mathbf{x}}{\mathbf{s}^{0\mathsf{T}}\mathbf{x}^0}\right\}$, then we have*

$$(i)\ \eta\,(\mathbf{x}^\mathsf{T}\mathbf{s}^0 + \mathbf{s}^\mathsf{T}\mathbf{x}^0) \le 3n\mu\,, \tag{43a}$$

$$(ii)\ \eta\,\|\mathbf{S}(\mathbf{x}^* - \mathbf{x}^0)\|_2 \le \eta\,\|\mathbf{S}\mathbf{x}^0\|_2 \le \eta\mathbf{s}^\mathsf{T}\mathbf{x}^0 \le 3n\mu\,, \tag{43b}$$

$$(iii)\ \eta\,\|\mathbf{X}(\mathbf{s}^0 + \mathbf{A}^\mathsf{T}\mathbf{y}^0 - \mathbf{c})\|_2 \le 2\eta\,\|\mathbf{X}\mathbf{s}^0\|_2 \le 2\eta\,\mathbf{x}^\mathsf{T}\mathbf{s}^0 \le 6n\mu\,. \tag{43c}$$

**Proof** We prove eqns. (43a)–(43c) below.

**Proof of eqn. (43a).** For completeness, we provide a proof of eqn. (43a) which is already discussed in [38]. Since $(\mathbf{x}^*, \mathbf{s}^*, \mathbf{y}^*) \in \mathcal{S}$, the following equalities hold:

$$\mathbf{A}\mathbf{x}^* = \mathbf{b} \tag{44a}$$

$$\mathbf{A}^\mathsf{T}\mathbf{y}^* + \mathbf{s}^* = \mathbf{c} \tag{44b}$$

Furthermore, $\mathbf{r} = \eta\mathbf{r}^0$ implies

$$\mathbf{A}\mathbf{x} - \mathbf{b} = \eta(\mathbf{A}\mathbf{x}^0 - b) \tag{45a}$$

$$\mathbf{A}^\mathsf{T}\mathbf{y} + \mathbf{s} - \mathbf{c} = \eta(\mathbf{A}^\mathsf{T}\mathbf{y}^0 + \mathbf{s}^0 - \mathbf{c}) \tag{45b}$$

Combining eqn. (44a) with eqn. (45a) and eqn. (44b) with eqn. (45b), we get

$$\mathbf{A}\left(\mathbf{x} - \eta\mathbf{x}^0 - (1 - \eta)\mathbf{x}^*\right) = \mathbf{0} \tag{46a}$$

$$\mathbf{A}^{\mathsf{T}}(\mathbf{y} - \eta\mathbf{y}^0 - (1-\eta)\mathbf{y}^*) + (\mathbf{s} - \eta\mathbf{s}^0 - (1-\eta)\mathbf{s}^*) = \mathbf{0} \tag{46b}$$

Multiplying the eqn. (46b) by $\left(\mathbf{x} - \eta\mathbf{x}^0 - (1-\eta)\mathbf{x}^*\right)^{\mathsf{T}}$ on the left and using eqn. (46a), we get

$$\left(\mathbf{x} - \eta\mathbf{x}^0 - (1-\eta)\mathbf{x}^*\right)^{\mathsf{T}}\left(\mathbf{s} - \eta\mathbf{s}^0 - (1-\eta)\mathbf{s}^*\right) = 0\,,$$

expanding which we get

$$\eta\left(\mathbf{x}^{0\mathsf{T}}\mathbf{s} + \mathbf{x}^{\mathsf{T}}\mathbf{s}^0\right) = \eta^2\mathbf{x}^{0\mathsf{T}}\mathbf{s}^0 + (1-\eta)^2(\mathbf{x}^*)^{\mathsf{T}}\mathbf{s}^* + \mathbf{x}^{\mathsf{T}}\mathbf{s}$$
$$+ \eta(1-\eta)\left(\mathbf{x}^{0\mathsf{T}}\mathbf{s}^* + (\mathbf{x}^*)^{\mathsf{T}}\mathbf{s}^0\right) - (1-\eta)\left((\mathbf{x}^*)^{\mathsf{T}}\mathbf{s} + \mathbf{x}^{\mathsf{T}}\mathbf{s}^*\right) \tag{47}$$

Next, we use the given conditions and rewrite eqn. (47) as

$$\eta\left(\mathbf{x}^{0\mathsf{T}}\mathbf{s} + \mathbf{s}^{0\mathsf{T}}\mathbf{x}\right) \leq \eta^2\mathbf{x}^{0\mathsf{T}}\mathbf{s}^0 + \mathbf{x}^{\mathsf{T}}\mathbf{s} + \eta(1-\eta)\left(\mathbf{x}^{0\mathsf{T}}\mathbf{s}^* + \mathbf{s}^{0\mathsf{T}}\mathbf{x}^*\right)$$
$$\leq \eta^2\mathbf{x}^{0\mathsf{T}}\mathbf{s}^0 + \mathbf{x}^{\mathsf{T}}\mathbf{s} + 2\eta(1-\eta)\mathbf{x}^{0\mathsf{T}}\mathbf{s}^0$$
$$\leq 2\eta\mathbf{x}^{0\mathsf{T}}\mathbf{s}^0 + \mathbf{x}^{\mathsf{T}}\mathbf{s} \leq 3\mathbf{x}^{\mathsf{T}}\mathbf{s} = 3n\mu\,, \tag{48}$$

where the first inequality in eqn. (48) follows from from a couple of facts. First, $(1-\eta)((\mathbf{x}^*)^{\mathsf{T}}\mathbf{s} + \mathbf{x}^{\mathsf{T}}\mathbf{s}^*) \geq 0$ as $(\mathbf{x}^*, \mathbf{s}^*) \geq \mathbf{0}$ and $(\mathbf{x}^0, \mathbf{s}^0) \geq \mathbf{0}$; second, as $(\mathbf{x}^*, \mathbf{s}^*, \mathbf{y}^*) \in \mathcal{S}$ (which implies $\mathbf{x}^* \circ \mathbf{s}^* = \mathbf{0}$), we have $(\mathbf{x}^*)^{\mathsf{T}}\mathbf{s}^* = 0$. Second inequality in eqn. (48) holds as $\mathbf{x}^* \leq \mathbf{x}^0$, $\mathbf{s}^* \leq \mathbf{s}^0$, $(\mathbf{x}^*, \mathbf{s}^*) \geq \mathbf{0}$ and $(\mathbf{x}^0, \mathbf{s}^0) \geq \mathbf{0}$; combining which we have $(\mathbf{x}^{0\mathsf{T}}\mathbf{s}^* + \mathbf{s}^{0\mathsf{T}}\mathbf{x}^*) \leq 2\mathbf{x}^{0\mathsf{T}}\mathbf{s}^0$. Third inequality in eqn. (48) is true as we have $\eta^2\mathbf{x}^{0\mathsf{T}} + 2\eta(1-\eta)\mathbf{x}^{0\mathsf{T}}\mathbf{s}^0 = 2\eta\mathbf{x}^{0\mathsf{T}}\mathbf{s}^0 - \eta^2\mathbf{x}^{0\mathsf{T}}\mathbf{s}^0 \leq 2\eta\mathbf{x}^{0\mathsf{T}}\mathbf{s}^0$. Final inequality holds as $\eta \leq \frac{\mathbf{x}^{\mathsf{T}}\mathbf{s}}{\mathbf{x}^{0\mathsf{T}}\mathbf{s}^0}$ .

**Proof of eqn. (43b).** The last inequality directly follows from eqn. (43a); second last inequality is also easy to prove as

$$\|\mathbf{S}\mathbf{x}^0\|_2 = \sqrt{\sum_{i=1}^{s}(s_i x_i^0)^2} \leq \sqrt{\left(\sum_{i=1}^{s} s_i x_i^0\right)^2} = \mathbf{s}^{\mathsf{T}}\mathbf{x}^0\,. \tag{49}$$

To prove the first inequality in eqn. (43b), we use the fact $\mathbf{x}^0 \geq \mathbf{x}^*$ as follows

$$\|\mathbf{S}\mathbf{x}^0\|_2^2 - \|\mathbf{S}(\mathbf{x}^* - \mathbf{x}^0)\|_2^2 = \sum_{i=1}^{n}(s_i x_i^0)^2 - \sum_{i=1}^{n} s_i^2\left((x_i^*)^2 + (x_i^0)^2 - 2x_i^* x_i^0\right)$$
$$= \sum_{i=1}^{n} s_i^2\left(2x_i^* x_i^0 - (x_i^*)^2\right) \geq 0\,.$$

**Proof of eqn. (43c).** This can be proven using a similar approach as in eqn. (43b). Last inequality directly follows from eqn. (43a); second last inequality is also easy to prove as

$$\|\mathbf{X}\mathbf{s}^0\|_2 = \sqrt{\sum_{i=1}^{n}(x_i s_i^0)^2} \leq \sqrt{\left(\sum_{i=1}^{n} x_i s_i^0\right)^2} = \mathbf{x}^{\mathsf{T}}\mathbf{s}^0\,. \tag{50}$$

For the first inequality, we proceed as follows

$$\|\mathbf{X}(\mathbf{s}^0 + \mathbf{A}^{\mathsf{T}}\mathbf{y}^0 - \mathbf{c})\|_2^2 = \|\mathbf{X}\mathbf{s}^0\|_2^2 + \|\mathbf{X}(\mathbf{A}^{\mathsf{T}}\mathbf{y}^0 - \mathbf{c})\|_2^2 + 2\mathbf{s}^{0\mathsf{T}}\mathbf{X}^{\mathsf{T}}\mathbf{X}(\mathbf{A}^{\mathsf{T}}\mathbf{y}^0 - \mathbf{c})$$
$$= \|\mathbf{X}\mathbf{s}^0\|_2^2 + \sum_{i=1}^{n} x_i^2(\mathbf{A}^{\mathsf{T}}\mathbf{y}^0 - \mathbf{c})_i^2 + 2\sum_{i=1}^{n} x_i^2 s_i^0(\mathbf{A}^{\mathsf{T}}\mathbf{y}^0 - \mathbf{c})_i$$
$$\leq \|\mathbf{X}\mathbf{s}^0\|_2^2 + \sum_{i=1}^{n}(x_i s_i^0)^2 + 2\sum_{i=1}^{n}(x_i s_i^0)^2$$

$$= \|\mathbf{X}\mathbf{s}^0\|_2^2 + \|\mathbf{X}\mathbf{s}^0\|_2^2 + 2\|\mathbf{X}\mathbf{s}^0\|_2^2 = 4\|\mathbf{X}\mathbf{s}^0\|_2^2 , \tag{51}$$

where the inequality in eqn. (51) follows from $x_i \geq 0$, $s_i^0 \geq 0$ and $\left|(\mathbf{A}^\mathsf{T}\mathbf{y}^0 - \mathbf{c})_i\right| \leq s_i^0$ for all $i = 1, 2, \ldots n$. This concludes the proof of Lemma 11. ∎

Our next result bounds $\|\mathbf{Q}^{-1/2}\mathbf{p}\|_2$ which will be instrumental in proving the final convergence bound.

**Lemma 12** *Let $(\mathbf{x}^0, \mathbf{y}^0, \mathbf{s}^0)$ be the initial point with $(\mathbf{x}^0, \mathbf{s}^0) > \mathbf{0}$ such that $\mathbf{x}^0 \geq \mathbf{x}^*$ and $\mathbf{s}^0 \geq \max\{\mathbf{s}^*, |\mathbf{c} - \mathbf{A}^\mathsf{T}\mathbf{y}^0|\}$ for some $(\mathbf{x}^*, \mathbf{y}^*, \mathbf{s}^*) \in \mathcal{S}$. Furthermore, let $(\mathbf{x}, \mathbf{y}, \mathbf{s}) \in \mathcal{N}(\gamma)$ with $\mathbf{r} = \eta\, \mathbf{r}^0$ for some $0 \leq \eta \leq 1$. If the sketching matrix $\mathbf{W} \in \mathbb{R}^{n \times w}$ satisfies the condition in eqn. (6), then*

$$\|\mathbf{Q}^{-1/2}\mathbf{p}\|_2 \leq \sqrt{2}\left(\frac{9n}{\sqrt{1-\gamma}} + \sigma\sqrt{\frac{n}{1-\gamma}} + \sqrt{n}\right)\sqrt{\mu}\,.$$

*Recall that,* $\mathbf{r} = (\mathbf{r}_p, \mathbf{r}_d) = (\mathbf{A}\mathbf{x} - \mathbf{b}, \mathbf{A}^\mathsf{T}\mathbf{y} + \mathbf{s} - \mathbf{c})$ *and* $\mathbf{r}^0 = (\mathbf{r}_p^0, \mathbf{r}_d^0) = (\mathbf{A}\mathbf{x}^0 - \mathbf{b}, \mathbf{A}^\mathsf{T}\mathbf{y}^0 + \mathbf{s}^0 - \mathbf{c})$.

**Proof** Note that after correcting the approximation error of the CG solver using $\mathbf{v}$, the primal and dual residuals $\mathbf{r} = (\mathbf{r}_p, \mathbf{r}_d)$ corresponding to an iterate $(\mathbf{x}, \mathbf{y}, \mathbf{s}) \in \mathcal{N}(\gamma)$ always lie on the line segment between zero and $\mathbf{r}^{(0)}$. In other words, $\mathbf{r} = \eta\mathbf{r}^{(0)}$ always holds for some $\eta \in [0, 1]$. This was formally proven in Lemma 3.3 of [38]. To bound $\|\mathbf{Q}^{-1/2}\mathbf{p}\|_2$, first we express $\mathbf{p}$ as in eqn. (3) and rewrite

$$\mathbf{Q}^{-1/2}\mathbf{p} = \mathbf{Q}^{-1/2}\left(-\mathbf{r}_p - \sigma\mu\mathbf{A}\mathbf{S}^{-1}\mathbf{1}_n + \mathbf{A}\mathbf{x} - \mathbf{A}\mathbf{D}^2\mathbf{r}_d\right) \tag{52}$$

Then, applying triangle inequality on $\|\mathbf{Q}^{-1/2}\mathbf{p}\|_2$ in eqn. (52), we get

$$\|\mathbf{Q}^{-1/2}\mathbf{p}\|_2 \leq \Delta_1 + \Delta_2 + \Delta_3 + \Delta_4\,, \tag{53}$$

where

$$\begin{aligned}
\Delta_1 &= \|\mathbf{Q}^{-1/2}\mathbf{r}_p\|_2\,, \\
\Delta_2 &= \sigma\mu\|\mathbf{Q}^{-1/2}\mathbf{A}\mathbf{D}(\mathbf{X}\mathbf{S})^{-1/2}\mathbf{1}_n\|_2\,, \\
\Delta_3 &= \|\mathbf{Q}^{-1/2}\mathbf{A}\mathbf{D}\mathbf{D}^{-1}\mathbf{x}\|_2\,, \\
\Delta_4 &= \|\mathbf{Q}^{-1/2}\mathbf{A}\mathbf{D}^2\mathbf{r}_d\|_2\,.
\end{aligned}$$

To bound $\Delta_1$, $\Delta_2$, $\Delta_3$ and $\Delta_4$ separately, we will heavily use the condition in eqn. (6). In particular, from eqn. (6), note that we have $\|\mathbf{Q}^{-1/2}\mathbf{A}\mathbf{D}\|_2 \leq \sqrt{2}$ as $\zeta \leq 1$.

**Bounding $\Delta_1$.** Putting $\mathbf{r}_p = \eta\,\mathbf{r}_p^0$, $\mathbf{r}_p^0 = \mathbf{A}\mathbf{x}^0 - \mathbf{b}$ and $\mathbf{b} = \mathbf{A}\mathbf{x}^*$, we rewrite $\Delta_1$ as

$$\begin{aligned}
\Delta_1 &= \eta\,\|\mathbf{Q}^{-1/2}\mathbf{A}(\mathbf{x}^0 - \mathbf{x}^*)\|_2 \\
&= \eta\,\|\mathbf{Q}^{-1/2}\mathbf{A}\mathbf{D}\mathbf{D}^{-1}(\mathbf{x}^0 - \mathbf{x}^*)\|_2 \\
&\leq \eta\,\|\mathbf{Q}^{-1/2}\mathbf{A}\mathbf{D}\|_2\|\mathbf{D}^{-1}(\mathbf{x}^0 - \mathbf{x}^*)\|_2 \\
&\leq \sqrt{2}\eta\,\|\mathbf{D}^{-1}(\mathbf{x}^0 - \mathbf{x}^*)\|_2 \\
&= \sqrt{2}\eta\,\|(\mathbf{X}\mathbf{S})^{-1/2}\mathbf{S}(\mathbf{x}^0 - \mathbf{x}^*)\|_2 \\
&\leq \sqrt{2}\eta\,\|(\mathbf{X}\mathbf{S})^{-1/2}\|_2\,\|\mathbf{S}(\mathbf{x}^0 - \mathbf{x}^*)\|_2\,, \tag{54}
\end{aligned}$$

where the above steps follow from submultiplicativity and eqn. (6). From eqn. (6), note that we have $\|\mathbf{Q}^{-1/2}\mathbf{A}\mathbf{D}\|_2 \leq \sqrt{2}$ as $\zeta \leq 1$. Now, applying eqn. (43b) and $\|(\mathbf{X}\mathbf{S})^{-1/2}\|_2 = \max_{1\leq i\leq n}\frac{1}{\sqrt{x_i s_i}}$, we further have

$$\begin{aligned}
\Delta_1 &\leq \sqrt{2}\max_{1\leq i\leq n}\frac{1}{\sqrt{x_i s_i}}\cdot 3n\mu \\
&\leq 3\sqrt{2}\,n\sqrt{\frac{\mu}{1-\gamma}}\,, \tag{55}
\end{aligned}$$

where the last inequality follows from $(\mathbf{x}, \mathbf{y}, \mathbf{s}) \in \mathcal{N}(\gamma)$.

**Bounding $\Delta_2$.** Applying submultiplicativity, we have

$$
\begin{aligned}
\Delta_2 &= \sigma\mu \, \|\mathbf{Q}^{-1/2}\,\mathbf{A}\mathbf{D}\,(\mathbf{X}\mathbf{S})^{-1/2}\mathbf{1}_n\|_2 \\
&\leq \sigma\mu \, \|\mathbf{Q}^{-1/2}\,\mathbf{A}\mathbf{D}\|_2 \|(\mathbf{X}\mathbf{S})^{-1/2}\mathbf{1}_n\|_2 \\
&\leq \sqrt{2}\,\sigma\mu \, \|(\mathbf{X}\mathbf{S})^{-1/2}\mathbf{1}_n\|_2 \\
&= \sqrt{2}\,\sigma\mu \, \sqrt{\sum_{i=1}^{n}\frac{1}{x_i s_i}} \;\leq\; \sqrt{2}\,\sigma\mu \, \sqrt{\sum_{i=1}^{n}\frac{1}{(1-\gamma)\mu}} \\
&= \sqrt{2}\,\sigma \, \sqrt{\frac{n\,\mu}{(1-\gamma)}}\,,
\end{aligned}
\tag{56}
$$

where the second last inequality follows from eqn. (6) and last inequality holds as $(\mathbf{x},\mathbf{y},\mathbf{s}) \in \mathcal{N}(\gamma)$.

**Bounding $\Delta_3$.** Putting $\mathbf{D} = \mathbf{S}^{-1/2}\mathbf{X}^{1/2}$; $\mathbf{x} = \mathbf{X}\,\mathbf{1}_n$ and

$$
\begin{aligned}
\Delta_3 &= \|\mathbf{Q}^{-1/2}\,\mathbf{A}\mathbf{D}\,(\mathbf{S}^{1/2}\mathbf{X}^{-1/2})\,\mathbf{X}\,\mathbf{1}_n\|_2 \\
&= \|\mathbf{Q}^{-1/2}\,\mathbf{A}\mathbf{D}\,(\mathbf{S}\mathbf{X})^{1/2}\,\mathbf{1}_n\|_2 \\
&\leq \|\mathbf{Q}^{-1/2}\,\mathbf{A}\mathbf{D}\|_2 \|(\mathbf{S}\mathbf{X})^{1/2}\,\mathbf{1}_n\|_2 \\
&\leq \sqrt{2}\,\sqrt{\sum_{i=1}^{n}x_i s_i} = \sqrt{2n\,\mu}\,,
\end{aligned}
\tag{57}
$$

where the inequalities follows respectively from submultiplicativity and eqn. (6).

**Bounding $\Delta_4$.** Putting $\mathbf{r}_d = \eta\,\mathbf{r}_d^0$, we have

$$
\begin{aligned}
\Delta_4 &= \eta\|\mathbf{Q}^{-1/2}\,\mathbf{A}\,\mathbf{D}^2\mathbf{r}_d^0\|_2 \\
&\leq \eta\|\mathbf{Q}^{-1/2}\,\mathbf{A}\mathbf{D}\|_2 \|(\mathbf{X}\mathbf{S})^{-1/2}\mathbf{X}\mathbf{r}_d^0\|_2 \\
&\leq \sqrt{2}\eta \, \|(\mathbf{X}\mathbf{S})^{-1/2}\mathbf{X}(\mathbf{A}^\top\mathbf{y}^0 + \mathbf{s}^0 - \mathbf{c})\|_2 \\
&\leq \sqrt{2}\eta \, \|(\mathbf{X}\mathbf{S})^{-1/2}\|_2 \, \|\mathbf{X}(\mathbf{A}^\top\mathbf{y}^0 + \mathbf{s}^0 - \mathbf{c})\|_2\,,
\end{aligned}
$$

where the above inequalities follow from submultiplicativity and eqn. (6). Now, applying eqn. (43c) and $\|(\mathbf{X}\mathbf{S})^{-1/2}\|_2 \leq \frac{1}{\sqrt{(1-\gamma)\mu}}$, we further have

$$
\Delta_4 \leq 6\sqrt{2}n\sqrt{\frac{\mu}{1-\gamma}}
\tag{58}
$$

**Final bound.** Combining eqns. (53), (55), (56), (57) and (58)

$$
\|\mathbf{Q}^{-1/2}\mathbf{p}\|_2 \leq \sqrt{2}\left(\frac{9n}{\sqrt{1-\gamma}} + \sigma\sqrt{\frac{n}{1-\gamma}} + \sqrt{n}\right)\sqrt{\mu}\,.
\tag{59}
$$

This concludes the proof of Lemma 12. ∎

**Lemma 13** *Let the sketching matrix $\mathbf{W}$ satisfy the conditions in eqns. (6) and (7). Then, after $t \geq \frac{\log(4\sqrt{6n}\,\psi/\gamma\sigma)}{\log(1/\varsigma)}$ iterations of the CG solver in Algorithm 1, we have the following:*

$$
\|\tilde{\mathbf{f}}^{(t)}\|_2 \leq \frac{\gamma\sigma}{4\sqrt{n}}\sqrt{\mu} \quad\text{and}\quad \|\mathbf{v}\|_2 \leq \frac{\gamma\sigma}{4}\mu\,,
$$

*where $\psi = \left(\frac{9n}{\sqrt{1-\gamma}} + \sigma\sqrt{\frac{n}{1-\gamma}} + \sqrt{n}\right)$ and $\tilde{\mathbf{f}}^{(t)} = \mathbf{Q}^{-1/2}\mathbf{A}\mathbf{D}^2\mathbf{A}^\top\mathbf{Q}^{-1/2}\tilde{\mathbf{z}}^t - \mathbf{Q}^{-1/2}\mathbf{p}$ is the residual of the solver.*

**Proof** Combining Lemma 12 and the condition in eqn. (7), we have

$$\|\tilde{\mathbf{f}}^{(t)}\|_2 \le \zeta^t \psi \sqrt{2\mu}. \tag{60}$$

Now, $\|\tilde{\mathbf{f}}^{(t)}\|_2 \le \frac{\gamma\sigma}{4\sqrt{n}}\sqrt{\mu}$ holds if $\sqrt{2}\psi\,\zeta^t\sqrt{\mu} \le \frac{\gamma\sigma}{4\sqrt{n}}\sqrt{\mu}$, which holds if $\left(\frac{1}{\zeta}\right)^t \ge \frac{4\sqrt{2n}\,\psi}{\gamma\sigma}$. The last inequality holds for our choice of $t$. Next, combining Lemma 4 and eqn. (60) we get

$$\|\mathbf{v}\|_2 \le \sqrt{3n\mu}\,\|\tilde{\mathbf{f}}^{(t)}\|_2 \le \sqrt{6n}\,\zeta^t\psi\mu$$

Therefore, $\|\mathbf{v}\|_2 \le \frac{\gamma\sigma\mu}{4}$ holds if $\sqrt{6n}\psi\,\zeta^t\psi\mu \le \frac{\gamma\sigma\mu}{4}$, which holds for our choice of $t$. Now, fixing $\gamma$, $\sigma$, and $\zeta$, after $t = \mathcal{O}(\log n)$ iterations of Algorithm 1 the conclusions of the lemma hold. ∎

### F.2 Determining Step-size, Bounding the Number of Iterations, and Proof of Theorem 1

Let $(\hat{\triangle}\mathbf{x}, \hat{\triangle}\mathbf{y}, \hat{\triangle}\mathbf{s})$ respectively satisfies eqns. (17), (18) and (14b). We rewrite the system in the following alternative form

$$\mathbf{A}\hat{\triangle}\mathbf{x} = -\mathbf{r}_p, \tag{61a}$$

$$\mathbf{A}^\mathsf{T}\hat{\triangle}\mathbf{y} + \hat{\triangle}\mathbf{s} = -\mathbf{r}_d, \tag{61b}$$

$$\mathbf{X}\hat{\triangle}\mathbf{s} + \mathbf{S}\hat{\triangle}\mathbf{x} = -\mathbf{XS}\,\mathbf{1}_n + \sigma\mu\,\mathbf{1}_n - \mathbf{v}. \tag{61c}$$

Indeed, first we now show how to satisfy eqns. (17), (18) and (14b) from eqn. (61). Pre-multiplying both sides of eqn. (61c) by $\mathbf{AS}^{-1}$ and noting that $\mathbf{D}^2 = \mathbf{XS}^{-1}$, we get

$$\mathbf{AD}^2\hat{\triangle}\mathbf{s} + \mathbf{A}\hat{\triangle}\mathbf{x} = -\mathbf{AX}\mathbf{1}_n + \sigma\mu\mathbf{AS}^{-1}\mathbf{1}_n - \mathbf{AS}^{-1}\mathbf{v}$$

$$\Rightarrow \mathbf{AD}^2\hat{\triangle}\mathbf{s} = \mathbf{r}_p - \mathbf{Ax} + \sigma\mu\mathbf{AS}^{-1}\mathbf{1}_n - \mathbf{AS}^{-1}\mathbf{v}. \tag{62}$$

Eqn. (62) holds as $\mathbf{AX}\mathbf{1}_n = \mathbf{Ax}$ and, from eqn. (61a), $\mathbf{A}\hat{\triangle}\mathbf{x} = -\mathbf{r}_p$. Next, pre-multiplying eqn. (61b) by $\mathbf{AD}^2$, we get

$$\mathbf{AD}^2\mathbf{A}^\mathsf{T}\hat{\triangle}\mathbf{y} + \mathbf{AD}^2\hat{\triangle}\mathbf{s} = -\mathbf{AD}^2\mathbf{r}_d$$

$$\Rightarrow \mathbf{AD}^2\mathbf{A}^\mathsf{T}\hat{\triangle}\mathbf{y} = -\mathbf{r}_p + \mathbf{Ax} - \sigma\mu\mathbf{AS}^{-1}\mathbf{1}_n - \mathbf{AD}^2\mathbf{r}_d + \mathbf{AS}^{-1}\mathbf{v} = \mathbf{p} + \mathbf{AS}^{-1}\mathbf{v}. \tag{63}$$

The first equality in eqn. (63) follows from eqn. (62) and the definition of $\mathbf{p}$ in eqn. (16). This establishes eqn. (18). Eqn. (14b) directly follows from eqn. (61b). Finally, we get eqn. (17) by pre-multiplying eqn. (61c) by $\mathbf{S}^{-1}$.

Next, we define each new point traversed by the algorithm as $(\mathbf{x}(\alpha), \mathbf{y}(\alpha), \mathbf{s}(\alpha))$, where

$$(\mathbf{x}(\alpha), \mathbf{y}(\alpha), \mathbf{s}(\alpha)) := (\mathbf{x}, \mathbf{y}, \mathbf{s}) + \alpha(\hat{\triangle}\mathbf{x}, \hat{\triangle}\mathbf{y}, \hat{\triangle}\mathbf{s}) \tag{64}$$

$$\mu(\alpha) := \mathbf{x}(\alpha)^T\mathbf{s}(\alpha)/n \tag{65}$$

$$\mathbf{r}(\alpha) := \mathbf{r}\left(\mathbf{x}(\alpha), \mathbf{s}(\alpha), \mathbf{y}(\alpha)\right). \tag{66}$$

The goal in this section is to bound the number of iterations required by Algorithm 2. Towards that end, we bound the magnitude of the step size $\alpha$. First, we provide an upper bound on $\alpha$, which allows us to show that each new point $(\mathbf{x}(\alpha), \mathbf{s}(\alpha), \mathbf{y}(\alpha))$ traversed by the algorithm stays within the neighborhood $\mathcal{N}(\gamma)$. Second, we provide a lower bound on $\alpha$, which allows us to bound the number of iterations required. We use multiple lemmas from [38], which we reproduce here, without their proofs.

First, we provide an upper bound on $\alpha$, ensuring that each new point $(\mathbf{x}(\alpha), \mathbf{y}(\alpha), \mathbf{s}(\alpha))$ traversed by the algorithm stays within the neighborhood $\mathcal{N}(\gamma)$.

**Lemma 14 (Lemma 3.5 of [38])** *Assume $(\hat{\triangle}\mathbf{x}, \hat{\triangle}\mathbf{y}, \hat{\triangle}\mathbf{s})$ satisfies eqns. (61) for some $\sigma > 0$, $(\mathbf{x}, \mathbf{y}, \mathbf{s}) \in \mathcal{N}(\gamma)$ (for $\gamma \in (0, 1)$), and $\|\mathbf{v}\|_2 \le \frac{\gamma\sigma\mu}{4}$. Then, $(\mathbf{x}(\alpha), \mathbf{y}(\alpha), \mathbf{s}(\alpha)) \in \mathcal{N}(\gamma)$ for every scalar $\alpha$ such that*

$$0 \le \alpha \le \min\left\{1, \frac{\gamma\sigma\mu}{4\|\hat{\triangle}\mathbf{x} \circ \hat{\triangle}\mathbf{s}\|_\infty}\right\}. \tag{67}$$

We now provide a lower bound on the values of $\bar{\alpha}$ and the corresponding $\mu(\bar{\alpha})$; see Algorithm 2.

**Lemma 15 (Lemma 3.6 of [38])** *In each iteration of Algorithm 2, if $\|\mathbf{v}\|_2 \leq \frac{\gamma\sigma\mu}{4}$, then the step size $\bar{\alpha}$ satisfies*

$$\bar{\alpha} \geq \min\left\{1, \frac{\min\{\gamma\sigma, (1 - \frac{5}{4}\sigma)\}\mu}{4\|\hat{\Delta}x \circ \hat{\Delta}s\|_\infty}\right\} \tag{68}$$

*and*

$$\mu(\bar{\alpha}) = \left[1 - \frac{\bar{\alpha}}{2}(1 - \frac{5}{4}\sigma)\right]\mu. \tag{69}$$

At this point, we have provided a lower bound (eqn. (68)) for the allowed values of the step size $\bar{\alpha}$. Next, we show that this lower bound is bounded away from zero. From eqn. (68) this is equivalent to showing that $\|\hat{\Delta}\mathbf{x} \circ \hat{\Delta}\mathbf{s}\|_\infty$ is bounded.

**Lemma 16 (Lemma 3.7 of [38] (slightly modified))** *Let $(\mathbf{x}^0, \mathbf{y}^0, \mathbf{s}^0)$ be the initial point with $(\mathbf{x}^0, \mathbf{s}^0) > 0$ and $(\mathbf{x}^0, \mathbf{s}^0) \geq (\mathbf{x}^*, \mathbf{s}^*)$ for some $(\mathbf{x}^*, \mathbf{y}^*, \mathbf{s}^*) \in \mathcal{S}$. Let $(\mathbf{x}, \mathbf{y}, \mathbf{s}) \in \mathcal{N}(\gamma)$ be such that $\mathbf{r} = \eta\mathbf{r}^0$ for some $\eta \in [0, 1]$ and $\|\mathbf{v}\|_2 \leq \frac{\gamma\sigma\mu}{4}$. Then, the search direction $(\hat{\Delta}\mathbf{x}, \hat{\Delta}\mathbf{y}, \hat{\Delta}\mathbf{s})$ produced by Algorithm 2 at each iteration satisfies*

$$\max\{\|\mathbf{D}^{-1}\hat{\Delta}\mathbf{x}\|_2, \|\mathbf{D}\hat{\Delta}\mathbf{s}\|_2\} \leq \left(1 + \frac{\sigma^2}{1 - \gamma} - 2\sigma\right)^{1/2}\sqrt{n\mu} + \frac{6n}{\sqrt{(1 - \gamma)}}\sqrt{\mu} + \frac{\gamma\sigma}{4\sqrt{1 - \gamma}}\sqrt{\mu}. \tag{70}$$

We should note here that the above lemma is slightly different than Lemma 3.7 of [38]. Indeed, Lemma 3.7 of [38] actually proves the following bound:

$$\max\{\|\mathbf{D}^{-1}\hat{\Delta}\mathbf{x}\|_2, \|\mathbf{D}\hat{\Delta}\mathbf{s}\|_2\} \leq \left(1 + \frac{\sigma^2}{1 - \gamma} - 2\sigma\right)^{1/2}\sqrt{n\mu} + \frac{6n}{\sqrt{(1 - \gamma)}}\sqrt{\mu} + \frac{\gamma\sigma}{4\sqrt{n}}\sqrt{\mu}. \tag{71}$$

Notice that there is slight difference in the last term in the right-hand side, which does not asymptotically change the bound. The underlying reason for this difference is the fact that [38] constructed the vector $\mathbf{v}$ differently. In our case, we need to bound $\|(\mathbf{XS})^{-1/2}\mathbf{v}\|_2$, which we do as follows:

$$\|(\mathbf{XS})^{-1/2}\mathbf{v}\|_2 \leq \|(\mathbf{XS})^{-1/2}\|_2 \|\mathbf{v}\|_2 \leq \frac{1}{\min_i \sqrt{x_i s_i}}\frac{\gamma\sigma\mu}{4}, \tag{72}$$

where in the above expression we use the fact that $\|(\mathbf{XS})^{-1/2}\|_2 = \frac{1}{\min_i \sqrt{x_i s_i}}$. Now as $(\mathbf{x}, \mathbf{y}, \mathbf{s}) \in \mathcal{N}(\gamma)$, we further have $x_i s_i \geq (1 - \gamma)\mu$ for all $i = 1 \ldots n$. Combining this with eqn. (72), we get

$$\|(\mathbf{XS})^{-1/2}\mathbf{v}\|_2 \leq \frac{\gamma\sigma\mu}{4\sqrt{(1 - \gamma)\mu}} = \frac{\gamma\sigma}{4\sqrt{1 - \gamma}}\sqrt{\mu}. \tag{73}$$

On the other hand, [38] had a different construction of $\mathbf{v}$ for which $\|(\mathbf{XS})^{-1/2}\mathbf{v}\|_2 = \|\tilde{\mathbf{f}}^{(t)}\|_2$ holds. Therefore they had the following bound:

$$\|(\mathbf{XS})^{-1/2}\mathbf{v}\|_2 = \|\tilde{\mathbf{f}}^{(t)}\|_2 \leq \frac{\gamma\sigma}{4\sqrt{n}}\sqrt{\mu}.$$

Also, note that after correcting the approximation error of the CG solver using $\mathbf{v}$, the primal and dual residuals $\mathbf{r} = (\mathbf{r}_p, \mathbf{r}_d)$ corresponding to an iterate $(\mathbf{x}, \mathbf{y}, \mathbf{s}) \in \mathcal{N}(\gamma)$ always lie on the line segment between zero and $\mathbf{r}^{(0)}$. In other words, $\mathbf{r} = \eta\mathbf{r}^{(0)}$ always holds for some $\eta \in [0, 1]$. This was formally proven in Lemma 3.3 of [38].

The next lemma bounds the number of iterations that Algorithm 2 needs when started with an infeasible point that is sufficiently positive.

**Lemma 17 (Theorem 2.6 of [38])** *Assume that the constants $\gamma$ and $\sigma$ are such that $\max\{\gamma^{-1}, (1 - \gamma)^{-1}, \sigma^{-1}, (1 - \frac{5}{4}\sigma)^{-1}\} = \mathcal{O}(1)$. Let the initial point $(\mathbf{x}^0, \mathbf{s}^0, \mathbf{y}^0)$ satisfy $(\mathbf{x}^0, \mathbf{s}^0) \geq (\mathbf{x}^*, \mathbf{s}^*)$ for some $(\mathbf{x}^*, \mathbf{s}^*, \mathbf{y}^*) \in \mathcal{S}$ and $\|\mathbf{v}\|_2 \leq \frac{\gamma\sigma\mu}{4}$. Algorithm 2 generates an iterate $(\mathbf{x}^k, \mathbf{s}^k, \mathbf{y}^k)$ satisfying $\mu_k \leq \epsilon\mu_0$ and $\|\mathbf{r}^k\|_2 \leq \epsilon\|\mathbf{r}^0\|_2$ after $\mathcal{O}(n^2 \log 1/\epsilon)$ iterations.*

Finally, Theorem 1 follows from Lemmas 13 and 17.

# Appendix G  Additional Notes on Experiments

| Problem | Size | Sketch IPM w/ Precond. CG | | | | Stand. IPM w/ Unprec. CG | | | IPM w/ Dir. |
|---------|------|-----|--------|---------|----------------|--------|---------|-------------------|---------|
| | $(m \times N)$ | $w$ | In. It. | Out. It. | $\kappa_{\text{Sk}}$ | In. It. | Out. It. | $\kappa_{\text{Stan}}$ | Out. It. |
| ARCENE | $(100 \times 10K)$ | 200 | **30** | 50 | 38.09 | **1.1K** | 59 | $4.4 \times 10^8$ | 50 |
| DEXTER | $(300 \times 20K)$ | 500 | **39** | 39 | 75.42 | **4.6K** | 39 | $7.6 \times 10^9$ | 39 |
| DrivFace | $(606 \times 6.4K)$ | $1K$ | **50** | 42 | 68.87 | **139K** | 43 | $17 \times 10^{12}$ | 42 |
| Gene RNA | $(801 \times 20K)$ | $2K$ | **27** | 44 | 20.03 | **101K** | 208 | $4.7 \times 10^{12}$ | 44 |

Table 1: Comparison of (our) sketched IPM with CG, standard IPM with CG, and Standard IPM with a direct solver, for the $\ell_1$-SVM problem on UCI Machine Learning Repository [20] data sets. Across all, $\tau = 10^{-9}$ and a relative error of $10^{-3}$ or less was achieved. We define $\kappa_{\text{Sk}} = \kappa(\mathbf{Q}^{-1/2}\mathbf{A}\mathbf{D}^2\mathbf{A}^{\mathsf{T}}\mathbf{Q}^{-1/2})$ and $\kappa_{\text{Stan}} = \kappa(\mathbf{A}\mathbf{D}^2\mathbf{A}^T)$.

## G.1  Support Vector Machines (SVMs)

The classical $\ell_1$-SVM problem is as follows. We consider the task of fitting an SVM to data pairs $S = \{(x_i, y_i)\}_{i=1}^m$, where $x_i \in \mathbb{R}^N$ and $y_i \in \{+1, -1\}$ is a label for each data pair. Here, $m$ is the number of training points, and $N$ is the feature dimension. The SVM problem with an $\ell_1$ regularizer has the following form.

$$\begin{aligned} \underset{w}{\text{minimize}} \quad & \|w\|_1 \\ \text{subject to} \quad & y_i(w^T x_i + b') \geq 1, \quad \forall i \in [m]. \end{aligned} \tag{74}$$

This problem can be written as an LP by introducing the variables $w^+$ and $w^-$, where $w = w^+ - w^-$. The objective becomes $\sum_j^n w_j^+ + w_j^-$, and we constrain $w_i^+ \geq 0$ and $w_i^- \geq 0$. Note that the size of the constraint matrix in the LP becomes $(m \times (2N + 1))$, where $m$ is the number of training points, and $N$ is the feature dimension.

## G.2  Random Data

We generate random synthetic instances of linear programs as follows. To generate $A \in \mathbb{R}^{m \times n}$, we set $a_{ij} \sim_{i.i.d.} U(0, 1)$ with probability $p$ and $a_{ij} = 0$ otherwise. We then add $\min\{m, n\}$ i.i.d. draws from $U(0, 1)$ to the main diagonal, to ensure each row of $A$ has at least one nonzero entry. We set $b = Ax + 0.1z$, where $x$ and $z$ are random vectors drawn from $N(0, 1)$. Finally, we set $c \sim N(0, 1)$.

## G.3  Real Data Descriptions

The following is how we made use a gene expression cancer RNA-Sequencing data set, taken from the UCI Machine Learning repository. It is part of the RNA-Seq (HiSeq) PANCAN data set [49], and is a random extraction of gene expressions from patients who have different types of tumors: BRCA, KIRC, COAD, LUAD and PRAD. We considered the binary classification task of identifying BRCA versus other types.

The following is how we made use of the DrivFace data set taken from the UCI Machine Learning repository. In the DrivFace data set, each sample corresponds to an image of a human subject, taken while driving in real scenarios. Each image is labeled as corresponding to one of 3 possible gaze directions (left, straight, or right). We considered the binary classification task of identifying two different gaze directions: (straight, or to either side left or right).

## G.4  Additional Experiments

Here we include additional experiments. Figure 2 illustrates the convergence and conditioning behavior for the DEXTER data set. We see a similar behavior as found for the ARCENE data set in Figure 1. Figure 3 displays more results for the ARCENE data set.

Figure 2: *DEXTER data set*: Our algorithm (Sk. IPM) requires an order of magnitude fewer inner iterations than the Standard IPM with CG, at each outer iteration, as demonstrated in (a). This is possible due to the improved conditioning of $\mathbf{Q}^{-1/2}\mathbf{AD}^2\mathbf{A}^\mathsf{T}\mathbf{Q}^{-1/2}$ compared to $\mathbf{AD}^2\mathbf{A}^T$, demonstrated in (b). For all, *tolCG* $= 10^{-5}$, $\tau = 10^{-9}$.

Figure 3: *ARCENE data set*: As $w$ increases, (a) the number of inner iterations decreases, and is relatively robust to *tolCG*, and, (b) the condition number decreases as well.



[Supplementary Material 2 · Dataset.pdf]

# Design of experiments for the NIPS 2003 variable selection benchmark

Isabelle Guyon – July 2003
isabelle@clopinet.com

Background:

Results published in the field of feature or variable selection (see e.g. the special issue of JMLR on variable and feature selection: http://www.jmlr.org/papers/special/feature.html) are for the most part on different data sets or used different data splits, which make them hard to compare. We formatted a number of datasets for the purpose of benchmarking variable selection algorithms in a controlled manner[1]. The data sets were chosen to span a variety of domains (cancer prediction from mass-spectrometry data, handwritten digit recognition, text classification, and prediction of molecular activity). One dataset is artificial. We chose data sets that had sufficiently many examples to create a large enough test set to obtain statistically significant results. The input variables are continuous or binary, sparse or dense. All problems are two-class classification problems. The similarity of the tasks allows participants to enter results on all data sets. Other problems will be added in the future.

Method:

Preparing the data included the following steps:

- Preprocessing data to obtain features in the same numerical range (0 to 999 for continuous data and 0/1 for binary data).
- Adding "random" features distributed similarly to the real features. In what follows we refer to such features as **probes** to distinguish them from the real features. This will allow us to rank algorithms according to their ability to filter out irrelevant features.
- Randomizing the order of the patterns and the features to homogenize the data.
- Training and testing on various data splits using simple feature selection and classification methods to obtain baseline performances.
- Determining the approximate number of *test examples* needed for the test set to obtain statistically significant benchmark results using the rule-of-thumb $n_{test} = 100/p$, where p is the test set error rate (see What size test set gives good error rate estimates? *I. Guyon, J. Makhoul, R. Schwartz, and V. Vapnik.* PAMI, 20 (1), pages 52--64, IEEE. 1998, http://www.clopinet.com/isabelle/Papers/test-size.ps.Z). Since the test error rate of the classifiers of the benchmark is unknown, we used the results of the baseline method and added a few more examples.
- Splitting the data into training, validation and test set. The size of the validation set is usually smaller than that of the test set to keep as much training data as possible.

Both validation and test set truth-values (labels) are withheld during the benchmark. The validation set serves as development test set. During the time allotted to the participants to try methods on the data, participants are allowed to send the *validation set results* (in

the form of classifier outputs) and obtain result scores. Such score are made available to all participants to stimulate research. At the end of the benchmark, the participants send their *test set results.* The scores on the test set results are disclosed simultaneously to all participants after the benchmark is over.

Data formats:
All the data sets are in the same format and include 8 files in ASCII format:
**dataname.param**: Parameters and statistics about the data
**dataname.feat**: Identities of the features (in the order the features are found in the data).
**dataname_train.data**: Training set (a spase or a regular matrix, patterns in lines, features in columns).
**dataname_valid.data**: Validation set.
**dataname_test.data**: Test set.
**dataname_train.labels**: Labels (truth values of the classes) for training examples.
**dataname_valid.labels**: Validation set labels (withheld during the benchmark).
**dataname_test.labels**: Test set labels  (withheld during the benchmark).
The matrix data formats used are:
- For regular matrices: a space delimited file with a new-line character at the end of each line.
- For sparse matrices with binary values: for each line of the matrix, a space delimited list of indices of the non-zero values. A new-line character at the end of each line.
- For sparse matrices with non-binary values: for each line of the matrix, a space delimited list of indices of the non-zero values followed by the value itself, separated from it index by a colon. A new-line character at the end of each line.

The results on each dataset should be formatted in 7 ASCII files:
**dataname_train.resu**: +-1 classifier outputs for training examples (mandatory for **final** submissions).
**dataname_valid.resu**: +-1 classifier outputs for validation examples (mandatory for **development and final** submissions).
**dataname_test.resu**: +-1 classifier outputs for test examples (mandatory for **final** submissions).
**dataname_train.conf**: confidence values for training examples (optional).
**dataname_valid.conf**: confidence values  for validation examples (optional).
**dataname_test.conf**: confidence values for test examples (optional).
**dataname.feat**: list of features selected (one integer feature number per line, starting from one, ordered from the most important to the least important if such order exists). If no list of features is provided, it will be assumed that all the features were used.
*Format for classifier outputs:*
- All .resu files should have one +-1 integer value per line indicating the prediction for the various patterns.
- All .conf files should have one decimal positive numeric value per line indicating classification confidence. The confidence values can be the absolute discriminant values. They do not need to be normalized to look like probabilities. They will be used to compute ROC curves and Area Under such Curve (AUC).

Result rating:
The classification results are rated with the balanced error rate (the average of the error rate on training examples and on test examples). The area under the ROC curve is also be computed, if the participants provide classification confidence scores in addition to class label predictions. But **the relative strength of classifiers is judged only on the balanced error rate**. The participants are invited to provide the list of features used. **For methods having performance differences that are not statistically significant, the method using the smallest number of features wins**. If no feature set is provided, it is assumed that all the features were used. The organizers may then provide the participants with one or several test sets containing only the features selected to verify the accuracy of the classifier when it uses those features only. The proportion of random probes in the feature set is also be computed. It is used to assess the relative strength of method with non-statistically significantly different error rates and a relative difference in number of features that is less than 5%. **In that case, the method with smallest number of random probes in the feature set wins**.

## Dataset A: ARCENE

1) **Topic**
The task of ARCENE is to distinguish **cancer** *versus* normal patterns from mass-spectrometric data. This is a two-class classification problem with continuous input variables.

2) **Sources**
   a. Original owners
The data were obtained from two sources: The National Cancer Institute (NCI) and the Eastern Virginia Medical School (EVMS). All the data consist of mass-spectra obtained with the SELDI technique. The samples include patients with cancer (ovarian or prostate cancer), and healthy or control patients.

NCI ovarian data:
The data were originally obtained from http://clinicalproteomics.steem.com/download-ovar.php. We use the 8/7/02 data set:
http://clinicalproteomics.steem.com/Ovarian%20Dataset%208-7-02.zip.
The data includes 253 spectra, including 91 controls and 162 cancer spectra.
Number of features: 15154.

NCI prostate cancer data:
The data were originally obtained from
http://clinicalproteomics.steem.com/JNCI%20Data%207-3-02.zip on the web page
http://clinicalproteomics.steem.com/download-prost.php.
There are a total of 322 samples: 63 samples with no evidence of disease and PSA level less than 1; 190 samples with benign prostate with PSA levels greater than 4; 26 samples with prostate cancer with PSA levels 4 through 10; 43 samples with prostate cancer with PSA levels greater than 10. Therefore, there are 253 normal samples and 69 disease samples. The original training set is composed of 56 samples:

- 25 samples with no evidence of disease and PSA level less than 1 ng/ml.
- 31 biopsy-proven prostate cancer with PSA level larger than 4 ng/ml.

But the exact split is not given in the paper or on the web site. The original test set contains the remaining 266 samples (38 cancer and 228 normal).
Number of features: 15154.

EVMS prostate cancer data:
The data is downloadable from:
http://www.evms.edu/vpc/seldi/.
The training data data includes 652 spectra from 326 patients (spectra are in duplicate) and includes 318 controls and 334 cancer spectra. Study population: 167 prostate cancer (84 state 1 and 2; 83 stage 3 and 4), 77 benign prostate hyperplasia, and 82 age-matched normals. The test data includes 60 additional patients. The labels for the test set are not provided with the data, so the test spectra are not used for the benchmark.
Number of features: 48538.

     b.  Donor of database

This version of the database was prepared for the NIPS 2003 variable and feature selection benchmark by Isabelle Guyon, 955 Creston Road, Berkeley, CA 94708, USA (isabelle@clopinet.com).

     c.  Date received: August 2003.

**3) Past usage**

NCI ovarian cancer original paper:
"Use of proteomic patterns in serum to identify ovarian cancer *Emanuel F Petricoin III, Ali M Ardekani, Ben A Hitt, Peter J Levine, Vincent A Fusaro, Seth M Steinberg, Gordon B Mills, Charles Simone, David A Fishman, Elise C Kohn, Lance A Liotta.* THE LANCET • Vol 359 • February 16, 2002 • www.thelancet.com" are so far not reproducible.
**Note: The data used is a newer set of spectra obtained after the publication of the paper and of better quality.**
100% accuracy is easily achieved on the test set using various data splits on this version of the data.
NCI prostate cancer original paper:
Serum proteomic patterns for detection of prostate cancer. Petricoin et al. Journal of the NCI, Vol. 94, No. 20, Oct. 16, 2002. The test results of the paper are shown in Table A.1.

| FP | FN | TP | TN | Error | 1-error | Specificity | Sensitivity |
|----|----|----|----|-------|---------|-------------|-------------|
| 51 | 2 | 36 | 177 | 20.30% | 79.70% | 77.63% | 94.74% |

**Table A.1: Results of Petricoin et al. on the NCI prostate cancer data. Fp=false positive, FN=false negative, TP=true positive, TN=true negative.**
Error=(FP+FN)/(FP+FN+TP+TN), Specificity=TN/(TN+FP), Sensitivity=TP/(TP+FN).

EVMS prostate cancer original paper:
Serum Protein Fingerprinting Coupled with a Pattern-matching Algorithm Distinguishes Prostate Cancer from Benign Prostate Hyperplasia and Healthy Men, Bao-Ling Adam, et al., CANCER RESEARCH 62, 3609–3614, July 1, 2002.

In the following excerpt from the original paper some baseline results are reported:
"Surface enhanced laser desorption/ionization mass spectrometry protein profiles of serum from 167 PCA patients, 77 patients with benign prostate hyperplasia, and 82 age-matched unaffected healthy men were used to train and develop a decision tree classification algorithm that used a nine-protein mass pattern that correctly classified 96% of the samples. A blinded test set, separated from the training set by a stratified random sampling before the analysis, was used to determine the sensitivity and specificity of the classification system. A sensitivity of 83%, a specificity of 97%, and a positive predictive value of 96% for the study population and 91% for the general population were obtained when comparing the PCA *versus* noncancer (benign prostate hyperplasia/healthy men) groups**."**

## 4) **Experimental design**

We merge the datasets from the three different sources (253+322+326=901 samples). We obtained 91+253+159=503 control samples (negative class) and 162+69+167=398 cancer samples (positive class). The motivations for merging datasets include:
- Obtaining enough data to be able to cut a sufficient size test set.
- Creating a problem where possibly non-linear classifiers and non-linear feature selection methods might outperform linear methods. The reason is that there will be in each class different clusters corresponding differences in disease, gender, and sample preparation.
- Finding out whether there are features that are generic of the separation cancer *vs*. normal across various cancers.

We designed a preprocessing that is suitable for mass-spec data and applied it to all the data sets to reduce the disparity between data sources. The preprocessing consists of the following steps:
- **Limiting the mass range:** We eliminated small masses under m/z=200 that include usually chemical noise specific to the MALDI/SELDI process (influence of the "matrix"). We also eliminated large masses over m/z=10000 because few features are usually relevant in that domain and we needed to compress the data.
- **Averaging the technical repeats:** In the EVMS data, two technical repeats were available. We averaged them because we wanted to have examples in the test set that are independent so that we can apply simple statistical tests.
- **Removing the baseline:** We subtracted in a window the median of the 20% smallest values. An example of baseline detection is shown in Figure A.1.
- **Smoothing:** The spectra were slightly smoothed with an exponential kernel in a window of size 9.
- **Re-scaling**: The spectra were divided by the median of the 5% top values.
- **Taking the square root**. The square root of the all values was taken.
- **Aligning the spectra**: We slightly shifted the spectra collections of the three datasets so that the peaks of the average spectrum would be better aligned (Figures A.2 and A.3). As a result, the mass-over-charge (m/z) values that identify the features in the aligned data are imprecise. We took the NCI prostate cancer m/z as reference.
- **Limiting more the mass range**: To eliminate border effects, the spectra border were cut.

- **Soft thresholding the values:** After examining the distribution of values in the data matrix, we subtracted a threshold and equaled to zero all the resulting values that were negative. In this way, we kept only about 50% of non-zero value, which represents significant data compression (see Figure A.4).
- **Quantizing:** We quantized the values to 1000 levels.

The resulting data set including all training and test data merged from the three sources has 901 patterns from 2 classes and 9500 features. We remove one pattern to obtain the round number 900. At every step, we checked that the change in performance of a linear SVM classifier trained and tested on a random split of the data was not significant. On that basis, we have some confidence that our preprocessing did not alter significantly the information content of the data. We further manipulated the data to add random "probes":

- We identified the region of the spectra with least information content using an interval search for the region that gave worst prediction performance of a linear SVM (indices 2250-4750). We replaced the features in that region by "random probes" obtained by randomly permuting the values in the columns of the data matrix.
- We identified another region of low information content: 6500-7000. We added 500 random probes that are permutations of those features.

After such manipulations, the data had 10000 features, including 7000 real features and 3000 random probes. The reason for not adding more probes is purely practical: non-sparse data cannot be compressed sufficiently to be stored and transferred easily in the context of a benchmark.

Figure A.1: Example of baseline detection (EVMS data).

**Figure A.2: Central part of the spectra before alignment.** We show in red the average NCI ovarian spectra, in blue the average NCI prostate spectra, and in green the average EVMS prostate spectra.

**Figure A.3: Central part of the spectra after alignment.** We show in red the average NCI ovarian spectra, in blue the average NCI prostate spectra, and in green the average EVMS prostate spectra.

**Figure A.4: Distributions of the values in the ARCENE data after preprocessing.**

**Figure A.5: Heat map of the training set of the ARCENE data.** We represent the data matrix (patients in line and features in columns). The values are clipped at 500 to increase the contrast. The values are then mapped to colors according to the color-map on the right. The stripe beyond the 10000 feature index indicated the class labels: +1=red, -1=green.

### 5) Number of examples and class distribution

|  | Positive ex. | Negative ex. | Total | Check sum |
|---|---|---|---|---|
| **Training set** | 44 | 56 | 100 | 70726744 |
| **Validation set** | 44 | 56 | 100 | 71410108 |
| **Test set** | 310 | 390 | 700 | 493023349 |
| **All** | 398 | 502 | 900 | 635160201 |

### 6) Type of input variables and variable statistics

| Real variables | Random probes | Total |
|---|---|---|
| 7000 | 3000 | 10000 |

All variables are **integer** quantized on 1000 levels. There are **no missing values**. The data is not very sparse, but for data compression reasons, we thresholded the values. Approximately 50% of the entries are non zero. The data was saved as a **non-sparse** matrix.

### 7) Results of the run of the lambda method and linear SVM

Before the benchmark, we ran some simple methods to determine what an appropriate number of examples should be. The "lambda" method (provided with the sample code) had approximately a 30% test error rate ans a linear SVM trained on all features a 15% error rate. The rule of thumb number_of_test_examples=100/test_errate=100/.15=667 led us to keep 700 examples for testing.
The best benchmark error rates are of the order 15%, which confirms that our estimate was correct.

## Dataset B: GISETTE

### 1) Topic
The task of GISETTE is to discriminate between to confusable handwritten **digits**: the four and the nine. This is a two-class classification problem with sparse continuous input variables.

### 2) Sources
#### a. Original owners
The data set was constructed from the MNIST data that is made available by Yann LeCun of the NEC Research Institute at http://yann.lecun.com/exdb/mnist/.
The digits have been size-normalized and centered in a fixed-size image of dimension 28x28. We show examples of digits in Figure B1.

**Figure B1: Two examples of digits from the MNIST database.**
We used only examples of fours and nines to prepare our dataset.
b. Donor of database
This version of the database was prepared for the NIPS 2003 variable and feature
selection benchmark by Isabelle Guyon, 955 Creston Road, Berkeley, CA 94708, USA
(isabelle@clopinet.com).
c. Date received: August 2003.

**3) Past usage**
Many methods have been tried on the MNIST database. Here is an abbreviated list from
http://yann.lecun.com/exdb/mnist/:

| METHOD | TEST ERROR RATE (%) |
| --- | --- |
| linear classifier (1-layer NN) | 12.0 |
| linear classifier (1-layer NN) [deskewing] | 8.4 |
| pairwise linear classifier | 7.6 |
| K-nearest-neighbors, Euclidean | 5.0 |
| K-nearest-neighbors, Euclidean, deskewed | 2.4 |
| 40 PCA + quadratic classifier | 3.3 |
| 1000 RBF + linear classifier | 3.6 |
| K-NN, Tangent Distance, 16x16 | 1.1 |
| SVM deg 4 polynomial | 1.1 |
| Reduced Set SVM deg 5 polynomial | 1.0 |
| Virtual SVM deg 9 poly [distortions] | 0.8 |
| 2-layer NN, 300 hidden units | 4.7 |
| 2-layer NN, 300 HU, [distortions] | 3.6 |
| 2-layer NN, 300 HU, [deskewing] | 1.6 |
| 2-layer NN, 1000 hidden units | 4.5 |

| | |
|---|---|
| 2-layer NN, 1000 HU, [distortions] | 3.8 |
| 3-layer NN, 300+100 hidden units | 3.05 |
| 3-layer NN, 300+100 HU [distortions] | 2.5 |
| 3-layer NN, 500+150 hidden units | 2.95 |
| 3-layer NN, 500+150 HU [distortions] | 2.45 |
| LeNet-1 [with 16x16 input] | 1.7 |
| LeNet-4 | 1.1 |
| LeNet-4 with K-NN instead of last layer | 1.1 |
| LeNet-4 with local learning instead of ll | 1.1 |
| LeNet-5, [no distortions] | 0.95 |
| LeNet-5, [huge distortions] | 0.85 |
| LeNet-5, [distortions] | 0.8 |
| Boosted LeNet-4, [distortions] | 0.7 |
| K-NN, shape context matching | 0.67 |

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

## 4) Experimental design

To construct the dataset, we performed the following steps:
- We selected a random subset of the "four" and "nine" patterns from the training and test sets of the MNIST.
- We normalized the database so that the pixel values would be in the range [0, 1]. We thresholded values below 0.5 to increase data sparsity.
- We constructed a feature set, which consists of the original variables (normalized pixels) plus a randomly selected subset of products of pairs of variables. The pairs were sampled such that each pair member is normally distributed in a region of the image slightly biased upwards. The rationale beyond this choice is that pixels that are discriminative of the "four/nine" separation are more likely to fall in that region (See Figure B2).
- We eliminated all features that had only zero values.
- Of the remaining features, we selected all the original pixels and complemented them with pairs to attain the number of 2500 features.
- Another 2500 pairs were used to construct "probes": the values of the features were individually permuted across patterns (column randomization). In this way we obtained probes that are similarly distributed to the other features.
- We randomized the order of the features.
- We quantized the data to 1000 levels.

- The data set was split into training, validation, and test set, by putting an equal amount of patterns of each class in every set.

In spite of the fact that the data is rather sparse (about 13% of the values are non-zero), we saved the data as a non-sparse matrix because we found that it can be compressed better in this way.

**Figure B2: Example of a randomly selected subset of pixels in the region of interest.**
Pairs of pixels used as features in dataset B use pixels drawn randomly according to such a distribution.

### 5) Number of examples and class distribution

|  | Positive ex. | Negative ex. | Total | Check sum |
|---|---|---|---|---|
| **Training set** | 3000 | 3000 | 6000 | 3197297133 |
| **Validation set** | 500 | 500 | 1000 | 529310977 |
| **Test set** | 3250 | 3250 | 6500 | 3404549076 |
| **All** | 6750 | 6750 | 13500 | 7131157186 |

### 6) Type of input variables and variable statistics

| Real variables | Random probes | Total |
|---|---|---|
| 2500 | 2500 | 5000 |

All variables are **integer** quantized on 1000 levels. There are **no missing values**. The data is rather **sparse**. Approximately 13% of the entries are non zero. The data was saved as a **non-sparse** matrix, because it compresses better in that format.

### 7) Results of the runs of the lambda and baseline methods

Before the benchmark, we ran some simple methods to determine what an appropriate number of examples should be. The "lambda" method (provided with the sample code) had approximately a 30% test error rate and a linear SVM trained on all features a 3.5% error rate.

The rule of thumb number_of_test_examples=100/test_errate=100/ 0.035= 2857. However, other explorations we made with on-linear SVMs and the examination of previous performances obtained on the entire MNIST dataset indicate that the best error rates could be below 2%. A test set of 6500 example should allow error rates as low as 1.5%. This motivated our test set size choice. The best benchmark error rates confirmed that our estimate was just right.

## Dataset C: DEXTER

### 1) Topic

The task of DEXTER is to filter **text**s about "corporate acquisitions". This is a two-class classification problem with sparse continuous input variables.

### 2) Sources

#### a. Original owners

The original data set we used is a subset of the well-known Reuters text categorization benchmark. The data was originally collected and labeled by Carnegie Group, Inc. and Reuters, Ltd. in the course of developing the CONSTRUE text categorization system.  It is hosted by the UCI KDD repository: http://kdd.ics.uci.edu/databases/reuters21578/reuters21578.html. David D. Lewis is hosting valuable resources about this data (see http://www.daviddlewis.com/resources/testcollections/reuters21578/). We used the "corporate acquisition" text classification class pre-processed by Thorsten Joachims <thorsten@joachims.org>. The data is one of the examples of the software package SVM-Light., see http://svmlight.joachims.org/. The example can be downloaded from ftp://ftp-ai.cs.uni-dortmund.de/pub/Users/thorsten/svm_light/examples/example1.tar.gz.

#### b. Donor of database

This version of the database was prepared for the NIPS 2003 variable and feature selection benchmark by Isabelle Guyon, 955 Creston Road, Berkeley, CA 94708, USA (isabelle@clopinet.com).

#### c. Date received: August 2003.

### 3) Past usage

Hundreds of articles have appeared on this data. For a list see: http://kdd.ics.uci.edu/databases/reuters21578/README.txt

Also, 446 citations including "Reuters" were found on CiteSeer:
http://citeseer.nj.nec.com.

### 4)  Experimental design

The original data formatted by Thorsten Joachims is in the "bag-of-words"
representation. There are 9947 features (of which 2562 are always zeros for all the
examples) that represent frequencies of occurrence of word stems in text. Some
normalizations have been applied that are not detailed by Thorsten Joachims in his
documentation. The task is to learn which Reuters articles are about "corporate
acquisitions".

The frequency of appearance of words in text is known to follow approximately Zipf's
law (for details, see e.g. http://linkage.rockefeller.edu/wli/zipf/). According to that law,
the frequency of occurrence of words, as a function of the rank k when the rank is
determined by the frequency of occurrence, is a power-law function $P_k \sim 1/k^a$ with the
exponent $a$ close to unity.
We estimated that a=0.9 gives us a reasonable approximation of the distribution of the
data (see Figures C.1 and C.2).

**Figure C.1: Comparison of the real data and the random probe data distributions.**
We plot the **number** of non-zero values of a given feature as a function of the rank of the
feature. The rank is given by the number of non-zero features. Red: real data. Blue:
simulated data.

**Figure C.2: Comparison of the real data and the random probe data distributions.** We plot the **sum** of non-zero values of a given feature as a function of the rank of the feature. The rank is given by the number of non-zero features. Red: real data. Blue: simulated data.

The following steps were taken to prepare our version of the dataset:
- We concatenated the original training set (2000 examples, class balanced) and test set (600 examples, class balanced).
- We added to the original 9947 features, 10053 features drawn at random according to Zipf law, to obtain a total of 20000 features. Fraction of non-zero values in the real data: 0.46%. Fraction of non-zero values in the simulated data: 0.5%.
- The feature values were quantized to 1000 levels.
- The order of the features and the order of the patterns were randomized.
- The data was split into training, validation, and test sets, with balanced numbers of examples of each class in each set.

5) **Number of examples and class distribution**

|                | Positive ex. | Negative ex. | Total | Check sum |
|----------------|-------------:|-------------:|------:|----------:|
| **Training set**   | 150  | 150  | 300  | 2885106  |
| **Validation set** | 150  | 150  | 300  | 2887313  |
| **Test set**       | 1000 | 1000 | 2000 | 18992356 |
| **All**            | 1300 | 1300 | 2600 | 24764775 |

### 6) Type of input variables and variable statistics

| Real variables | Random probes | Total |
|---|---|---|
| 9947 | 10053 | 20000 |

All variables are **integer** quantized on 1000 levels. There are **no missing values**. The data is very **sparse**. Approximately 0.5% of the entries are non zero. The data was saved as a **sparse-integer** matrix.

### 7) Results of the run of the lambda method and linear SVM

Before the benchmark, we ran some simple methods to determine what an appropriate number of examples should be. The "lambda" method (provided with the sample code) had approximately a 20% test error rate and a linear SVM trained on all features a 5.8% error rate.

The rule of thumb number_of_test_examples=100/test_errate=100/ 0.058= 1724 made it likely that 2000 test examples will be sufficient to obtains statistically significant results. The benchmark test results confirmed that this estimate was correct.

## Dataset D: DOROTHEA

### 1) Topic

The task of DOROTHEA is to predict which compounds bind to **Thrombin**. This is a two-class classification problem with sparse binary input variables.

### 2) Sources

    a. Original owners

The dataset with which DOROTHEA was created is one of the KDD (Knowledge Discovery in Data Mining) Cup 2001. The original dataset and papers of the winners of the competition are available at: http://www.cs.wisc.edu/~dpage/kddcup2001/. DuPont Pharmaceuticals graciously provided this data set for the KDD Cup 2001 competition. All publications referring to analysis of this data set should acknowledge DuPont Pharmaceuticals Research Laboratories and KDD Cup 2001.

    b. Donor of database

This version of the database was prepared for the NIPS 2003 variable and feature selection benchmark by Isabelle Guyon, 955 Creston Road, Berkeley, CA 94708, USA (isabelle@clopinet.com).

    c. Date received: August 2003.

### 3) Past usage

    a. References

There were 114 participants to the competition that turned in results. The winner of the competition is Jie Cheng (Canadian Imperial Bank of Commerce). His presentation is available at: http://www.cs.wisc.edu/~dpage/kddcup2001/Hayashi.pdf.

The data was also studied by Weston and collaborators:

J. Weston, F. Perez-Cruz, O. Bousquet, O. Chapelle, A. Elisseeff and B. Schoelkopf. "Feature Selection and Transduction for Prediction of Molecular Bioactivity for Drug Design". Bioinformatics.
At lot of information is available from Jason Weston's web page, including valuable statistics about the data:
http://www.kyb.tuebingen.mpg.de/bs/people/weston/kdd/kdd.html.

#### b. Synopsis of the original data

One binary attribute (active A or inactive I) must be predicted.

Drugs are typically small organic molecules that achieve their desired activity by binding to a target site on a receptor. The first step in the discovery of a new drug is usually to identify and isolate the receptor to which it should bind, followed by testing many small molecules for their ability to bind to the target site. This leaves researchers with the task of determining what separates the active (binding) compounds from the inactive (non-binding) ones. Such a determination can then be used in the design of new compounds that not only bind, but also have all the other properties required for a drug (solubility, oral absorption, lack of side effects, appropriate duration of action, toxicity, etc.).

The original training data set consisted of 1909 compounds tested for their ability to bind to a target site on thrombin, a key receptor in blood clotting. The chemical structures of these compounds are not necessary for our analysis and were not included. Of the training compounds, 42 are active (bind well) and the others are inactive. To simulate the real-world drug design environment, the test set contained 634 additional compounds that were in fact generated based on the assay results recorded for the training set. Of the test compounds, 150 bind well and the others are inactive. The compounds in the test set were made after chemists saw the activity results for the training set, so the test set had a higher fraction of actives than did the training set in the original data split.

Each compound is described by a single feature vector comprised of a class value (A for active, I for inactive) and 139,351 binary features, which describe three-dimensional properties of the molecule. The definitions of the individual bits are not included we only know that they were generated in an internally consistent manner for all 1909 compounds. Biological activity in general, and receptor binding affinity in particular, correlate with various structural and physical properties of small organic molecules. The task is to determine which of these properties are critical in this case and to learn to accurately predict the class value.

In evaluating the accuracy, a differential cost model was used, so that the sum of the costs of the actives will be equal to the sum of the costs of the inactives.

#### c. Results

To outperform these results, the paper of Weston et al., 2002, utilizes the combination of an efficient feature selection method and a classification strategy that capitalizes on the differences in the distribution of the training and the test set. First they select a small number of relevant features (less than 40) using an unbalanced correlation score:

$$f_j = \sum_{y_i=1} X_{ij} - \lambda \sum_{y_i=-1} X_{ij}$$

where the score for feature j is $f_j$, the training data is a matrix X where the columns are the features and the examples are the rows, and a larger score is assigned to a higher rank. The coefficient $\lambda$ is a positive constant. The authors suggest to take $\lambda > 3$ to select features

that have non-zero entries only for positive examples. This score encodes the prior information that the data is unbalanced and that only positive correlations are likely to be useful. The score has an information theoretic motivation, see the paper for details.

### 4) Experimental design

The original data set was modified for the purpose of the feature and variable selection benchmark:
- The original training and test sets were merged.
- The features were sorted according to the $f_j$ criterion with $\lambda=3$, computed using the original test set (which is richer is positive examples).
- Only the top ranking 100000 original features were kept.
- The all zero patterns were removed, except one that was given label –1.
- For the second half lowest ranked features, the order of the patterns was individually randomly permuted (in order to create "random probes").
- The order of the patterns and the order of the features were globally randomly permuted to mix the original training and the test patterns and remove the feature order.
- The data was split into training, validation, and test set while respecting the same proportion of examples of the positive and negative class in each set.

We are aware that out design biases the data in favor of the selection criterion $f_j$. It remains to be seen however whether other criteria can perform better, even with that bias.

### 5) Number of examples and class distribution

|                | Positive ex. | Negative ex. | Total | Check sum |
|----------------|--------------|--------------|-------|-----------|
| **Training set**   | 78           | 722          | 800   | 713978    |
| **Validation set** | 34           | 316          | 350   | 330556    |
| **Test set**       | 78           | 722          | 800   | 731829    |
| **All**            | 190          | 1760         | 1950  | 1776363   |

We mapped Active compounds to the target value +1 (positive examples) and Inactive compounds to the target value –1 (negative examples). We provide in the last column the total number of non-zero values in the data sets.

### 6) Type of input variables and variable statistics

| Real variables | Random probes | Total   |
|----------------|---------------|---------|
| 50 000         | 50 000        | 100 000 |

All variables are **binary**. There are **no missing values**. The data is very **sparse**. Less than 1% of the entries are non zero (1776363/ (1950*100000)). The data was saved as a **sparse-binary** matrix.

The following table summarizes the number of non-zero features in various categories of examples in the entire data set.

| Type | Min | Max | Median |
|---|---|---|---|
| Positive examples | 687 | 11475 | 846 |
| Negative examples | 653 | 3185 | 783 |
| All | 653 | 11475 | 787 |

### 7) Results of the run of the lambda method

Before the benchmark, we ran some simple methods to determine what an appropriate number of examples should be. The "lambda" method (provided with the sample code) had a 21% test error rate. We chose this method because it outperformed methods used in the KDD benchmark on this dataset, according to the paper of Weston et al and we could not outperform it with the linear SVM.

The rule of thumb number_of_test_examples=100/test_errate=100/0.21=476 made it likely that 800 test examples will be sufficient to obtains statistically significant results. This was slightly underestimated: the best benchmark results are around 11% error, thus 900-1000 test examples would have been better.

## Dataset E: MADELON

### 1) Topic

The task of MADELON is to classify **random** data. This is a two-class classification problem with sparse binary input variables.

### 2) Sources

The data is synthetic. It was generated by the program hypercube_data.m, which is appended.

### 3) Past usage

None, although the idea of the program is inspired by:
Grafting: Fast, Incremental Feature Selection by Gradient Descent in Function Space
*Simon Perkins, Kevin Lacker, James Theiler*; JMLR, 3(Mar):1333-1356, 2003.
http://www.jmlr.org/papers/volume3/perkins03a/perkins03a.pdf

### 4) Experimental design

To draw random data, the program takes the following steps:
- Each class is composed of a number of Gaussian clusters. N(0,1) is used to draw for each cluster num_useful_feat examples of independent features.
- Some covariance is added by multiplying by a random matrix A, with uniformly distributed random numbers between -1 and 1.
- The clusters are then placed at random on the vertices of a hypercube in a num_useful_feat dimensional space. The hypercube vertices are placed at values ± class_sep.
- Redundant features are added. They are obtained by multiplying the useful features by a random matrix B, with uniformly distributed random numbers between -1 and 1.
- Some of the previously drawn features are repeated by drawing randomly from useful and redundant features.

- Useless features (random probes) are added using N(0,1).
- All the features are then shifted and rescaled randomly to span 3 orders of magnitude.
- Random noise is then added to the features according to N(0,0.1).
- A fraction flip_y of labels are randomly exchanged.

To illustrate how the program works, we show a small example generating a XOR-type problem. There are only 2 relevant features, 2 redundant features, and 2 repeated features. Another 14 random probes were added. A total of 100 examples were drawn (25 per cluster). Ten percent of the labels were flipped.

In Figure E.1, we show all the scatter plots of pairs of features, for the useful and redundant features. For the two first features, we recognize a XOR-type pattern. For the last feature, we see that after rotation, we get a feature that alone separates the data pretty well.

In Figure E.2, we show the heat map of the data matrix. In Figure E.3, we show the same matrix after random permutations of the rows and columns and grouping of the examples per class. We notice that the data looks pretty much like white noise to the eye.

We then drew the data used for the benchmark with the following choice of parameters:

| | |
|---|---|
| num_class=2; | % Number of classes. |
| num_pat_per_cluster=250; | % Number of patterns per cluster. |
| num_useful_feat=5; | % Number of useful features. |
| num_clust_per_class=16; | % Number of cluster per class. |
| num_redundant_feat=5; | % Number of redundant features. |
| num_repeat_feat=10; | % Number of repeated features. |
| num_useless_feat=480; | % Number of useless features. |
| class_sep=2; | % Scaling factor controlling cluster separation. |
| flip_y = 0.01; | % Fraction of flipped labels. |

Figure E.4 and E.5 show the appearance of the data.

**Figure E.1: Scatter plots of the XOR-type example data for pairs of useful and redundant features.** Histograms of the examples for the corresponding features are shown on the diagonal.

**Figure E.2: Heat map of the XOR-type example data.** We show all the coefficients of the data matrix. The intensity indicates the magnitude of the coefficients. The color indicates the sign. In lines, we show the 100 examples drawn (25 per cluster). I columns, we show the 20 features. Only the first 6 ones are relevant: 2 useful, 2 redundant, 2 repeated. The data have been shifted and scaled by column to look "more natural". The last column shows the target values, with some "flipped" labels.

After adding noise

**Figure E.3: Heat map of the XOR-type example data.** This is the same matrix as the one shown in Figure E.2. However, the examples have been randomly permuted and grouped per class. The features have also been randomly permuted. Consequently, after normalization, the data looks very uninformative to the eye.

**Figure E.4: Scatter plots of the benchmark data for pairs of useful and redundant features.** We can see that the two classes overlap completely in all pairs of features. This is normal because 5 dimensions are needed to separate the data.

**Figure E.5: Heat map of the benchmark data for the relevant features (useful, redundant, and repeated).** We see the clustered structure of the data.

## 5) Number of examples and class distribution

|                | Positive ex. | Negative ex. | Total | Check sum |
|----------------|-------------:|-------------:|------:|----------:|
| **Training set** | 1000 | 1000 | 2000 | 488026911 |
| **Validation set** | 300 | 300 | 600 | 146425645 |
| **Test set** | 900 | 900 | 1800 | 439236341 |
| **All** | 2200 | 2200 | 4400 | 1073688897 |

Two additional test sets of the same size were drawn similarly and reserved to be able to test the features selected by the benchmark participants, in case it becomes important to make sure they trained only on those features.

## 6) Type of input variables and variable statistics

| Real variables | Random probes | Total |
|----------------|---------------|-------|
| 20 | 480 | 500 |

All variables are **integer**. There are **no missing values**. The data is **not sparse**. The data was saved as a **non-sparse** matrix.

## 7) Results of the run of the lambda method

Before the benchmark, we ran some simple methods to determine what an appropriate number of examples should be. The "lambda" method (provided with the sample code) performs rather poorly on this highly non-linear problem (41% error).

We used the K-nearest neighbor method, with K=3, with only the 5 useful features. With the 2000 training examples and 2000 test examples, we obtained 10% error.

The rule of thumb number_of_test_examples=100/test_errate=100/0.1=1000 makes it likely that 1800 test examples will be sufficient to obtains statistically significant results. The benchmark results confirmed that this was a good (conservative) estimate.

## Appendix A: Matlab code of the lambda method

```
function idx=lambda_feat_select(X, Y, num)
%idx=lambda_feat_select(X, Y, num)
% Feature selection method that ranks according to the dot
% product with the target vector. Note that this criterion
% may not deliver good results if the features are not
% centered and normalized with respect to the example distribution.

% Isabelle Guyon -- August 2003 -- isabelle@clopinet.com

fval=Y'*X;
[sval, si]=sort(-fval);
idx=si(1:num);
```

```matlab
function [W,b]=lambda_classifier(X, Y)
%[W,b]=lambda_classifier(X, Y)
% This simple but efficient two-class linear classifier
% of the type Y_hat=X*W'+b
% was invented by Golub et al.
% Inputs:
% X -- Data matrix of dim (num examples, num features)
% Y -- Output matrix of dim (num examples, 1)
% Returns:
% W -- Weight vector of dim (1, num features)
% b -- Bias value.

% Isabelle Guyon -- August 2003 -- isabelle@clopinet.com

Posidx=find(Y>0);
Negidx=find(Y<0);
Mu1=mean(X(Posidx,:));
Mu2=mean(X(Negidx,:));
Sigma1=std(X(Posidx,:),1);
Sigma2=std(X(Negidx,:),1);
W=(Mu1-Mu2)./(Sigma1+Sigma2);
B=(Mu1+Mu2)/2;
b=-W*B';
```

## Appendix B: Matlab code for generating synthetic data

```matlab
function [XP,YP,ixrp,iyrp, xrp,yrp,all_C,A,B,rf,shift,scale ] =
hypercube_data(num_class, num_useful_feat, num_clust_per_class,
num_pat_per_cluster, num_redundant_feat, num_repeat_feat, num_useless_feat,
class_sep, flip_y, num_repeat_val, rnd, debug, xrp, yrp,all_C,A,B,rf,shift,scale)
%[XP,YP,ixrp,iyrp, xrp,yrp,all_C,A,B,rf,shift,scale ] = hypercube_data(num_class 1,
num_useful_feat 2, num_clust_per_class 3, num_pat_per_cluster 4, num_redundant_feat
5, num_repeat_feat 6, num_useless_feat 7, class_sep 8, flip_y 9, num_repeat_val 10, rnd
11, debug 12, xrp 13, yrp 14, all_C 15, A 16, B 17,rf 18,shift 19,scale 20)
% Draws a pattern recognition problem at random, for a num_class-class problem.
% Useful features:
%   Each class is composed of a number of Gaussian clusters that are on the
%   vertices of a hypercube in a subspace of dimension num_useful_feat.
%   N(0,1) is used to draw the examples of independent features for each cluster.
%   Some covariance is added by multiplying by a random matrix A,
%   with uniformly distributed random numbers between -1 and 1.
%   The clusters are then placed on the hypercube vertices.
%   The hypercube vertices are placed at values +-class_sep.
% Redundant features:
%   Useful features are multiplied by a random matrix B,
%   with uniformly distributed random numbers between -1 and 1.
```

% Repeated features:
%   Drawn randomly from useful and redundant features.
% Useless features:
%   Additional features drawn at random not related to the concept.
% Features are then shifted and rescaled randomly to span 3 orders of magnitude.
% Random noise is then added to the features according to N(0,.1) to create several
replicates.
% if flip_y is provided, a random fraction flip_y of labels are randomly exchanged.
% -- Aknowledgements: The idea is inspired by the work of Simon Perkins.
% Inputs:
%  num_class          -- Number of classes
%  num_useful_feat      -- Number of features initially drawn to explain the concept
%  num_clust_per_class  -- Number of cluster per class
%  num_pat_per_cluster  -- Number of patterns per cluster // all balanced for now, can be
generalized to imbalanced classes (can take subset of samples of each class)
%  num_redundant_feat   -- Number of features linearly dependent upon the useful
features
%  num_repeat_feat      -- Number of features repeating the previous ones (drawn at
random)
%  num_useless_feat     -- Number of features dran at random regardless of class label
information
%  class_sep          -- Factor multiplying the hypercube dimension.
%  flip_y             -- Fraction of y labels to be randomly exchanged.
%  num_repeat_val      -- number of times each entry is repeated (modulo some noise).
%  rnd               -- Flag to enable or disable random permutations.
%  debug             -- 0/1 flag.
% Returns:
%  XP                -- Matrix (num_pat, num_feat, num_repeat_val) of randomly permuted
features
%  YP                -- Vector of 0,1...num_class target class labels (in random order, to be
used eventually for clustering)
%  ixrp              -- permutation matrix to be used to restore the original feature order
%  iyrp              -- permutation matrix to be used to restore the original pattern order
(class labels of the same class are consecutive
%                     and there are the same number of example per class, before label
corruption)
%                     Y=YP(iyrp); X=XP(iyrp,ixrp);
% all_C             -- A matrix 2^num_useful_feat*num_useful_feat of
%                     hypercube vertices where to place the cluter centers.
% A                -- Matrix used to correlate the useful features.
% B                -- Matrix used to create dependent (redundant) features.
% rf               -- Indices of repeated features.
% shift            -- Shift applied.
% scale            -- Scale applied.

% Isabelle Guyon -- July 2003 -- isabelle@clopinet.com

```
if nargin<8, class_sep=1; end
if nargin<9, flip_y=0; end
if nargin<10, num_repeat_val=1; end
if nargin<11, rnd=0; end % disable random permutation
if nargin<12, debug=0; end
if nargin<13, xrp=[]; end
if nargin<14, yrp=[]; end
if nargin<15, all_C=[]; end
if nargin<16, A={}; end
if nargin<17, B=[]; end
if nargin<18, rf=[]; end
if nargin<19, shift=[]; end
if nargin<20, scale=[]; end

% Count features and patterns
num_feat=num_useful_feat + num_repeat_feat + num_redundant_feat +
num_useless_feat;
num_pat_per_class=num_pat_per_cluster*num_clust_per_class;
num_pat=num_pat_per_class*num_class;
X=zeros(num_pat, num_feat);

% Attribute class labels
y=0:num_class-1;
Y=repmat(y, num_pat_per_class, 1);
Y=Y(:);

% Hypercube design
is_XOR=0;
if num_useful_feat==2 & num_class==2 & num_clust_per_class==2,
   is_XOR=1;
   all_C=[-1 -1; 1 1; 1 -1; -1 1]; % XOR
else
   if isempty(all_C)
      fprintf('New C\n');
      all_C=2*ff2n(num_useful_feat)-1;
      rndidx=randperm(size(all_C,1));
      all_C=all_C(rndidx,:);
   end
end

% Draw A
if isempty(A)
   fprintf('New A\n');
   for k=1:num_class*num_clust_per_class
      A{k} = 2*rand(num_useful_feat, num_useful_feat)-1;
```

```matlab
        end
end
% Loop over all clusters
for k=1:num_class*num_clust_per_class
    % define the range of patterns of that cluster
    kmin=(k-1)*num_pat_per_cluster+1;
    kmax=kmin+num_pat_per_cluster-1;
    kidx=kmin:kmax;
    % Draw n features independently at random
    X(kidx,1:num_useful_feat)=random('norm', 0, 1, num_pat_per_cluster,
num_useful_feat);
    % Multiply by a random matrix to create some co-variance of the features
    X(kidx,1:num_useful_feat)=X(kidx,1:num_useful_feat)*A{k};
    % Shift the center off zero to separate the clusters
    C=all_C(k,:)*class_sep;
    X(kidx,1:num_useful_feat) = X(kidx,1:num_useful_feat) + repmat(C,
num_pat_per_cluster, 1);
end

if debug,
    featdisplay(normalize_data([X(:,1:num_useful_feat),Y])); title('Useful features');
    figure; scatterplot(X(:, 1:num_useful_feat), Y); title('Useful features');
end

% Create redundant features by multiplying by a random matrix
if isempty(B),
    fprintf('New B\n');
    B = 2*rand(num_useful_feat, num_redundant_feat)-1;
end
X(:,num_useful_feat+1:num_useful_feat+num_redundant_feat)=X(:,1:num_useful_feat)
*B;

if debug,
    featdisplay(normalize_data([X(:,1:num_useful_feat+num_redundant_feat),Y]));
title('Useful+redundant features');
    figure; scatterplot(X(:, 1:num_useful_feat+num_redundant_feat), Y);
title('Useful+redundant features');
end

% Repeat num_repeat_feat features, chosen at random among useful and redundant feat
nf=num_useful_feat+num_redundant_feat;
if isempty(rf)
    fprintf('New rf\n');
    rf=round(1+rand(num_repeat_feat,1)*(nf-1));
end
X(:,nf+1:nf+num_repeat_feat)=X(:,rf);
```

```matlab
if debug,
featdisplay(normalize_data([X(:,1:num_useful_feat+num_redundant_feat+num_repeat_f
eat),Y]));
    title('Useful+redundant+repeated features');
end

% Add useless features : these are uncorrelated with one another, but could be correlated
:=)
X(:,num_feat-num_useless_feat+1:num_feat)=random('norm', 0, 1, num_pat,
num_useless_feat);

if debug,
    featdisplay(normalize_data([X,Y]));
    title('All features');
end

% Add random y label errors
num_err_pat = round(num_pat*flip_y);
rp=randperm(num_pat);
fi=rp(1:num_err_pat);
Y(fi)=mod(Y(fi)+round(rand(num_err_pat,1)*(num_class-1)), num_class);

if debug,
    featdisplay(normalize_data([X,Y]));
    title('All features + flipped labels');
end

% Randomly shift and scale
if isempty(shift)
    fprintf('New shift\n');
    shift=rand(num_feat,1);
end
if isempty(scale)
    fprintf('New scale\n');
    scale=1+100*rand(num_feat,1);
end
X=X+repmat(shift',num_pat,1);
X=X.*repmat(scale',num_pat,1);

if debug,
    featdisplay([X,100*normalize_data(Y)]);
    title('All features + flipped labels + scale shifted');
end

% Randomly permute the features and patterns
```

```matlab
if isempty(xrp)
   fprintf('New xrp, yrp\n');
   if rnd
      xrp=randperm(num_feat);
      yrp=randperm(num_pat);
         else
      xrp=1:num_feat;
      yrp=1:num_pat;
         end
end
XP0=X(yrp,xrp);
YP=Y(yrp);

if debug,
   [ys,pattidx]=sort(YP);
   featdisplay(normalize_data([XP0(pattidx,:),YP(pattidx)]));
   title('After permutation and data normalization');
end

% Create inverse random indices
ixrp(xrp)=1:num_feat;
iyrp(yrp)=1:num_pat;

% Create several replicates by adding a little bit of random noise
XP=zeros(num_pat, num_feat, num_repeat_val);
for k=1:num_repeat_val
   N=random('norm', 0, .1*sqrt(num_repeat_val), num_pat, num_feat);
   XP(:,:,k)=XP0.*(1+N);
end

if debug,
   featdisplay(normalize_data([XP(pattidx,:),YP(pattidx)]));
   title('After adding noise');
end
```

## Footnotes

[1] In this document, we do not make a distinction between features and variables. The benchmark addresses the problem of selecting input variables. Those may actually be features derived from the original variables using a preprocessing.