[Reviews · NeurIPS 2020]

Review 1

Summary and Contributions: I have read the response and the discussion of reviewers. The reviewer learned that there exist a line of prior theoretical works about sketching-based IPM which are very relevant to this work and which deserve more detailed discussion and ideally they could be included as comparison in experiments, to validate the practical advantageous as claimed in the "prior work" section. In general the reviewer believes that this is a good piece of work with new theoretical contribution in a subclass of IPM with infeasible start, but current version fall short a bit in numerical side which makes the paper borderline, considering the fact that NeurIPS exercises a high standard. =========================================== This paper propose a novel sketching-based infeasible IPM with preconditioned conjugate gradient for efficiently solving linear programming tasks. A theoretical convergence analysis is provided, showing the same convergence rate as standard infeasible IPM. Numerical reuslts demonstrates the computational efficiency of this approach due to sketching comparing to standard infeasible IPM.

Strengths: The paper provides a new practical approach for designing fast IIPM, using sketching techniques to perform dimensionality reduction for computational efficiency. The theoretical results showing that the IIPM with sketching have the same convergence as standard IIPM, are sound and valuable for the community.

Weaknesses: The numerical study in the current version is limited, since it only compare to standard IPM, while as cited (reference 41, 51) there are already a number of sketching-based algorithms which are readily applicable to LP. The author(s) only argue that these methods' performance in high-dimension regime are unclear -- then why not show it numerically? Another issue is that the author(s) use Gaussian sketch in experiments, which the review believes that it is not a good choice -- Gaussian sketch, although admits good theoretical property, is not recommended in practice since it is computationally expensive. In practice some efficient sketching method like the count-sketch, randomized orthongonal system sketch (fast Johnson-Lindenstrauss) are used for sketching-based frameworks. The reviewer would suggest the author(s) to include a set of experiments using these practical sketching schemes.

Correctness: The proof seems correct after a quick check.

Clarity: Yes, the paper is clearly presented and easy to follow.

Relation to Prior Work: Yes, the relationships to previous works are clearly and sufficiently discussed, and presented in details.

Reproducibility: Yes

Additional Feedback: If the author(s) can address the issue mentioned above, the review will consider to increase the score.


Review 2

Summary and Contributions: The paper proposes a solver for linear programming (LP). The solver uses Interior Point Methods (IPM) and solves the linear system -arising at each IPM iteration- via a preconditioned conjugate gradient. The novelty lies in the design of a new preconditioner based on matrix sketching. The paper also proposes a correction technique (to compensate for the error introduced when computing approximate solutions to the linear system) that ensures convergence to a feasible and approximately optimal solution of the LP.

Strengths: - linear programming is important and developing faster/more scalable solvers may impact several ML applications. - the idea of using matrix sketching to compute the preconditioner for eq (3) seems reasonable for wide/tall systems. - the use of the same sketching matrix in both the preconditioning and the correction vector v is clever and efficient. - the technical results look correct and the derivation is sound. - the paper is clear and easy to read.

Weaknesses: - the last paragraph in the “Prior Work” section, focusing on sketching techniques, does not do a great job at pointing out novelty. It talks about sketching being used for different problems, but it is unclear if those related works also focused on solving linear systems (maybe in a different context) and can be directly applied to eq (3). - related to the previous point: relevant references seem to be missing. For instance: [Faster Kernel Ridge Regression Using Sketching and Preconditioning Haim Avron, Kenneth L. Clarkson, and David P. Woodruff, SIAM Journal on Matrix Analysis and Applications 2017 38:4, 1116-1138.] [Preconditioning Kaczmarz method by sketching, Alexandr Katrutsa, Ivan Oseledets, https://arxiv.org/pdf/1903.01806.pdf] - the experimental evaluation only compares the proposed approach against an un-preconditioned conjugate gradient (CG) method. As expected, un-preconditioned CG takes many inner iterations. Why not comparing against standard preconditioners and possibly evaluate timing? Currently, the reader cannot figure out if or when the proposed approach is convenient over existing preconditioned CG methods.

Correctness: The claims and method look correct. The evaluation is correct, but incomplete.

Clarity: The paper is well written.

Relation to Prior Work: Matrix sketching has been used to get a preconditioner for linear systems in related work (including [Faster Kernel Ridge Regression Using Sketching and Preconditioning Haim Avron, Kenneth L. Clarkson, and David P. Woodruff, SIAM Journal on Matrix Analysis and Applications 2017 38:4, 1116-1138]). The paper does not clarify why the proposed idea is fundamentally novel.

Reproducibility: Yes

Additional Feedback: - typos: line 34 “defintion” - Line 128: “amplified using standard techniques” calls for a reference. - line 200 says “W […], which we construct as discussed in Section 1.3”, but that section does not clarify how to construct W, only states its existence. - typo: line 288 ”IIPM” POST REBUTTAL: After reading the reviews and the rebuttal, I still think this is a good paper. However, I think the experimental results could be stronger and I see that R1 and R3 share similar concerns. Moreover, the authors did not really address my questions about novelty with respect to related work. I still think this is an accept, but I'm going to change my score to "accept" rather than "strong accept".


Review 3

Summary and Contributions: This paper considers the problem of solving linear programs with infeasible-start long-step interior point methods (IPMs), a prominent class of linear programming methods. The paper provides a method for approximately implementing the steps of such an IPM, i.e. approximately solving a linear system, in nearly linear time when the constraint matrix is sufficiently tall (/ wide) (depending on whether the primal or dual is considered). Further, the paper shows how to efficiently and simply modify the steps of the infeasible start IPM to account for the error induced by the approximate solver so that the overall number of iterations required by the interior point method is not impaired. The linear system solver used is a combination of previous subspace embedding techniques from randomized linear algebra and the conjugate gradient (CG) method and along the way the paper proves convergence results of CG that may be of further interest. Finally, the paper provides experiments which corroborate the theoretical claims of the paper. UPDATE AFTER AUTHOR FEEDBACK AND DISCUSSION: Thank you to the authors for their thoughtful response. After reading the response and further discussion, my core view of the paper is similar to what I wrote in the review. I think this is a nice result on the theory of infeasible-start primal-dual interior point methods that would be great to publish in some venue. However, for the reasons raised the ultimate novelty of the techniques used and how surprising the result is, is less clear. Further, that this paper does not appear to be improving the best theoretical complexity or best practical runtimes of linear programming, makes it difficult to raise the score.

Strengths: This paper cleanly advances the theory for infeasible-start long-step IPMs, a prominent class of IPMs, for solving tall linear programs (a prominent class of linear programs that can arise in certain ML-applications when there is an abundance of data). The paper provides a natural efficient preconditioning technique for solving the linear systems that arise when implementing the method, a simple way for handling the errors the system induces, and analysis of the performance of the resulting methods. Consequently, this paper improves upon the theory of infeasible start IPM by providing new techniques for leveraging approximate linear system solvers (previous methods were more computationally expensive) and efficiently implement them. The paper analyzes a natural linear system solving technique, combining subspace embeddings (a known powerful matrix sketching result) with conjugate gradient (a known powerful linear system solver), to achieve this result. It is also possibly of further interest how the paper combines the preconditioning technique with its technique for modifying the steps to handle error in infeasible start IPMs. Ultimately, this paper could promote further research on infeasible-start long-step IPMs and lead to faster algorithms for solving certain large scale problems.

Weaknesses: This paper provides a nice advance in the theory of infeasible-start long-step IPMs, however the novelty of the approach taken and the relation of the work in the paper to prior work could use further clarity. First, solving regression problems in an A in nearly linear time, when A has many more rows than columns has been the subject of a line of research, e.g. [8], [38], and “Iterative Row Sampling.” These results, including ones based on the subspace embedding result used in this paper, readily extend to solving linear systems in A^T A and this has been used by the Theoretical Computer Science papers mentioned for implementing short step IPMs. Consequently, I think it would have been beneficial to state earlier that the paper is using the known linear system solving machinery of subspace embeddings to build preconditioners (rather than just saying that “Randomized Linear Algebra” is used) and put this in the context of prior work. There may be novelty in the particular way in which the paper is using conjugate gradient and subspace embeddings, however the paper would be strengthened if it articulated how this is different than this previous literature; as the appendix points out, conjugate gradient can be replaced with other iterative methods which possibly puts the approach considered closer to the ones from the literature. In light of the previous paragraph, I think more of the novelty in the paper may lie in exactly how they handle the error from approximate linear system solves in a way sensitive to the design of the preconditioner. However, here I think it should be noted that prior IPM results in related spaces, e.g. short-step methods or dual methods, have studied the effect of such error. Further, from this literature that the guarantees of the long-step IPM can be preserved if systems are solved to inverse polynomial accuracy in the right norm seems reasonable. Consequently, while it is nice that the paper handles the error in such a clean way, the novelty of this approach in light of the previous work should be commented on. (The paper does touch upon why error analysis for infeasible-start long-step primal-dual methods may be more difficult than in other cases, but explaining the non-applicability of other approaches would be beneficial). Further, the paper motivates its study of infeasible-start long-step IPMs (over perhaps feasible start IPMs or short-step methods) due to its practicality and mentions how certain theoretical results on implementing steps efficiently (i.e. “inverse maintenance”) are not used in practice. Consequently, the paper would strengthened if it could argue that the method proposed achieved faster end-to-end runtimes for solving linear programs. However, while the empirical section does provide iteration bounds which corroborates its theoretical findings, it doesn’t give full runtime bounds. This is perhaps problematic as the method proposed requires computing an SVD of a matrix which could cause large runtime issues. Also, the paper mentions that [13] which provided substantial empirical experiments on using preconditioned Krylov methods, but doesn’t compare the details in a way to see if this paper is making an improvement. Consequently, ultimately the paper doesn’t seem to justify that it improves either the theory or practice for linear programming, though it does improve the theory for a prominent class of practical methods. Lastly (and perhaps more minor) there are theoretical details not considered in this work that are in others in the area and it would be beneficial to comment on. In particular, it is known that computing SVDs and applying conjugate gradient can cause numerical stability issues. Therefore, while the paper is improving the theoretical analysis of infeasible start IPMs in some ways, without discussing this the methods performance when precision is taken into account it, in theory under certain computational assumptions it may be worse. The paper would be strengthened by discussing this a little.

Correctness: As for as I can tell the paper is correct, though I have not verified the proofs.

Clarity: The paper is fairly well-written though there are a few places where a little extra writing clarify would be beneficial (as mentioned in “weaknesses” and “additional feedback” sections).

Relation to Prior Work: The paper does cite relevant work, however as mentioned in the in “weaknesses” and “additional feedback” sections there is some additional work and information that I believe should be mentioned more explicitly, e.g. that it has been known how to approximately solve regression problems in matrix A with n rows and d columns in O(nnz(a) + poly(d)) (up to log factors) since [8] and that some of these results readily extend to solving systems in A^T A.

Reproducibility: Yes

Additional Feedback: * Line 38: As discussed in the “weaknesses” section there is a prominent line of work in theory showing that these linear systems can be solved in nearly linear time when they are sufficiently large. Further, the solver in this paper is a variant of these papers (with Richardson replaced with conjugate) and this solver has been used in IPMs. This should be mentioned earlier to accurately summarize the state-of-the-art understanding of these systems in the context of IPMs. (Yes, they may not have been used as extensively in practice and it is fine for the writing to make this clear, but in my opinion, it should be mentioned.) * Line 55-56: I think it would be beneficial to be more explicit about what is being used from the (RLA) literature, e.g. by saying that subspace embeddings are used and give a citation. These are known techniques for solving linear systems and aiding the implementation of IPMs and therefore it would be beneficial to say the technique explicitly. * Line 64-65: A more explicit comparison of the steps and why this is more efficient would be helpful. * Line 153: Why is the work of [14] closer to the work of this paper than many of the other papers cited in the Theoretical Computer Science literature? Some of these works also reason about approximate linear system solving. * Line 193: I believe the result in [12] is slightly stronger in that they showed that there is a distribution over W (independent of Z) such that this condition holds with the stated probability. It might be beneficial to say this more explicitly since it will be used algorithmically later. * Line 166: What preconditioner is used in [13]? It would be helpful to compare more completely. * Line 199: It might be beneficial to state something about how the SVD can be computed efficiently. * Line 211: A SVD of a matrix can be computed more efficiently than m^3, see “Pseudospectral Shattering, the Sign Function, and Diagonalization in Nearly Matrix Multiplication Time” * Line 288: IIPM --> IPM ? * Line 479: It would interesting to justify this more completely. If in the experiments CG is replaced with Richardson or steepest descent, is the slow down considerable? While the practical issues of Richardson in general make sense, in the IPM the preconditioned system being solved is in some sense well-conditioned and therefore this is less clear.


Review 4

Summary and Contributions: The manuscript considered infeasible interior-point methods for linear programs which have way more variables than constraints (or vice-versa by the dual). The key ingredients of the algorithm are i) sketching the wide system matrix, ii) designing a preconditioner based on the sketch, and iii) applying conjugate gradient to solve the modified Newton step. The authors analyzed the iteration complexity of the proposed algorithm.

Strengths: The combination of randomized numerical linear algebra and and interior point method is interesting. The assumption of tall/wide linear programs is well motivated - many machine learning problems enjoy this property. The complexity analysis appears to be technically solid.

Weaknesses: - In several places (line 93-97, line 217 - 223), the authors claim that their convergence result for the CG in equation (7) is novel. However, it is a simple application of Theorem 8 in [5] and the well-conditionedness of their preconditioned system matrix. - The heavy lifting parts of the analysis are respectively done by the randomized numerical linear algebra literature [12], convergence analysis of conjugate gradient [5], and analysis of long-step infeasible IPMs [37]. The combination is novel though.

Correctness: The analysis appears to be correct. Both inner and outer iteration complexities are tested and reported in numerical simulation. I think it's also important to report the actual cpu time. This is especially important when comparing with the direct solver method. In Algorithm 1 and the main text around it, \Sigma_Q^{-1/2} is claimed to be the matrix of singular values for ADW. However, this doesn't seem to match the definition of Q^{-1/2} and equation (11). Should \Sigma_Q^{1/2} be the singular value matrix?

Clarity: The presentation is clear and well-organized in most parts. I would suggest including a bit more background knowledge on randomized numerical linear algebra. For example, it was not mentioned that W can be chosen in a data-oblivious manner. Since W is applied to V^T, the right singular vector matrix of AD, (see equation (12)), it gives the readers the impression that W might depend on V^T, while the latter is very expensive to compute. Some introduction on how W is generated and what it looks like would be valuable. Step 4 of Algorithm 2 mentions "step (3)". Should it be step 2? Step 5 also mentions "step (a)", should it be step 2 too?

Relation to Prior Work: - Since the algorithm has a similar spirit to those IPMs with inexact Newton step, a review on previous work in this direction would be helpful. - Can you include a reference for the claim in line 95 (residue of CG may oscillate)?

Reproducibility: Yes

Additional Feedback:

[Author Response · NeurIPS 2020]

We would like to thank the reviewers for their detailed and helpful feedback. We will address all of the typos and
make another editorial pass to incorporate additional reviewer suggestions (such as, adding relevant references etc.) to
improve the readability of the paper. We address their concerns below:

**1. Comparing experiments with [41,51] and extending beyond Gaussian sketches (R1)**: Thanks for bringing it up.
We emphasize that although we focus on a practice-oriented approach (as compared to recent TCS work), this paper
focuses on theoretical aspects of using randomized preconditioning in the context of IPM (whereas the algorithms of
[41,51] deal with second-order optimization methods and are of general-purpose), and our experiments are preliminary
proof-of-concept showing that the linear solvers in IPM can be accelerated via our technique. For the same reason, we
only considered Gaussian sketches in our experiments. We do not anticipate significant differences if other sketching
matrices are used and we will extend our experiments to accommodate sparse embeddings as well.

**2. Relation to prior works on sketching (R2, R3)**: Thanks for such insightful comments. We do believe that we
highlighted prior works on sketching and preconditioning, especially the ones dealing with solving regression problems
or linear systems, but, in light of this suggestion, we will update the 'prior works' section in the final version of the paper.
As R3 further suggested, instead of just saying RLA is used, we will state earlier in the paper that the construction of
our preconditioner depends on the known randomized linear system solving tools from subspace embeddings. In this
context, we will also include the references pointed out by R2 and R3, as well as other relevant papers that we might
have missed out.

**3. Extending our experiments (R2, R3)**: In addition to R1's suggestions on extending our empirical evaluations,
we can also include additional experiments comparing our algorithm with other standard preconditioning techniques
applied to IPMs for solving linear programs, specifically, as suggested by R3, the inner-iteration preconditioning of
[13]. As a supplement to our main numerical results and as suggested by R3, we will also include in our experiments
how the convergence gets affected in practice if CG is replaced with SD (or Richardson) in Algorithm 1. We agree with
R2 and R3 on the fact that additional experiments will further validate the efficiency of our proposed approach, but we
do believe that our experimental results are already a strong proof-of-concept that our proposed theory works well in
practice.

**4. Outlining the novelty (R3)**: Thanks for the detailed feedback. In the final version, as R3 suggested, we will attempt
to further emphasize the novelty, including the way we handled the error incurred by the iterative solver by proposing a
fast, sketching-based solution to a linear invariant that needed to be exactly satisfied. We do note that the construction
of our solution is original *and* computationally efficient. We will also revisit and perhaps improve the description of our
contributions with respect to the prior works on inexact IPMs for LP.

**5. Why is [14] closer to this paper? (R3)**: In our opinion, this is due to the fact that their objective is quite similar:
the analysis of an approximate solver in each iteration. However, important differences do exist between their work and
ours, as mentioned in our work.

**6. SVD of ADW could cause large runtime issues (R3)**: In the analysis, we use an SVD to provide a cleaner and
simpler theoretical analysis and since it does not increase the asymptotic complexity. We can replace the use of SVD
with a Cholesky factorization and significantly improve the constants.

**7. Reporting running times (R3, R4)**: We refrained from reporting running times to avoid direct comparisons
with heavily optimized benchmark LP solvers; in industrial-grade solvers the "true" algorithmic efficiency is grossly
confounded by built-in optimization strategies. For better comparisons with other standard preconditioners (please see
item 3) as well as the direct solvers, we will include the running times of our experiments in the updated version.

**8. Outlining the novelty (R4)**: As also recommended by R2 and R3 (please see item 2), we will update the "prior
work" to further emphasize the line of research on sketching and preconditioning. In addition, as per the suggestions by
R3 , we will further elaborate on the novelty of our paper with respect to handling the error due to the approximate
solver (please see item 4).

**9. Singular values of ADW (R4)**: Thanks for pointing out the typo. The matrix of the singular values of $\mathbf{ADW}$ is
indeed $\mathbf{\Sigma}_{\mathbf{Q}}^{1/2}$, not $\mathbf{\Sigma}_{\mathbf{Q}}^{-1/2}$. We will fix it in the revised version.

**10. Construction of W (R4)**: We will add a short note on how to construct $\mathbf{W}$.

**11. Citing prior works on the oscillatory behavior of CG residual (R4)**: By "...the norms of the CG residuals may
oscillate", we implied the non-monotonic decrease of the residual norms in CG. For example, in Theorem 8 of [5], if we
have $\frac{\kappa(\mathbf{A})-1}{2} > 1$ *i.e.* if $\|r_k\| \leq \beta \|r_{k-1}\|$ for $\beta > 1$, it doesn't guarantee $\|r_k\| \leq \|r_{k-1}\|$. In practice, similar behavior
of CG residuals also discussed in many papers including: *CG vs. MINRES: An Empirical Comparison* by Fong and
Saunders (2012). We will add this reference in the updated version.

[Meta-Review · NeurIPS 2020]

The paper was overall well-received, and R4 in particular liked the combination of randomized lin algebra with IPM and the solid technical analysis. R3 brought up some major points and thought of this as a borderline paper, in part because of a narrow scope of applicability. However, overall, the AC and SAC agree this is an interesting paper (as well as well-written and technically solid), and is enough to be over the bar for NeurIPS. R3 presents a concern that some of the presentation relative to past methods is a bit misleading, and this should be addressed in the minor revisions. Please see R3s review for full details.